# FairGrad: Fairness Aware Gradient Descent

**Gaurav Maheshwari**                                                        *gaurav.maheshwari@inria.fr*
*Univ. Lille, Inria, CNRS, Centrale Lille, UMR 9189 - CRIStAL, F-59000 Lille, France*

**Michaël Perrot**                                                           *michael.perrot@inria.fr*
*Univ. Lille, Inria, CNRS, Centrale Lille, UMR 9189 - CRIStAL, F-59000 Lille, France*

**Reviewed on OpenReview:** *https://openreview.net/forum?id=0f8tU3QwWD*

## Abstract

We address the problem of group fairness in classification, where the objective is to learn models that do not unjustly discriminate against subgroups of the population. Most existing approaches are limited to simple binary tasks or involve difficult to implement training mechanisms which reduces their practical applicability. In this paper, we propose Fair-Grad, a method to enforce fairness based on a re-weighting scheme that iteratively learns group specific weights based on whether they are advantaged or not. FairGrad is easy to implement, accommodates various standard fairness definitions, and comes with minimal overhead. Furthermore, we show that it is competitive with standard baselines over various datasets including ones used in natural language processing and computer vision.

FairGrad is available as a PyPI package at - https://pypi.org/project/fairgrad

## 1 Introduction

Fair Machine Learning addresses the problem of learning models that are free of any discriminatory behavior against a subset of the population. For instance, consider a company developing a model to predict whether a person would be a suitable hire based on their biography. A possible source of discrimination here can be if, in the data available to the company, individuals that are part of a subgroup formed based on their gender, ethnicity, or other sensitive attributes, are consistently labelled as unsuitable hires regardless of their true competency due to historical bias. This kind of discrimination can be measured by a fairness notion called Demographic Parity (Calders et al., 2009). If the data is unbiased, another source of discrimination may be the model itself that consistently mislabels the competent individuals of a subgroup as unsuitable hires. This can be measured by a fairness notion called Equality of Opportunity (Hardt et al., 2016).

Several such fairness notions have been proposed in the literature as different problems call for different measures. They can be divided into two major paradigms, namely (i) Individual Fairness (Dwork et al., 2012; Kusner et al., 2017) where the idea is to treat similar individuals similarly regardless of the sensitive group they belong to, and (ii) Group Fairness (Calders et al., 2009; Hardt et al., 2016; Zafar et al., 2017a; Denis et al., 2021) where the underlying idea is that no sensitive group should be disadvantaged compared to the overall reference population. In this paper, we focus on group fairness in the context of classification where we only assume access to the sensitive attributes during the training phase.

The existing approaches for group fairness in Machine Learning may be divided into three main paradigms. First, pre-processing methods aim at modifying a dataset to remove any intrinsic unfairness that may exist in the examples. The underlying idea is that a model learned on this modified data is more likely to be fair (Dwork et al., 2012; Kamiran & Calders, 2012; Zemel et al., 2013; Feldman et al., 2015; Calmon et al., 2017). Then, post-processing approaches modify the predictions of an accurate but unfair model so that it becomes fair (Kamiran et al., 2010; Hardt et al., 2016; Woodworth et al., 2017; Iosifidis et al., 2019; Chzhen et al., 2019). Finally, in-processing methods aim at learning a model that is fair and accurate in a single step (Calders & Verwer, 2010; Kamishima et al., 2012; Goh et al., 2016; Zafar et al., 2017a;b; Donini et al.,

```python
# The library is available at https://pypi.org/project/fairgrad.
from fairgrad.torch import CrossEntropyLoss

# Same as PyTorch's loss with some additional meta data.
# A fairness rate of 0.01 is a good rule of thumb for standardized data.
criterion = CrossEntropyLoss(y_train, s_train,
fairness_measure, fairness_rate=0.01)

# The dataloader and model are defined and used in the standard way.
for x, y, s in data_loader:
  optimizer.zero_grad()
  loss = criterion(model(x), y, s)
  loss.backward()
  optimizer.step()
```

Figure 1: A standard training loop where the PyTorch's loss is replaced by FairGrad's loss.

2018; Krasanakis et al., 2018; Agarwal et al., 2018; Wu et al., 2019; Cotter et al., 2019; Iosifidis & Ntoutsi, 2019; Jiang & Nachum, 2020; Lohaus et al., 2020; Roh et al., 2020; Ozdayi et al., 2021). In this paper, we propose a new in-processing group fairness approach based on a re-weighting scheme that may also be used as a kind of post-processing approach by fine-tuning existing classifiers.

**Motivation.** In-processing approaches can be further divided into several sub-categories (Caton & Haas, 2020). Common amongst them are methods that cast the fairness task as a constrained optimization problem, and then relax the fairness constraints under consideration to simplify the learning process (Zafar et al., 2017a; Donini et al., 2018; Wu et al., 2019). Indeed, standard fairness notions are usually difficult to handle due to their non-convexity and non-differentiability. Unfortunately, these relaxations may be far from the actual fairness measures, leading to sub-optimal models (Lohaus et al., 2020). Similarly, several approaches address the fairness problem by designing specific algorithms and solvers. This is, for example, done by reducing the optimization procedure to a simpler problem (Agarwal et al., 2018), altering the underlying solver (Cotter et al., 2019), or using adversarial learning (Raff & Sylvester, 2018). However, these approaches are difficult to adapt to existing systems as they require special training procedures or changes in the model. They are also limited in the range of problems to which they can be applied. For example, the work of Agarwal et al. (2018) can only be applied in a binary classification setting, while the work of Ozdayi et al. (2021) is limited to two sensitive groups. Furthermore, they may come with several hyper-parameters that need to be carefully tuned to obtain fair models. For instance, the scaling parameter in adversarial learning (Raff & Sylvester, 2018; Li et al., 2018) or the number of iterations in inner optimization for bi-level optimization based mechanisms (Ozdayi et al., 2021). The complexity of the existing methods might hinder their deployment in practical settings. Hence, there is a need for simpler methods that are straightforward to integrate into existing training loops.

**Contributions.** In this paper, we present FairGrad, a general purpose approach to enforce fairness in empirical risk minimization solved using gradient descent. We propose to dynamically update the influence of the examples after each gradient descent update to precisely reflect the fairness level of the models obtained at each iteration and guide the optimization process in a relevant direction. Hence, the underlying idea is to use lower weights for examples from advantaged groups than those from disadvantaged groups. Our method is inspired by recent re-weighting approaches that also propose to change the importance of each group while learning a model (Krasanakis et al., 2018; Iosifidis & Ntoutsi, 2019; Jiang & Nachum, 2020; Roh et al., 2020; Ozdayi et al., 2021). We discuss these works in Appendix A. Interestingly, we also find that FairGrad can be seen as solving a kind of constrained optimization problem. In Section 2.2, we expand upon this link and show how FairGrad can be seen as a solution that connects these two kinds of methods.

A key advantage of FairGrad is that it is straightforward to incorporate into standard gradient based solvers that support examples re-weighting like Stochastic Gradient Descent. Hence, we developed a Python library (provided in the supplementary material) where we augmented standard PyTorch losses to accommodate our approach. From a practitioner point of view, it means that using FairGrad is as simple as replacing their existing loss from PyTorch with our custom loss and passing along some meta data, while the rest of

the training loop remains identical. This is illustrated in Figure 1. It is interesting to note that FairGrad only brings one extra hyper-parameter, the fairness rate, besides the usual optimization ones (learning rates, batch size, . . . ). Moreover, FairGrad incurs minimal computational overhead during training as it relies on objects that are already computed for standard gradient descent, namely the predictions on the current batch and the loss incurred by the model for each example. In particular, the overhead is independent of the number of parameters of the model. Furthermore, as many in-processing approaches in fairness (Cotter et al., 2019; Roh et al., 2020), FairGrad does not introduce any overhead at test time.

Overall, FairGrad is a lightweight fairness solution that is compatible with various group fairness notions, including exact and approximate fairness, can handle both multiple sensitive groups and multiclass problems, and can fine tune existing unfair models. Through extensive experiments, we also show that, in addition to its versatility, FairGrad is competitive with several standard baselines in fairness on both standard datasets as well as complex natural language processing and computer vision tasks.

## 2 Problem Setting, Notations, and Related Work

In the remainder of this paper, we assume that we have access to a feature space $\mathcal{X}$, a finite discrete label space $\mathcal{Y}$, and a set $\mathcal{S}$ of values for the sensitive attribute. We further assume that there exists a distribution $\mathcal{D} \in \mathcal{D}_{\mathcal{Z}}$ where $\mathcal{D}_{\mathcal{Z}}$ is the set of all distributions over $\mathcal{Z} = \mathcal{X} \times \mathcal{Y} \times \mathcal{S}$. Our goal is then to learn an accurate model $h_\theta \in \mathcal{H}$, with learnable parameters $\theta \in \mathbb{R}^d$, such that $h_\theta : \mathcal{X} \to \mathcal{Y}$ is fair with respect to a given fairness definition that depends on the sensitive attribute. In Section 2.1, we formally define the family of fairness measures that are compatible with our approach and provide several examples of popular notions encompassed by our fairness definition.

As usual in machine learning, we will assume that $\mathcal{D}$ is unknown and that we only get to observe a finite dataset $\mathcal{T} = \{(x_i, y_i, s_i)\}_{i=1}^n$ of $n$ examples drawn i.i.d. from $\mathcal{D}$. Let $\mathbb{P}(E(X, Y, S))$ represent the probability that an event $E$ happens with respect to $(X, Y, S) \sim \mathcal{D}$ while $\widehat{\mathbb{P}}(E(x, y, s)) = \frac{1}{n} \sum_{i=1}^n \mathbb{I}_{E(x_i, y_i, s_i)}$ is an empirical estimate with respect to $\mathcal{T}$ where $\mathbb{I}_P$ is the indicator function which is 1 when the property $P$ is verified and 0 otherwise. In the remainder of this paper, all our derivations will be considered in the finite sample setting and we will assume that what was measured on our finite sample is sufficiently close to what would be obtained if one had access to the overall distribution. This seems reasonable in light of the previous work on generalization in standard machine learning (Shalev-Shwartz & Ben-David, 2014) and the recent work of Woodworth et al. (2017) or Mangold et al. (2022) which show that the kind of fairness measures we consider in this paper tend to generalize well when the hypothesis space is not too complex, as measured respectively by the VC or the Natarajan Dimension (Shalev-Shwartz & Ben-David, 2014). Since these generalization results only rely on a capacity measure of the hypothesis space and are otherwise algorithm agnostic, they are applicable to the models returned by FairGrad when they have finite VC or Natarajan dimensions. This is for example the case for linear models.

### 2.1 Fairness Definition

We assume that the data may be partitioned into $K$ disjoint groups denoted $\mathcal{T}_1, \ldots, \mathcal{T}_k, \ldots, \mathcal{T}_K$ such that $\bigcup_{k=1}^K \mathcal{T}_k = \mathcal{T}$ and $\bigcap_{k=1}^K \mathcal{T}_k = \emptyset$. These groups highly depend on the fairness notion under consideration. They might correspond to the usual sensitive groups, as is the case for Accuracy Parity (see Example 1), or might be subgroups of the usual sensitive groups, as in Equalized Odds where the subgroups are defined with respect to the true labels (see Example 2 in Appendix B). For each group, we assume that we have access to a function $\widehat{F}_k : \mathcal{D}^n \times \mathcal{H} \to \mathbb{R}$ such that $\widehat{F}_k > 0$ when the group $k$ is advantaged by the given classifier and $\widehat{F}_k < 0$ when the group $k$ is disadvantaged. Furthermore, we assume that the magnitude of $\widehat{F}_k$ represents the degree to which the group is (dis)advantaged. Finally, we assume that each $\widehat{F}_k$ can be rewritten as follows:

$$\widehat{F}_k(\mathcal{T}, h_\theta) = C_k^0 + \sum_{k'=1}^K C_k^{k'} \widehat{\mathbb{P}}(h_\theta(x) \neq y | \mathcal{T}_{k'}) \tag{1}$$

where the constants $C$ are group specific and independent of $h_\theta$. The probabilities $\widehat{\mathbb{P}}(h_\theta(x) \neq y | \mathcal{T}_{k'})$ represent the error rates of $h_\theta$ over each group $\mathcal{T}_{k'}$ with a slight abuse of notation. Below, we show that Accuracy

Parity (Zafar et al., 2017a) respects this definition. In Appendix B, we show that Equality of Opportunity (Hardt et al., 2016), Equalized Odds (Hardt et al., 2016), and Demographic Parity (Calders et al., 2009) also respect this definition. It means that using this generic formulation allows us to simultaneously reason about multiple fairness notions.

**Example 1** (**Accuracy Parity (AP) (Zafar et al., 2017a)**)**.** A model $h_\theta$ is fair for Accuracy Parity when the probability of being correct is independent of the sensitive attribute, that is, $\forall r \in \mathcal{S}$

$$\widehat{\mathbb{P}}\left(h_\theta(x) = y \mid s = r\right) = \widehat{\mathbb{P}}\left(h_\theta(x) = y\right).$$

It means that we need to partition the space into $K = |\mathcal{S}|$ groups and, $\forall r \in \mathcal{S}$, we define $\widehat{F}_{(r)}$ as the fairness level of group $(r)$

$$\widehat{F}_{(r)}(\mathcal{T}, h_\theta) = \widehat{\mathbb{P}}\left(h_\theta(x) \neq y\right) - \widehat{\mathbb{P}}\left(h_\theta(x) \neq y \mid s = r\right)$$
$$= (\widehat{\mathbb{P}}\left(s = r\right) - 1)\widehat{\mathbb{P}}\left(h_\theta(x) \neq y \mid s = r\right) + \sum_{(r') \neq (r)} \widehat{\mathbb{P}}\left(s = r'\right)\widehat{\mathbb{P}}\left(h_\theta(x) \neq y \mid s = r'\right)$$

where the law of total probability was used to obtain the last equality. Thus, Accuracy Parity satisfies all our assumptions with $C_{(r)}^{(r)} = \widehat{\mathbb{P}}\left(s = r\right) - 1$, $C_{(r)}^{(r')} = \widehat{\mathbb{P}}\left(s = r'\right)$ with $r' \neq r$, and $C_{(r)}^0 = 0$.

It is worth noting that FairGrad applies to any fairness measure that respects the definition above, even when there is a large number of groups. However, the performance of FairGrad may degrade when there are only a few samples per group, as fairness estimations become unreliable. In this case, the risk is that the learned model is fair on the training set but does not generalize well to new examples. To circumvent some of these issues works such as Hebert-Johnson et al. (2018); Kearns et al. (2018) have extended fairness definitions to multi-group settings and proposed mechanisms to optimize them. In this work, we focus on classical fairness definitions and keep the line of research to extend Fairgrad to these alternative definitions for the future.

## 2.2 Related Work

Various in-processing methods have been proposed in the fair machine learning literature. Amongst them, many methods rely on formulating the problem as either a constrained optimization, which is later relaxed to an unconstrained case or using re-weighting techniques where examples are dynamically re-weighed based on the fairness levels of the model (Caton & Haas, 2020). In this sub-section, we will provide a brief overview of these methods and explain the similarities and differences between FairGrad and the corresponding approaches. Additionally, we will also demonstrate how FairGrad can be seen as a solution that connects these two streams of work. For more details about very closely related works, please refer to Appendix A.

**Constrained Optimization** The problem of fair machine learning can be seen as the following constrained optimization problem (Cotter et al., 2019; Agarwal et al., 2018):

$$\operatorname*{arg\,min}_{h_\theta \in \mathcal{H}} \widehat{\mathbb{P}}\left(h_\theta(x) \neq y\right)$$
$$\text{s.t. } \forall k \in [K], \widehat{F}_k(\mathcal{T}, h_\theta) = 0. \tag{2}$$

This problem can then be reformulated as an unconstrained optimization problem using Lagrange multipliers. More specifically, with multipliers denoted by $\lambda_1, \ldots, \lambda_K$, the unconstrained objective that should be minimized for $h_\theta \in \mathcal{H}$ and maximized for $\lambda_1, \ldots, \lambda_K \in \mathbb{R}$ is:

$$\mathcal{L}\left(h_\theta, \lambda_1, \ldots, \lambda_K\right) = \widehat{\mathbb{P}}\left(h_\theta(x) \neq y\right) + \sum_{k=1}^{K} \lambda_k \widehat{F}_k(\mathcal{T}, h_\theta). \tag{3}$$

Several strategies may then be employed to find a saddle point for this objective[1]. Agarwal et al. (2018) first relax the problem by searching for a distribution over the models rather than a single optimal hypothesis.

---

[1]These min-max formulations are not new in the literature and was already used in the 1940's (Wald, 1945). More recently, Madry et al. (2018) employed the formulation to make deep neural networks more robust against adversarial attacks. Similarly, Ben-Tal et al. (2012) modeled uncertainty in input via this formulation.

Then, they alternate between using an exponentiated gradient step to find $\lambda_1, \ldots, \lambda_K \in \mathbb{R}$ and a procedure based on cost sensitive learning to find the next $h_\theta$ to add to their distribution. Similarly, Cotter et al. (2019) also search for a distribution over the models using an alternating approach based on Lagrange multipliers where they relax objective (3) by replacing the error rate with a loss term. To update the $\lambda$ multipliers, unlike Agarwal et al. (2018), they use projected gradient descent based on the original fairness terms. To search the next $h_\theta$ to add to their distribution of models they use a projected gradient descent update over a relaxed overall objective function where the fairness measures are replaced with smooth upper bounds.

In this work, we also use an alternating approach based on objective (3). However, we look for a single model rather than a distribution of models. To this end, at each iteration, we update $\lambda$ using a projected gradient descent step similar to Cotter et al. (2019), that is using the original fairness measures. To solve for $h_\theta$, contrary to Cotter et al. (2019), we first show that Objective (3), with fixed $\lambda$, may be rewritten as a weighted sum of group-wise error rates. This is similar in spirit to the cost-sensitive learning method of Agarwal et al. (2018) but can be applied beyond simple binary classification. We then follow Cotter et al. (2019) and replace in our new objective the error rate terms with a loss function, albeit not necessarily an upper bound, to obtain meaningful gradient directions.

**Re-weighting** Another way to learn fair models is to use a re-weighting approach where each example $x$ is associated with a weight $w_x \in \mathbb{R}$ so that minimizing the following objective for $h_\theta$ outputs a fair model:

$$\mathcal{W}(h_\theta) = \widehat{\mathbb{E}}\left(w_x \mathbb{I}_{\{h_\theta(x) \neq y\}}\right).$$

The underlying idea for the methods which posit the problem as above is to propose a cost function that outputs weights for each example. On the one hand, the weights can be determined in a pre-processing step (Kamiran & Calders, 2012), based on the statistics of the data under consideration. On the other hand, the weights may evolve with $h_\theta$, that is they are dynamically updated each time the model changes during the training process (Roh et al., 2020).

In this work, to find $h_\theta$, we also use a dynamic re-weighting approach where the weights change at each iteration. To choose the weights, we initially give the same importance to each example. Then, we increase the weights of disadvantaged examples and decrease the weights of advantaged examples proportionally to the fairness level of the current model for their group. An important feature of our approach, unlike other re-weighting approaches, is that we do not constrain ourselves to positive weights but rather allow the use of negative weights. Indeed, we show in Lemma 1 that the latter are sometimes necessary to learn fair models.

To summarize, we first frame the task as a constrained optimization problem, similar to Cotter et al. (2019) and Agarwal et al. (2018). We then propose an alternating approach, where we update $\lambda$ at each iteration using a projected gradient descent step similar to Cotter et al. (2019). However, in order to learn the model $h_\theta$, we show that Objective (3), with fixed $\lambda$, can be rewritten as a weighted sum of group-wise error rates. This step can be interpreted as an instance of dynamic re-weighting where the weights change at each iteration. Thus our method can be seen as a connection between constrained optimization and re-weighting.

## 3 FairGrad

In the previous section, we argued that FairGrad is connected to both constrained optimization and re-weighting approaches. In this section, we provide details on our method and we present it starting from the constrained optimization point of view as we believe it makes it easier to understand how the weights are selected and updated. We begin by discussing FairGrad for exact fairness and then extend it to $\epsilon$-fairness.

### 3.1 FairGrad for Exact Fairness

To solve the fairness problem described in equation 3, we propose to use an alternating approach where the hypothesis and the multipliers are updated one after the other[2]. We begin by describing our method to update the multipliers and then the model.

---

[2]It is worth noting that, here, we do not have formal duality guarantees and that the problem is not even guaranteed to have a fair solution. Nevertheless, the approach seems to work well in practice as can be seen in the experiments.

**Updating the Multipliers.** To update $\lambda_1, \ldots, \lambda_K$, we will use a standard gradient ascent procedure. Hence, given that the gradient of Problem (3) is

$$\nabla_{\lambda_1, \ldots, \lambda_K} \mathcal{L}(h_\theta, \lambda_1, \ldots, \lambda_K) = \begin{pmatrix} \widehat{F}_1(\mathcal{T}, h_\theta) \\ \vdots \\ \widehat{F}_K(\mathcal{T}, h_\theta) \end{pmatrix}$$

we have the following update rule $\forall k \in [K]$:

$$\lambda_k^{T+1} = \lambda_k^T + \eta_\lambda \widehat{F}_k(\mathcal{T}, h_\theta^T)$$

where $\eta_\lambda$ is a rate that controls the importance of each update. In the experiments, we use a constant rate of 0.01 as our initial tests showed that it is a good rule of thumb when the data is properly standardized.

**Updating the Model.** To update the parameters $\theta \in \mathbb{R}^D$ of the model $h_\theta$, we use a standard gradient descent. However, first, we notice that, given our fairness definition, Equation (3) can be written as

$$\mathcal{L}(h_\theta, \lambda_1, \ldots, \lambda_K) = \sum_{k=1}^{K} \widehat{\mathbb{P}}(h_\theta(x) \neq y | \mathcal{T}_k) \left[ \widehat{\mathbb{P}}(\mathcal{T}_k) + \sum_{k'=1}^{K} C_{k'}^k \lambda_{k'} \right] + \sum_{k=1}^{K} \lambda_k C_k^0. \tag{4}$$

where $\sum_{k=1}^{K} \lambda_k C_k^0$ is independent of $h_\theta$ by definition. Hence, at iteration $t$, the update rule becomes

$$\theta^{T+1} = \theta^T - \eta_\theta \sum_{k=1}^{K} \left[ \widehat{\mathbb{P}}(\mathcal{T}_k) + \sum_{k'=1}^{K} C_{k'}^k \lambda_{k'} \right] \nabla_\theta \widehat{\mathbb{P}}(h_\theta(x) \neq y | \mathcal{T}_k)$$

where $\eta_\theta$ is the usual learning rate that controls the importance of each parameter update. Here, we obtain our group specific weights $\forall_k, w_k = \left[ \widehat{\mathbb{P}}(\mathcal{T}_k) + \sum_{k'=1}^{K} C_{k'}^k \lambda_{k'} \right]$, that depend on the current fairness level of the model through $\lambda_1, \ldots, \lambda_K$, the relative size of each group through $\widehat{\mathbb{P}}(\mathcal{T}_k)$, and the fairness notion under consideration through the constants $C$. The exact values of these constants are given in Section 2.1 and Appendix B for various group fairness notions. Overall, they are such that, at each iteration, the weights of the advantaged groups are reduced and the weights of the disadvantaged groups are increased.

The main limitation of the above update rule is that one needs to compute the gradient of $0-1$-losses since $\nabla_\theta \widehat{\mathbb{P}}(h_\theta(x) \neq y | \mathcal{T}_k) = \frac{1}{n_k} \sum_{(x,y) \in \mathcal{T}_k} \nabla_\theta \mathbb{I}_{\{h_\theta(x) \neq y\}}$. Unfortunately, this usually does not provide meaningful optimization directions. To address this issue, we follow the usual trend in machine learning and replace the $0-1$-loss with one of its continuous and differentiable surrogates that provides meaningful gradients. For instance, in our experiments, we use the cross entropy loss.

### 3.2 Computational Overhead of FairGrad.

We summarize our approach in Algorithm 1, where we have used italic font to highlight the steps inherent to FairGrad that do not appear in classic gradient descent. We consider batch gradient descent rather than full gradient descent as it is a popular scheme. We empirically investigate the impact of the batch size in Section 4.7. The main difference is Step 5 (in italic font), that is the computation of the group-wise fairness levels. However, these can be cheaply obtained from the predictions of $h_\theta^{(t)}$ on the current batch which are always available since they are also needed to compute the gradient. Hence, the computational overhead of FairGrad is very limited.

### 3.3 Importance of Negative Weights.

A key property of FairGrad is that we allow the use of negative weights, that is $\left[ \widehat{\mathbb{P}}(\mathcal{T}_k) + \sum_{k'=1}^{K} C_{k'}^k \lambda_{k'} \right]$ may become negative, while existing methods (Roh et al., 2020; Iosifidis & Ntoutsi, 2019; Jiang & Nachum, 2020) restrict themselves to positive weights. In this section, we show that these negative weights are important as they are sometimes necessary to learn fair models. Hence, in the next lemma, we provide sufficient conditions so that negative weights are mandatory if one wants to enforce Accuracy Parity.

---

**Algorithm 1** FairGrad for Exact Fairness

---

**Input**: Groups $\mathcal{T}_1, \ldots, \mathcal{T}_K$, Functions $\widehat{F}_1, \ldots, \widehat{F}_K$, Function class $\mathcal{H}$ of models $h_\theta$ with parameters $\theta \in \mathbb{R}^D$, Learning rates $\eta_\lambda, \eta_\theta$, and Iterator *iter* that returns batches of examples.

**Output**: A fair model $h_\theta^*$.

1: Initialize *the group specific weights* and the model.
2: **for** B in *iter* **do**
3:     Compute the predictions of the current model on the batch B.
4:     Compute the group-wise losses using the predictions.
5:     *Compute the current fairness level using the predictions and update the group-wise weights.*
6:     Compute the overall *weighted* loss using the *group-wise weights*.
7:     Compute the gradients based on the loss and update the model.
8: **end for**
9: **return** the trained model $h_\theta^*$

---

**Lemma 1** (Negative weights are necessary.)**.** *Let the fairness notion be Accuracy Parity (Example 1). Let $h_\theta^*$ be the most accurate and fair model. Then using negative weights is necessary as long as*

$$\min_{\substack{h_\theta \in \mathcal{H} \\ h_\theta\, unfair}} \max_{\mathcal{T}_k} \widehat{\mathbb{P}}\left(h_\theta(x) \neq y | \mathcal{T}_k\right) < \widehat{\mathbb{P}}\left(h_\theta^*(x) \neq y\right).$$

*Proof.* The proof is provided in Appendix C. $\square$

The previous condition can sometimes be verified in practice. As a motivating example, assume a binary setting with only two sensitive groups $\mathcal{T}_1$ and $\mathcal{T}_{-1}$. Let $h_\theta^{-1}$ be the model minimizing $\widehat{\mathbb{P}}\left(h_\theta(x) \neq y | \mathcal{T}_{-1}\right)$ and assume that $\widehat{\mathbb{P}}\left(h_\theta^{-1}(x) \neq y\right) < \widehat{\mathbb{P}}\left(h_\theta^{-1}(x) \neq y | \mathcal{T}_{-1}\right)$, that is group $\mathcal{T}_{-1}$ is disadvantaged for accuracy parity. Given $h_\theta^*$ the most accurate and fair model, we have

$$\min_{\substack{h_\theta \in \mathcal{H} \\ h_\theta\, unfair}} \max_{\mathcal{T}_k} \widehat{\mathbb{P}}\left(h_\theta(x) \neq y | \mathcal{T}_k\right) = \widehat{\mathbb{P}}\left(h_\theta^{-1}(x) \neq y | \mathcal{T}_{-1}\right) < \widehat{\mathbb{P}}\left(h_\theta^*(x) \neq y\right)$$

as otherwise we would have a contradiction since the fair model would also be the most accurate model for group $\mathcal{T}_{-1}$ since $\widehat{\mathbb{P}}\left(h_\theta^*(x) \neq y\right) = \widehat{\mathbb{P}}\left(h_\theta^*(x) \neq y | \mathcal{T}_{-1}\right)$ by definition of Accuracy Parity. In other words, a dataset where the most accurate model for a given group still disadvantages it requires negative weights. This might be connected to the notion of leveling down (Zietlow et al., 2022; Mittelstadt et al., 2023), where fairness can only be achieved by harming all the groups or bringing advantaged groups closer to disadvantaged groups by harming them. It is generally an artifact of strictly egalitarian fairness measures. Investigating this negative effect is an important research direction that goes beyond the scope of this paper. Nevertheless, a potential solution to mitigate it is to use other kind of fairness definitions. As a first step in this direction, in the next section we extend FairGrad to $\epsilon$-fairness where strict equality is relaxed.

### 3.4 FairGrad for $\epsilon$-fairness

In the previous section, we considered exact fairness and we showed that this could be achieved by using a re-weighting approach. Here, we extend this procedure to $\epsilon$-fairness where the fairness constraints are relaxed and a controlled amount of violations is allowed. Usually, $\epsilon$ is a user defined parameter but it can also be set by the law, as it is the case with the 80% rule in the US (Biddle, 2006). The main difference with exact fairness is that each equality constraint in Problem (2) is replaced with two inequalities of the form

$$\forall k \in [K], \widehat{F}_k(\mathcal{T}, h_\theta) \leq \epsilon$$
$$\forall k \in [K], \widehat{F}_k(\mathcal{T}, h_\theta) \geq -\epsilon.$$

The main consequence is that we need to maintain twice as many Lagrange multipliers and that the group-wise weights are slightly different. Since the two procedures are similar, we omit the details here but provide them in Appendix D for the sake of completeness.

# 4 Experiments

In this section, we present several experiments that demonstrate the competitiveness of FairGrad as a procedure to learn fair models for classification. We begin by presenting results over standard fairness datasets and a Natural language Processing dataset in Section 4.4. We then study the behaviour of the $\epsilon$-fairness variant of FairGrad in Section 4.5. Next, we showcase the fine-tuning ability of FairGrad on a Computer Vision dataset in Section 4.6. Finally, we investigate the impact of batch size on the learned model in Section 4.7 and present results related to the computational overhead incurred by FairGrad in Section 4.8.

## 4.1 Datasets

In the main paper, we consider 4 different datasets and postpone the results on another 6 datasets to Appendix E.3 as they follow similar trends. We also postpone the detailed descriptions of these datasets as well as the pre-processing steps to Appendix E.2.

We consider commonly used fairness datasets, namely **Adult Income** (Kohavi, 1996) and **CelebA** (Liu et al., 2015). Both are binary classification datasets with binary sensitive attributes (gender). We also consider a variant of the Adult Income dataset where we add a second binary sensitive attribute (race) to obtain a dataset with 4 disjoint sensitive groups. For both datasets, we use 20% of the data as a test set and the remaining 80% as a train set. We further divide the train set into two and keep 25% of the training examples as a validation set. For each repetition, we randomly shuffle the data before splitting it, and thus we have unique splits for each random seed. Lastly, we standardize each features independently by subtracting the mean and scaling to unit variance which were estimated on the training set.

To showcase the wide applicability of FairGrad, we consider the **Twitter Sentiment**[3] (Blodgett et al., 2016) dataset from the Natural Language Processing community. It consists of $200k$ tweets with binary sensitive attribute (race) and binary sentiment score. We employ the same setup, splits, and the pre-processing as proposed by Han et al. (2021) and Elazar & Goldberg (2018) and create bias in the dataset by changing the proportion of each subgroup (race-sentiment) in the training set. Following the footsteps of Elazar & Goldberg (2018) we encode the tweets using the DeepMoji (Felbo et al., 2017) encoder with no fine-tuning, which has been pre-trained over millions of tweets to predict their emoji, thereby predicting the sentiment. We also employ the **UTKFace** dataset[4] (Zhang et al., 2017) from the Computer Vision community. It consists of $23,708$ images tagged with race, age, and gender with pre-defined splits.

## 4.2 Performance Measures

For fairness, we consider the four measures introduced in Section 2.1 and Appendix B, namely Equalized Odds (EOdds), Equality of Opportunity (EOpp), Accuracy Parity (AP), and Demographic Parity (DP). For each specific fairness notion, we report the average absolute fairness level of the different groups over the test set, that is $\frac{1}{K}\sum_{k=1}^{K}\left|\widehat{F}_k(\mathcal{T}, h_\theta)\right|$ (lower is better). To assess the utility of the learned models, we use their accuracy levels over the test set, that is $\frac{1}{n}\sum_{i=1}^{n}\mathbb{I}_{h_\theta(x_i)=y_i}$ (higher is better). All the results reported are averaged over 5 independent runs and standard deviations are provided. Note that, in the main paper, we graphically report a subset of the results over the aforementioned datasets. We provide detailed results in Appendix E.3, including the missing pictures as well as complete tables with accuracy levels, fairness levels, and fairness level of the most well-off and worst-off groups for all the relevant methods.

## 4.3 Methods

We compare FairGrad to a wide variety of baselines, namely:

- **Unconstrained**, which is oblivious to any fairness measure and is trained using a standard batch gradient descent method.

---

[3]http://slanglab.cs.umass.edu/TwitterAAE/
[4]https://susanqq.github.io/UTKFace/

- **Adversarial** learning based method where we employ adversarial mechanism (Goodfellow et al., 2014) using a gradient reversal layer (Ganin & Lempitsky, 2015), similar to GRAD-Pred (Raff & Sylvester, 2018), where an adversary, with an objective to predict the sensitive attribute, is added to the unconstrained model

- Bi-level optimization based method implemented in the form of **BiFair** (Ozdayi et al., 2021)

- Re-weighting based methods in the form of **FairBatch** (Roh et al., 2020). We also compare against a simpler baseline called **Weighted ERM** where each example is reweighed based on the size of the sensitive group the example belongs to in the beginning. Unlike FairBatch these weights are not updated during training.

- Constrained optimization based method as proposed by Cotter et al. (2019). We refer to this method as **Constraints** in this article.

- **Reduction** implements the exponentiated gradient based fair classification approach as proposed by Agarwal et al. (2018).

In all our experiments, we consider two different hypothesis classes. On the one hand, we use linear models implemented in the form of neural networks with no hidden layers. On the other hand, we use a more complex, non-linear architecture with three fully-connected hidden layers of respective sizes 128, 64, and 32. We use ReLU as our activation function with batch normalization and dropout. In both cases, we optimize the cross-entropy loss.

In several experiments, we only consider subsets of the baselines due to the limitations of the methods. For instance, BiFair was designed to handle binary labels and binary sensitive attributes and thus is not considered for the datasets with more than two sensitive groups or two labels. Furthermore, we implemented it using the authors code that is freely available online but does not include AP as a fairness measure, thus we do not report results related to this measure for BiFair. Similarly, we also implemented FairBatch from the authors code which does not support AP as a fairness measure, thus we also exclude it from the comparison for this measure. For Constraints, we based our implementation on the publicly available authors library but were only able to reliably handle linear models and thus we do not consider this baseline for non-linear models. Finally, for Adversarial, we used our custom made implementation. However, it is only applicable when learning non-linear models since it requires at least one hidden layer to propagate its reversed gradient.

Apart from the common hyper-parameters such as dropout, several baselines come with their own set of hyper-parameters. For instance, BiFair has the *inner loop length*, which controls the number of iterations in its inner loop, while Adversarial has the *scaling*, which re-weights the adversarial branch loss and the task loss. We provide details of common and approach specific hyper-parameters with their range in Appendix E.1.

With several hyper-parameters for each approach, selecting the best combination is often crucial to avoid undesirable behaviors such as over-fitting (Maheshwari et al., 2022). In this paper, we opt for the following procedure. First, for each method, we consider all the $X$ possible hyper-parameter combinations and we run the training procedure for 50 epochs for each combination. Then, we retain all the models returned by the last 5 epochs, that is, for a given method, we have $5X$ models and the goal is to select the best one among them. Since we have access to two performance measures, we can select either the most accurate model, the most fair, or a trade-off between the two depending on the end goal. Here, we chose to focus on the third option and select the model with the lowest fairness score between certain accuracy intervals. More specifically, let $\alpha^*$ be the highest validation accuracy among the $5X$ models. We choose the model with the lowest validation fairness score amongst all models with a validation accuracy in the interval $[\alpha^* - k, \alpha^*]$. In this work, we fix k to 0.03.

### 4.4 Results for Exact Fairness

We report the results over the Adult Income dataset using a linear model, the Adult Income dataset with multiple groups with a non-linear model, and the Twitter sentiment dataset using both linear and nonlinear models in Figures 2, 3, and 4 respectively. In these figures, the best methods are closer to the bottom right

corner. If a method is closer to the bottom left corner, it has good fairness but reduced accuracy. Similarly, a method closer to the top right corner has good accuracy but poor fairness.

The main take-away from these experiments is that there is no fairness enforcing method that is consistently better than the others in terms of both accuracy and fairness. All of them have strengths, that is datasets and fairness measures where they obtain good results, and weaknesses, that is datasets and fairness measures for which they are sub-optimal. FairBatch induces better accuracy than the other approaches over Adult with linear model and EOdds and only pays a small price in fairness. However, it is significantly worse in terms of fairness over the Adult Multigroup dataset with a non-linear model. Similarly, BiFair is sub-optimal on Adult with EOpp, while being comparable to the other approaches on the Twitter Sentiment dataset. We observed similar trends on the other datasets, available in Appendix E.3, with different methods coming out on top for different datasets and fairness measures.

Interestingly, FairGrad generally outperforms other approaches in terms of fairness, albeit with a slight loss in accuracy. These observations are even more amplified in the Accuracy Parity and Equalized Odds settings. Moreover, it is generally more robust and tends to show a lower standard deviation in accuracy and fairness than the other approaches. Even in terms of accuracy, the largest difference is over the Crime dataset, where the difference between FairGrad and Unconstrained is 0.04. However, in most cases, the difference is within 0.02. In terms of the multi-group setup, we find similar observations, that is FairGrad outperforms other approaches in fairness, albeit with a drop in accuracy. In fact, for Equality of Opportunity FairGrad almost outperforms all approaches in terms of fairness and accuracy. Overall, FairGrad performs reasonably well in all the settings we considered with no obvious weaknesses, that is no datasets with the lowest accuracy and fairness compared to the baselines.

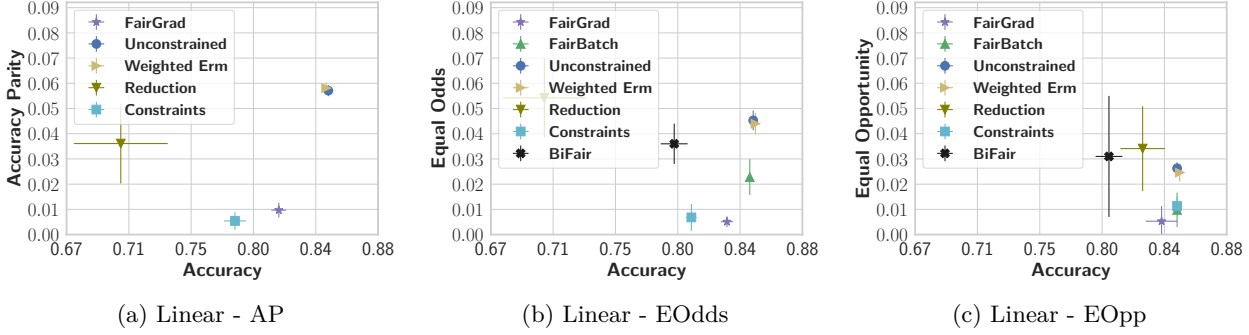

(a) Linear - AP        (b) Linear - EOdds        (c) Linear - EOpp

Figure 2: Results for the Adult dataset using Linear Models.

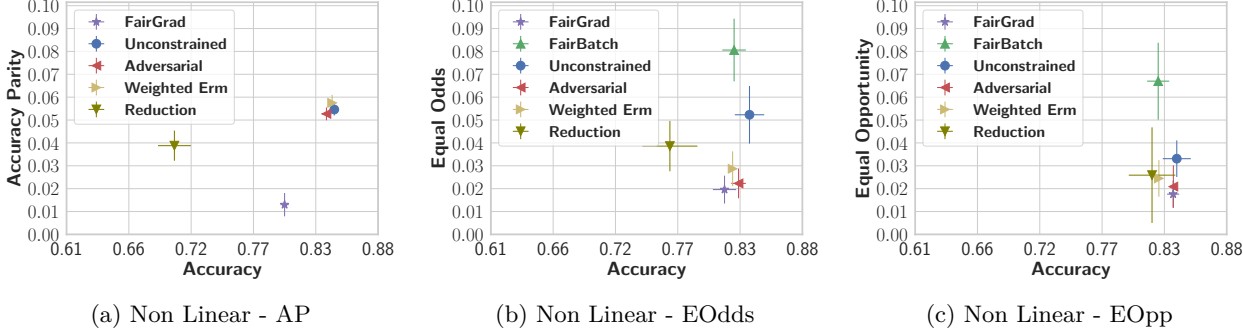

(a) Non Linear - AP        (b) Non Linear - EOdds        (c) Non Linear - EOpp

Figure 3: Results for the Adult Multigroup dataset using Non Linear models.

## 4.5 Accuracy Fairness Trade-off

In this second set of experiments, we demonstrate the capability of FairGrad to support approximate fairness (see Section 3.4). In Figure 5, we show the performance, as accuracy-fairness pairs, of several models learned

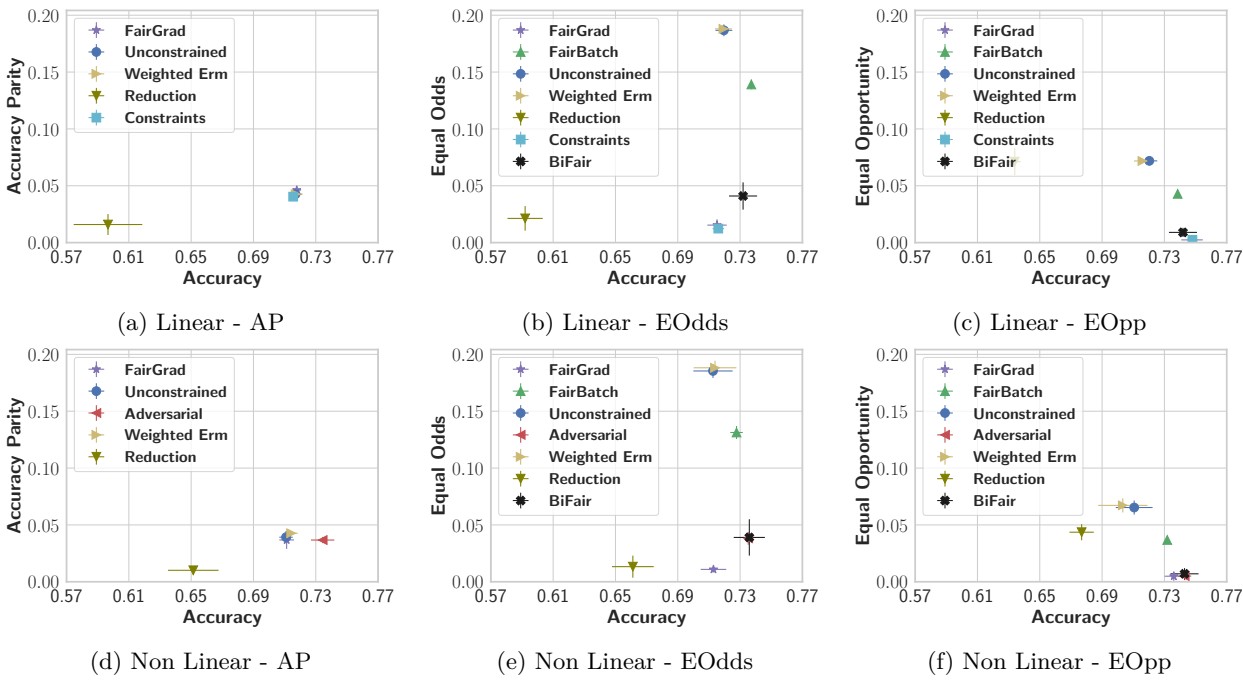

Figure 4: Results for the Twitter Sentiment dataset for Linear and Non Linear Models.

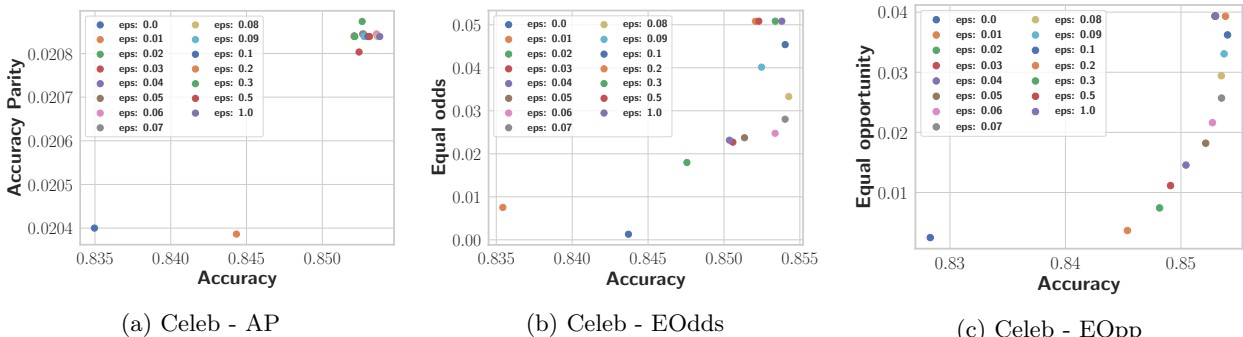

Figure 5: Results for CelebA using Linear models. The Unconstrained Linear model achieves a test accuracy of 0.8532 with fairness level of 0.0499 for EOdds, 0.0204 for AP, and 0.0387 for EOpp.

on the CelebA dataset by varying the fairness level parameter $\epsilon$. These results suggest that FairGrad respects the constraints well. Indeed, the average absolute fairness level (across all the groups, see Section 4.2) achieved by FairGrad is either the same or less than the given threshold. It is worth mentioning that FairGrad is designed to enforce $\epsilon$-fairness for each constraint individually which is slightly different from the summarized quantity displayed here. Finally, as the fairness constraint is relaxed, the accuracy of the model increases, reaching the same performance as Unconstrained when the fairness level of the latter is below $\epsilon$.

## 4.6 FairGrad as a Fine-Tuning Procedure

While FairGrad has primarily been designed to learn fair classifiers from scratch, it can also be used to fine-tune an existing classifier to achieve better fairness. To showcase this, we fine-tune the ResNet18 (He et al., 2016) model, developed for image recognition, over the UTKFace dataset (Zhang et al., 2017), consisting of human face images tagged with Gender, Age, and Race information. Following the same process as Roh et al. (2020), we use Race as the sensitive attribute and consider two scenarios. Either we consider Demographic Parity as the fairness measure and use the gender (binary) as the target label or we consider Equalized Odds

Table 1: Results for the UTKFace dataset where a ResNet18 is fine-tuned using different strategies.

| Method | s=Race ; y=Gender | | s=Race ; y=Age | |
|---|---|---|---|---|
| | Accuracy | DP | Accuracy | EOdds |
| Unconstrained | $0.8691 \pm 0.0075$ | $0.0448 \pm 0.0066$ | $0.6874 \pm 0.0080$ | $0.0843 \pm 0.0089$ |
| FairGrad | $0.8397 \pm 0.0085$ | $0.0111 \pm 0.0064$ | $0.6491 \pm 0.0082$ | $0.0506 \pm 0.0059$ |

Table 2: Batch size effect on the CelebA dataset with Linear Models and EOdds as the fairness measure.

| Batch Size | 8 | 16 | 32 | 64 | 128 | 256 | 512 | 1024 | 2048 |
|---|---|---|---|---|---|---|---|---|---|
| Accuracy | 0.8186 | 0.8234 | 0.8215 | 0.8268 | 0.8273 | 0.8286 | 0.8292 | 0.8289 | 0.8303 |
| Accuracy Std | 0.0013 | 0.006 | 0.0028 | 0.0025 | 0.0031 | 0.0008 | 0.0027 | 0.0017 | 0.0031 |
| Fairness | 0.0031 | 0.0091 | 0.0045 | 0.0036 | 0.0051 | 0.0046 | 0.004 | 0.0038 | 0.0057 |
| Fairness Std | 0.0042 | 0.0062 | 0.0012 | 0.0014 | 0.0025 | 0.0032 | 0.0026 | 0.0019 | 0.0018 |

and predict the age (multi-valued). The results are displayed in Table 1. In both settings, FairGrad learns models that are more fair than an Unconstrained fine-tuning procedure, albeit at the expense of accuracy.

### 4.7 Impact of the Batch-size

In this section, we evaluate the impact of batch size on the fairness and accuracy level of the learned model. Indeed, at each iteration, in order to minimize the overhead associated with FairGrad (see Section 3.1), we update the weights using the fairness level of the model estimated solely on the current batch. When these batches are small, these estimates are unreliable and might lead the model astray. In Table 2 we present the performances of several linear models learned with different batch sizes on the CelebA dataset. Over this dataset, we observe that FairGrad consistently learns a fair model across all batch sizes and obtains reasonable accuracy since Unconstrained has an accuracy of 0.8532 for this problem. Nevertheless, we still recommend the practitioners to use a larger batch size whenever possible as we observe a slight reduction in terms of fairness standard deviations.

### 4.8 Computational Overhead

In this last experiment, we evaluate the overhead of FairGrad, by reporting the wall clock time in seconds to train for an epoch with the Unconstrained approach and our method in various settings.

- We show the effect of model size by varying the number of hidden layers of the model over the Adult Income dataset, which consists of $45,222$ records. We used an Intel Xeon E5-2680 CPU to train.

- We consider a large convolutional neural network (ResNet18 (He et al., 2016)) fine tuned over the UTK-Face dataset consisting of $23,708$ images. We trained the model using a Tesla P100 GPU.

- We experiment with a large transformer (bert-base-uncased (Devlin et al., 2019)) fine tuned over the Twitter Sentiment Dataset consisting of $200k$ tweets. We trained it using a Tesla P100 GPU.

We present results of the computation overhead of FairGrad in Table 3. We find that the overhead is limited and should not be critical in most applications as it does not depend on the complexity of the model but, instead, on the number of examples and the batch size. Overall, these observations are in line with the arguments presented in Section 3.2.

## 5 Conclusion

In this paper, we proposed FairGrad, a fairness aware gradient descent approach based on a re-weighting scheme. We showed that it can be used to learn fair models for various group fairness definitions and is able

Table 3: The computational overhead of FairGrad in various settings. BS here refers to Batch Size, and the Unconstrained and FairGrad columns refers to the average time in seconds taken by these approaches for an epoch, respectively. Delta refers to the difference in time between these two approaches.

| Setting | Parameters | BS | Unconstrained | FairGrad | Delta |
|---|---|---|---|---|---|
| Linear model - Adult Dataset -CPU | 106 | 512 | $0.277 \pm 0.031$ | $0.307 \pm 0.01$ | 0.03 |
| 2 layers -Adult Dataset -CPU | 1762 | 512 | $0.315 \pm 0.036$ | $0.316 \pm 0.029$ | 0.01 |
| 5 layers -Adult Dataset -CPU | 21346 | 512 | $0.370 \pm 0.042$ | $0.394 \pm 0.025$ | 0.02 |
| 10 layers -Adult Dataset -CPU | 39042 | 512 | $0.483 \pm 0.021$ | $0.499 \pm 0.034$ | 0.02 |
| 20 layers -Adult Dataset -CPU | 80642 | 512 | $0.672 \pm 0.034$ | $0.689 \pm 0.026$ | 0.02 |
| ResNet18 trained -UTKFace -GPU | 11177538 | 64 | $31.173 \pm 0.085$ | $31.588 \pm 0.055$ | 0.42 |
| Bert Twitter Sentiment -GPU | 109505310 | 32 | $2246.342 \pm 3.20$ | $2294.382 \pm 4.01$ | 48.04 |

to handle multiclass problems as well as settings where there is multiple sensitive groups. We empirically showed the competitiveness of our approach against several baselines on standard fairness datasets and on a Natural Language Processing task. We also showed that it can be used to fine-tune an existing model on a Computer Vision task. Finally, since it is based on gradient descent and has a small overhead, we believe that FairGrad could be used for a wide range of applications, even beyond classification.

**Limitations and Societal Impact**

While appealing, FairGrad also has limitations. It implicitly assumes that a set of weights that would lead to a fair model exists but this might be difficult to verify in practice. Thus, even if in our experiments FairGrad seems to behave quite well, a practitioner using this approach should not trust it blindly. It remains important to always check the actual fairness level of the learned model. On the other hand, we believe that, due to its simplicity and its versatility, FairGrad could be easily deployed in various practical contexts and, thus, could contribute to the dissemination of fair models.

**Acknowledgements**

This work has been supported by the Région Hauts de France (Projet STaRS Equité en apprentissage décentralisé respectueux de la vie privée) and Agence Nationale de la Recherche under grant number ANR-19-CE23-0022. The authors would also like to thank Michael Lohaus and anonymous reviewers for helpful discussions and feedbacks.

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

## Appendix

In this appendix, we provide details that were omitted in the main paper. First, in Section A, we review several works closely related to ours. Then, in Section B, we show that several well known group fairness measures are compatible with FairGrad. In Section C, we prove Lemma 1. Next, in Section D, we derive the update rules for FairGrad with $\epsilon$-fairness. Finally, in Section E, we provide additional experiments.

## A   Related Work

The fairness literature is extensive and we refer the interested reader to recent surveys (Caton & Haas, 2020; Mehrabi et al., 2021) to get an overview of the subject. Here, we focus on recent works that are more closely related to our approach.

**BiFair (Ozdayi et al., 2021).**   This paper proposes a bilevel optimization scheme for fairness. The idea is to use an outer optimization scheme that learns weights for each example so that the trade-off between fairness and accuracy is as favorable as possible while an inner optimization scheme learns a model that is as accurate as possible. One limitation of this approach is that it does not directly optimize the fairness level of the model but rather a relaxation that does not provide any guarantees on the goodness of the learned predictor. Furthermore, it is limited to binary classification with a binary sensitive attribute. In this paper, we also learn weights for the examples in an iterative way. However, we use a different update rule. Furthermore, we focus on exact fairness definitions rather than relaxations and our objective is to learn accurate models with given levels of fairness rather than a trade-off between the two. Finally, our approach is not limited to the binary setting.

**FairBatch (Roh et al., 2020).**   This paper proposes a batch gradient descent approach to learn fair models. More precisely, the idea is to draw a batch of examples from a skewed distribution that favors the disadvantaged groups by oversampling them. In this paper, we propose to use a re-weighting approach which could also be interpreted as altering the distribution of the examples based on their fairness level if all the weights were positive. However, we allow the use of negative weights, and we prove that they are sometimes necessary to achieve fairness. Furthermore, we employ a different update rule for the weights.

**AdaFair (Iosifidis & Ntoutsi, 2019).** This paper proposes a boosting based framework to learn fair models. The underlying idea is to modify the weights of the examples depending on both the performances of the current strong classifier and the group memberships. Hence, examples that belong to the disadvantaged group and are incorrectly classified receive higher weights than the examples that belong to the advantaged group and are correctly classified. In this paper, we use a similar high level idea but we use different weights that do not depend on the accuracy of the model but solely on its fairness. Furthermore, rather than a boosting based approach, we consider problems that can be solved using gradient descent. Finally, while AdaFair only focuses on Equalized Odds, we show that our approach works with several fairness notions.

**Identifying and Correcting Label Bias in Machine Learning (Jiang & Nachum, 2020).** This paper tackles the fairness problem by assuming that the observed labels are biased compared to the true labels. The goal is then to learn a model with respect to the true labels using only the observed labels. To this end, it proposes to use an iterative re-weighting procedure where positive example-wise weights and the model are alternatively updated. In this paper, we also propose a re-weighting approach. However, we use different weights that are not necessarily positive. Furthermore, our approach is not limited to binary labels and can handle multiclass problems.

## B    Reformulation of Various Group Fairness Notion

In this section, we present several group fairness notions which respect our fairness definition presented in Section 2.1.

**Example 2** (**Equalized Odds (EOdds) (Hardt et al., 2016)**). A model $h_\theta$ is fair for Equalized Odds when the probability of predicting the correct label is independent of the sensitive attribute, that is, $\forall l \in \mathcal{Y}, \forall r \in \mathcal{S}$

$$\widehat{\mathbb{P}}\left(h_\theta(x) = l \,|\, s = r, y = l\right) = \widehat{\mathbb{P}}\left(h_\theta(x) = l \,|\, y = l\right).$$

It means that we need to partition the space into $K = |\mathcal{Y} \times \mathcal{S}|$ groups and, $\forall l \in \mathcal{Y}, \forall r \in \mathcal{S}$, we define $\widehat{F}_{(l,r)}$ as

$$
\begin{aligned}
\widehat{F}_{(l,r)}(\mathcal{T}, h_\theta) &= \widehat{\mathbb{P}}\left(h_\theta(x) \neq l \,|\, y = l\right) - \widehat{\mathbb{P}}\left(h_\theta(x) \neq l \,|\, s = r, y = l\right) \\
&= \sum_{(l,r') \neq (l,r)} \widehat{\mathbb{P}}\left(s = r'|y = l\right) \widehat{\mathbb{P}}\left(h_\theta(x) \neq l \,|\, s = r', y = l\right) \\
&\quad - (1 - \widehat{\mathbb{P}}\left(s = r|y = l\right)) \widehat{\mathbb{P}}\left(h_\theta(x) \neq l \,|\, s = r, y = l\right)
\end{aligned}
$$

where the law of total probability was used to obtain the last equation. Thus, Equalized Odds satisfies all our assumptions with $C_{(l,r)}^{(l,r)} = \widehat{\mathbb{P}}\left(s = r|y = l\right) - 1$, $C_{(l,r)}^{(l,r')} = \widehat{\mathbb{P}}\left(s = r'|y = l\right)$, $C_{(l,r)}^{(l',r')} = 0$ with $r' \neq r$ and $l' \neq l$, and $C_{(l,r)}^0 = 0$.

**Example 3** (**Equality of Opportunity (EOpp) (Hardt et al., 2016)**). A model $h_\theta$ is fair for Equality of Opportunity when the probability of predicting the correct label is independent of the sensitive attribute for a given subset $\mathcal{Y}' \subset \mathcal{Y}$ of labels called the desirable outcomes, that is, $\forall l \in \mathcal{Y}', \forall r \in \mathcal{S}$

$$\widehat{\mathbb{P}}\left(h_\theta(x) = l \,|\, s = r, y = l\right) = \widehat{\mathbb{P}}\left(h_\theta(x) = l \,|\, y = l\right).$$

It means that we need to partition the space into $K = |\mathcal{Y} \times \mathcal{S}|$ groups and, $\forall l \in \mathcal{Y}, \forall r \in \mathcal{S}$, we define $\widehat{F}_{(l,r)}$ as

$$
\widehat{F}_{(l,r)}(\mathcal{T}, h_\theta) =
\begin{cases}
\widehat{\mathbb{P}}\left(h_\theta(x) = l \,|\, s = r, y = l\right) \\
\quad - \widehat{\mathbb{P}}\left(h_\theta(x) = l \,|\, y = l\right) & \forall (l,r) \in \mathcal{Y}' \times \mathcal{S} \\
0 & \forall (l,r) \in \mathcal{Y} \times \mathcal{S} \setminus \mathcal{Y}' \times \mathcal{S}
\end{cases}
$$

which can then be rewritten in the correct form in the same way as Equalized Odds, the only difference being that $C_{(l,r)}^\cdot = 0, \forall (l,r) \in \mathcal{Y} \times \mathcal{S} \setminus \mathcal{Y}' \times \mathcal{S}$.

**Example 4** (**Demographic Parity (DP) (Calders et al., 2009)**). A model $h_\theta$ is fair for Demographic Parity when the probability of predicting a binary label is independent of the sensitive attribute, that is, $\forall l \in \mathcal{Y}, \forall r \in \mathcal{S}$

$$\widehat{\mathbb{P}}\left(h_\theta(x) = l \,|\, s = r\right) = \widehat{\mathbb{P}}\left(h_\theta(x) = l\right).$$

It means that we need to partition the space into $K = |\mathcal{Y} \times \mathcal{S}|$ groups and, $\forall l \in \mathcal{Y}, \forall r \in \mathcal{S}$, we define $\widehat{F}_{(l,r)}$ as

$$
\begin{aligned}
\widehat{F}_{(l,r)}(\mathcal{T}, h_\theta) &= \widehat{\mathbb{P}}\left(h_\theta(x) \neq l\right) - \widehat{\mathbb{P}}\left(h_\theta(x) \neq l \,|\, s = r\right) \\
&= \left(\widehat{\mathbb{P}}\left(y = l, s = r\right) - \widehat{\mathbb{P}}\left(y = l \,|\, s = r\right)\right) \widehat{\mathbb{P}}\left(h_\theta(x) \neq y \,|\, s = r, y = l\right) \\
&\quad + \sum_{(l,r') \neq (l,r)} \widehat{\mathbb{P}}\left(y = l, s = r'\right) \widehat{\mathbb{P}}\left(h_\theta(x) \neq y \,|\, s = r', y = l\right) \\
&\quad + \left(\widehat{\mathbb{P}}\left(y = \bar{l} \,|\, s = r\right) - \widehat{\mathbb{P}}\left(y = \bar{l}, s = r\right)\right) \widehat{\mathbb{P}}\left(h_\theta(x) \neq y \,|\, s = r, y = \bar{l}\right) \\
&\quad - \sum_{(\bar{l},r') \neq (\bar{l},r)} \widehat{\mathbb{P}}\left(y = \bar{l}, s = r'\right) \widehat{\mathbb{P}}\left(h_\theta(x) \neq y \,|\, s = r', y = \bar{l}\right) \\
&\quad \widehat{\mathbb{P}}\left(y = \bar{l}\right) - \widehat{\mathbb{P}}\left(y = \bar{l} \,|\, s = r\right)
\end{aligned}
$$

where the law of total probability was used to obtain the last equation. Thus, Demographic Parity satisfies all our assumptions with $C_{(l,r)}^{(l,r)} = \widehat{\mathbb{P}}\left(y = l, s = r\right) - \widehat{\mathbb{P}}\left(y = l \,|\, s = r\right)$, $C_{(l,r)}^{(l,r')} = \widehat{\mathbb{P}}\left(y = l, s = r'\right)$ with $r' \neq r$, $C_{(l,r)}^{(\bar{l},r)} = \widehat{\mathbb{P}}\left(y = \bar{l} \,|\, s = r\right) - \widehat{\mathbb{P}}\left(y = \bar{l}, s = r\right)$, $C_{(l,r)}^{(\bar{l},r')} = -\widehat{\mathbb{P}}\left(y = \bar{l}, s = r'\right)$ with $r' \neq r$, and $C_{(l,r)}^0 = \widehat{\mathbb{P}}\left(y = \bar{l}\right) - \widehat{\mathbb{P}}\left(y = \bar{l} \,|\, s = r\right)$.

## C  Proof of Lemma 1

**Lemma** (Negative weights are necessary.). *Assume that the fairness notion under consideration is Accuracy Parity. Let $h_\theta^*$ be the most accurate and fair model. Then using negative weights is necessary as long as*

$$\min_{\substack{h_\theta \in \mathcal{H} \\ h_\theta \, unfair}} \max_{\mathcal{T}_k} \widehat{\mathbb{P}}\left(h_\theta(x) \neq y | \mathcal{T}_k\right) < \widehat{\mathbb{P}}\left(h_\theta^*(x) \neq y\right).$$

*Proof.* To prove this Lemma, one first need to notice that, for Accuracy Parity, since $\sum_{k=1}^K \widehat{\mathbb{P}}\left(\mathcal{T}_k\right) = 1$ we have that

$$\sum_{k'=1}^K C_k^{k'} = \left(\widehat{\mathbb{P}}\left(\mathcal{T}_k\right) - 1\right) + \sum_{\substack{k'=1 \\ k' \neq k}}^K \widehat{\mathbb{P}}\left(\mathcal{T}_{k'}\right) = 0.$$

This implies that

$$\sum_{k=1}^K \left[\widehat{\mathbb{P}}\left(\mathcal{T}_k\right) + \sum_{k'=1}^K C_{k'}^k \lambda_{k'}\right] = 1.$$

This implies that, whatever our choice of $\lambda$, the weights will always sum to one. In other words, since we also have that $\sum_{k=1}^K \lambda_k C_k^0 = 0$ by definition, for a given hypothesis $h_\theta$, we have that

$$\max_{\lambda_1, \dots, \lambda_K \in \mathbb{R}} \sum_{k=1}^K \widehat{\mathbb{P}}\left(h_\theta(x) \neq y | \mathcal{T}_k\right) \left[\widehat{\mathbb{P}}\left(\mathcal{T}_k\right) + \sum_{k'=1}^K C_{k'}^k \lambda_{k'}\right] \tag{5}$$

$$= \max_{\substack{w_1, \dots, w_K \in \mathbb{R} \\ s.t. \sum_k w_k = 1}} \sum_{k=1}^K \widehat{\mathbb{P}}\left(h_\theta(x) \neq y | \mathcal{T}_k\right) w_k \tag{6}$$

where, given $w_1, \ldots, w_K$, the original values of lambda can be obtained by solving the linear system $C\lambda = w$ where

$$C = \begin{pmatrix} C_1^1 & \cdots & C_K^1 \\ \vdots & & \vdots \\ C_1^K & \cdots & C_K^K \end{pmatrix}, \quad \lambda = \begin{pmatrix} \lambda_1 \\ \vdots \\ \lambda_K \end{pmatrix}, \quad w = \begin{pmatrix} w_1 - \widehat{\mathbb{P}}(\mathcal{T}_1) \\ \vdots \\ w_K - \widehat{\mathbb{P}}(\mathcal{T}_K) \end{pmatrix}$$

which is guaranteed to have infinitely many solutions since the rank of the matrix $C$ is $K - 1$ and the rank of the augmented matrix $(C|w)$ is also $K - 1$. Here we are using the fact that $\widehat{\mathbb{P}}(\mathcal{T}_k) \neq 0, \forall k$ since all the groups have to be represented to be taken into account.

We will now assume that all the weights are positive, that is $w_k \geq 0, \forall k$. Then, the best strategy to solve Problem (6) is to put all the weight on the worst off group $k$, that is set $w_k = 1$ and $w_{k'} = 0, \forall k' \neq k$. It implies that

$$\max_{\substack{w_1, \ldots, w_K \in \mathbb{R} \\ s.t. \sum_k w_k = 1}} \sum_{k=1}^{K} \widehat{\mathbb{P}}(h_\theta(x) \neq y | \mathcal{T}_k) w_k = \max_k \widehat{\mathbb{P}}(h_\theta(x) \neq y | \mathcal{T}_k).$$

Furthermore, notice that, for fair models with respect to Accuracy Parity, we have that $\widehat{\mathbb{P}}(h_\theta(x) \neq y | \mathcal{T}_k) = \widehat{\mathbb{P}}(h_\theta(x) \neq y), \forall k$. Thus, if it holds that

$$\min_{\substack{h_\theta \in \mathcal{H} \\ h_\theta \text{unfair}}} \max_{\mathcal{T}_k} \widehat{\mathbb{P}}(h_\theta(x) \neq y | \mathcal{T}_k) < \widehat{\mathbb{P}}(h_\theta^*(x) \neq y)$$

where $h_\theta^*$ is the most accurate and fair model, then the optimal solution of Problem (3) in the main paper will be unfair. It implies that, in this case, using positive weights is not sufficient and negative weights are necessary. $\qquad\square$

## D   FairGrad for $\epsilon$-fairness

To derive FairGrad for $\epsilon$-fairness we first consider the following standard optimization problem

$$\arg\min_{h_\theta \in \mathcal{H}} \widehat{\mathbb{P}}(h_\theta(x) \neq y)$$
$$\text{s.t. } \forall k \in [K], \widehat{F}_k(\mathcal{T}, h_\theta) \leq \epsilon$$
$$\forall k \in [K], \widehat{F}_k(\mathcal{T}, h_\theta) \geq -\epsilon.$$

We, once again, use a standard multipliers approach to obtain the following unconstrained formulation:

$$\mathcal{L}(h_\theta, \lambda_1, \ldots, \lambda_K, \delta_1, \ldots, \delta_K) = \widehat{\mathbb{P}}(h_\theta(x) \neq y) + \sum_{k=1}^{K} \lambda_k \left(\widehat{F}_k(\mathcal{T}, h_\theta) - \epsilon\right) - \delta_k \left(\widehat{F}_k(\mathcal{T}, h_\theta) + \epsilon\right) \quad (7)$$

where $\lambda_1, \ldots, \lambda_K$ and $\delta_1, \ldots, \delta_K$ are the multipliers that belong to $\mathbb{R}^+$, that is the set of positive reals. Once again, to solve this problem, we will use an alternating approach where the hypothesis and the multipliers are updated one after the other.

**Updating the Multipliers.**   To update the values $\lambda_1, \ldots, \lambda_K$, we will use a standard gradient ascent procedure. Hence, noting that the gradient of the previous formulation is

$$\nabla_{\lambda_1, \ldots, \lambda_K} \mathcal{L}(h_\theta, \lambda_1, \ldots, \lambda_K, \delta_1, \ldots, \delta_K) = \begin{pmatrix} \widehat{F}_1(\mathcal{T}, h_\theta) - \epsilon \\ \vdots \\ \widehat{F}_K(\mathcal{T}, h_\theta) - \epsilon \end{pmatrix}$$

$$\nabla_{\delta_1,\ldots,\delta_K} \mathcal{L}\left(h_\theta, \lambda_1, \ldots, \lambda_K, \delta_1, \ldots, \delta_K\right) = \begin{pmatrix} -\widehat{F}_1(\mathcal{T}, h_\theta) - \epsilon \\ \vdots \\ -\widehat{F}_K(\mathcal{T}, h_\theta) - \epsilon \end{pmatrix}$$

we have the following update rule $\forall k \in [K]$

$$\lambda_k^{T+1} = \max\left(0, \lambda_k^T + \eta\left(\widehat{F}_k\left(\mathcal{T}, h_\theta^T\right) - \epsilon\right)\right)$$

$$\delta_k^{T+1} = \max\left(0, \delta_k^T - \eta\left(\widehat{F}_k\left(\mathcal{T}, h_\theta^T\right) + \epsilon\right)\right)$$

where $\eta$ is a fairness rate that controls the importance of each weight update.

**Updating the Model.** To update the parameters $\theta \in \mathbb{R}^D$ of the model $h_\theta$, we proceed as before, using a gradient descent approach. However, first, we notice that given the fairness notions that we consider, Equation (7) is equivalent to

$$\mathcal{L}\left(h_\theta, \lambda_1, \ldots, \lambda_K, \delta_1, \ldots, \delta_K\right) = \sum_{k=1}^{K} \widehat{\mathbb{P}}\left(h_\theta(x) \neq y | \mathcal{T}_k\right) \left[\widehat{\mathbb{P}}\left(\mathcal{T}_k\right) + \sum_{k'=1}^{K} C_{k'}^k\left(\lambda_{k'} - \delta_{k'}\right)\right] \qquad (8)$$
$$- \sum_{k=1}^{K}\left(\lambda_k + \delta_k\right)\epsilon + \sum_{k=1}^{K}(\lambda_k - \delta_k)C_k^0.$$

Since the additional terms in the optimization problem do not depend on $h_\theta$, the main difference between exact and $\epsilon$-fairness is the nature of the weights. More precisely, at iteration $t$, the update rule becomes

$$\theta^{T+1} = \theta^T - \eta_\theta \sum_{k=1}^{K} \left[\widehat{\mathbb{P}}\left(\mathcal{T}_k\right) + \sum_{k'=1}^{K} C_{k'}^k\left(\lambda_{k'} - \delta_{k'}\right)\right] \nabla_\theta \widehat{\mathbb{P}}\left(h_\theta(x) \neq y | \mathcal{T}_k\right)$$

where $\eta_\theta$ is a learning rate. Once again, we obtain a simple re-weighting scheme where the weights depend on the current fairness level of the model through $\lambda_1, \ldots, \lambda_K$ and $\delta_1, \ldots, \delta_K$, the relative size of each group through $\widehat{\mathbb{P}}\left(\mathcal{T}_k\right)$, and the fairness notion through the constants $C$.

## E    Extended Experiments

In this section, we provide additional details related to the baselines and the hyper-parameters tuning procedure. We then provide descriptions of the datasets and finally the results.

### E.1    Baselines

- **Adversarial**: One of the common ways of removing sensitive information from the model's representation is via adversarial learning. Broadly, it consists of three components, namely an encoder, a task classifier, and an adversary. On the one hand, the objective of the adversary is to predict sensitive information from the encoder. On the other hand, the encoder aims to create representations that are useful for the downstream task (task classifier) and, at the same time, fool the adversary. The adversary is generally connected to the encoder via a gradient reversal layer (Ganin & Lempitsky, 2015) which acts like an identity function during the forward pass and scales the loss with a parameter $-\lambda$ during the backward pass. In our setting, the encoder is a Multi-Layer Perceptron with two hidden layers of size 64 and 128 respectively, and the task classifier is another Multi-Layer Perceptron with a single hidden layer of size 32. The adversary is the same as the main task classifier. We use a ReLU as the activation function with the dropout set to 0.2 and employ batch normalization with default PyTorch parameters. As a part of the hyper-parameter tuning, we did a grid search over $\lambda$, varying it between 0.1 to 3.0 with an interval of 0.2.

- **BiFair (Ozdayi et al., 2021)**: For this baseline, we fix the weight parameter to be of length 8 as suggested in the code released by the authors[5]. In this fixed setting, we perform a grid search over the following hyper-parameters:

  - Batch Size: 128, 256, 512
  - Weight Decay: 0.0, 0.001
  - Fairness Loss Weight: 0.5, 1, 2, 4
  - Inner Loop Length: 5, 25, 50

- **Constraints**: We use the implementation available in the TensorFlow Constrained Optimization[6] library with default hyper-parameters.

- **FairBatch**: We use the implementation publicly released by the authors[7].

- **Weighted ERM**: We reweigh each example in the dataset based on inverse of the proportion of the sensitive group it belongs to.

- **Reduction**: We use the implementation available in the Fairlearn[8] with default hyper-parameters.

In our initial experiments, we varied the batch size, and learning rates for both Constraints and FairBatch. However, we found that the default hyper-parameters as specified by the authors result in the best performances. In the spirit of being comparable in terms of hyper-parameter search budget, we also fix all hyper-parameters of FairGrad, apart from the batch size and weight decay. We experiment with two different batch sizes namely, 64 or 512 for the standard fairness dataset. Similarly, we also experiment with three weight decay values namely, 0.0, 0.001 and 0.01. Note that we also vary weight decay and batch sizes for FairBatch, Adversarial, Unconstrained, and BiFair.

For all our experiments, apart from BiFair, we use Batch Gradient Descent as the optimizer with a learning rate of 0.1 and a gradient clipping of 0.05 to avoid exploding gradients. For BiFair, we employ the Adam optimizer as suggested by the authors with a learning rate of 0.001. For FairGrad, FairBatch and Unconstrained, we considered 6 hyper-parameters combinations. For BiFair, we considered 72 such combinations, while for Adversarial, there were 90 combinations.

## E.2 Datasets

Here, we provide additional details on the datasets used in our experiments. We begin by describing the standard fairness datasets for which we follow the pre-processing procedure described in Lohaus et al. (2020).

- **Adult**[9]: The dataset (Kohavi, 1996) is composed of 45222 instances, with 14 features each describing several attributes of a person. The objective is to predict the income of a person (below or above $50k$) while remaining fair with respect to gender (binary in this case). Following the pre-processing step of Wu et al. (2019), only 9 features were used for training.

- **CelebA**[10]: The dataset (Liu et al., 2015) consists of $202,599$ images, along with 40 binary attributes associated with each image. We use 38 of these as features while keeping gender as the sensitive attribute and "Smiling" as the class label.

- **Dutch**[11]: The dataset (Žliobaite et al., 2011) is composed of $60,420$ instances with each instance described by 12 features. We predict "Low Income" or "High Income" as dictated by the occupation as the main classification task and gender as the sensitive attribute.

---

[5]https://github.com/TinfoilHat0/BiFair
[6]https://github.com/google-research/tensorflow_constrained_optimization
[7]https://github.com/yuji-roh/fairbatch
[8]https://fairlearn.org/
[9]https://archive.ics.uci.edu/ml/datasets/adult
[10]https://mmlab.ie.cuhk.edu.hk/projects/CelebA.html
[11]https://sites.google.com/site/conditionaldiscrimination/

- **Compas**[12]: The dataset (Larson et al., 2016) contains 6172 data points, where each data point has 53 features. The goal is to predict if the defendant will be arrested again within two years of the decision. The sensitive attribute is race, which has been merged into "White" and "Non White" categories.

- **Communities and Crime**[13]: The dataset (Redmond & Baveja, 2002) is composed of 1994 instances with 128 features, of which 29 have been dropped. The objective is to predict the number of violent crimes in the community, with race being the sensitive attribute.

- **German Credit**[14]: The dataset (Dua et al., 2017) consists of 1000 instances, with each having 20 attributes. The objective is to predict a person's creditworthiness (binary), with gender being the sensitive attribute.

- **Gaussian**[15]: It is a toy dataset with binary task label and binary sensitive attribute, introduced in Lohaus et al. (2020). It is constructed by drawing points from different Gaussian distributions. We follow the same mechanism as described in Lohaus et al. (2020), and sample 50000 data points for each class.

- **Adult Folktables**[16]: This dataset (Ding et al., 2021) is an updated version of the original Adult Income dataset. We use California census data with gender as the sensitive attribute. There are 195665 instances, with 9 features describing several attributes of a person. We use the same preprocessing step as recommended by the authors.

For all these datasets, we use a 20% of the data as a test set and 80% as a train set. We further divide the train set into two and keep 25% of the training examples as a validation set. For each repetition, we randomly shuffle the data before splitting it, and thus we had unique splits for each random seed. We use the following seeds: $10, 20, 30, 40, 50$ for all our experiments. As a last pre-processing step, we centered and scaled each feature independently by substracting the mean and dividing by the standard deviation both of which were estimated on the training set.

**Twitter Sentiment Analysis**[17]: The dataset (Blodgett et al., 2016) consists of $200k$ tweets with binary sensitive attribute (race) and binary sentiment score. We follow the setup proposed by Han et al. (2021) and Elazar & Goldberg (2018) and create bias in the dataset by changing the proportion of each subgroup (race-sentiment) in the training set. With two sentiment classes being happy and sad, and two race classes being AAE and SAE, the training data consists of 40% AAE-happy, 10% AAE-sad, 10% SAE-happy, and 40% SAE-sad. The test set remains balanced. The tweets are encoded using the DeepMoji (Felbo et al., 2017) encoder with no fine-tuning, which has been pre-trained over millions of tweets to predict their emoji, thereby predicting the sentiment. Note that the train-test splits are pre-defined and thus do not change based on the random seed of the repetition.

### E.3 Detailed Results

---

[12]https://github.com/propublica/compas-analysis
[13]http://archive.ics.uci.edu/ml/datasets/communities+and+crime
[14]https://archive.ics.uci.edu/ml/datasets/Statlog+%28German+Credit+Data%29
[15]https://github.com/mlohaus/SearchFair/blob/master/examples/get_synthetic_data.py
[16]https://github.com/zykls/folktables
[17]https://slanglab.cs.umass.edu/TwitterAAE/

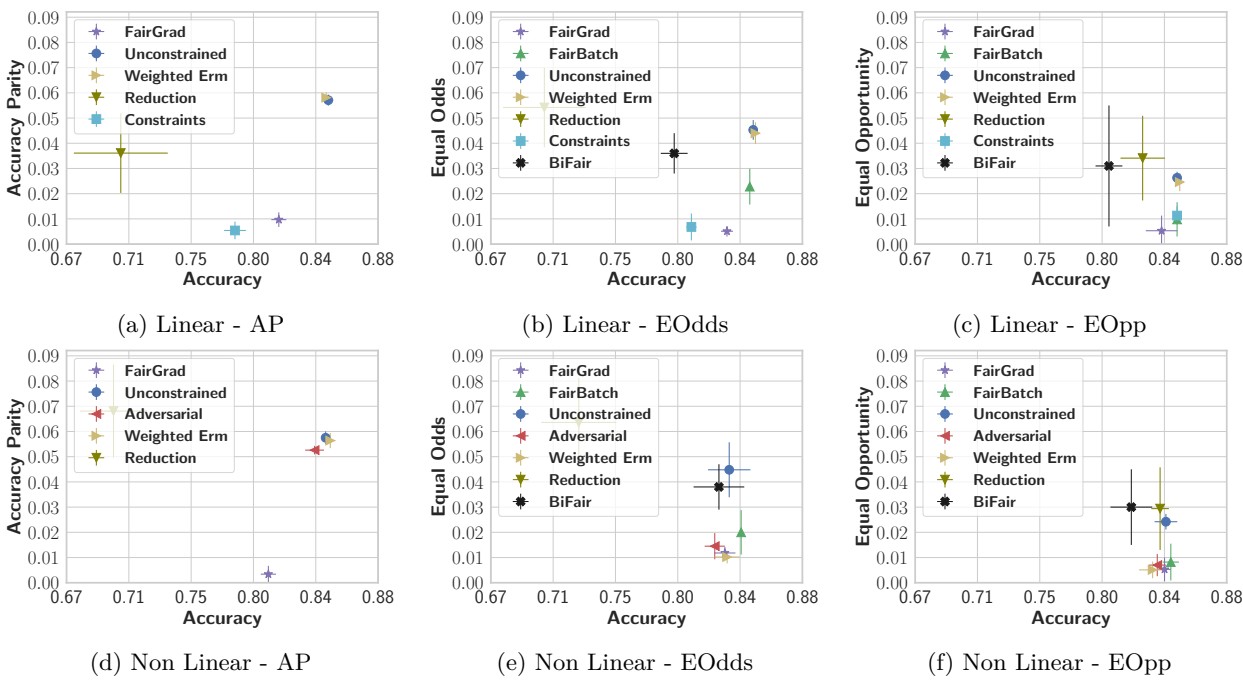

Figure 6: Results for the Adult dataset with different fairness measures.

Table 4: Results for the Adult dataset with Linear Models. All the results are averaged over 5 runs. Here MEAN ABS., MAXIMUM, and MINIMUM represent the mean absolute fairness value, the fairness level of the most well-off group, and the fairness level of the worst-off group, respectively.

| METHOD (L) | ACCURACY ↑ | FAIRNESS | | | |
|---|---|---|---|---|---|
| | | MEASURE | MEAN ABS. ↓ | MAXIMUM | MINIMUM |
| Unconstrained | $0.8456 \pm 0.0033$ | **AP** | $0.0571 \pm 0.0022$ | $0.077 \pm 0.0029$ | $-0.0373 \pm 0.0017$ |
| Constant | $0.751 \pm 0.0$ | **AP** | $0.102 \pm 0.0$ | $0.138 \pm 0.0$ | $0.067 \pm 0.0$ |
| Weighted ERM | $0.8442 \pm 0.0016$ | **AP** | $0.0581 \pm 0.0021$ | $0.0783 \pm 0.0028$ | $-0.0379 \pm 0.0014$ |
| Constrained | $0.783 \pm 0.007$ | **AP** | $0.005 \pm 0.003$ | $0.007 \pm 0.005$ | $0.004 \pm 0.002$ |
| Reduction | $0.7064 \pm 0.0315$ | **AP** | $0.0361 \pm 0.0158$ | $0.0235 \pm 0.0103$ | $-0.0487 \pm 0.0214$ |
| FairGrad | $0.8124 \pm 0.005$ | **AP** | $0.0097 \pm 0.0029$ | $0.0131 \pm 0.004$ | $-0.0063 \pm 0.0019$ |
| Unconstrained | $0.846 \pm 0.0028$ | **Eodds** | $0.0453 \pm 0.0039$ | $0.048 \pm 0.0043$ | $-0.0878 \pm 0.01$ |
| Constant | $0.748 \pm 0.0$ | **Eodds** | $0.0 \pm 0.0$ | $0.0 \pm 0.0$ | $0.0 \pm 0.0$ |
| Weighted ERM | $0.8475 \pm 0.0024$ | **Eodds** | $0.044 \pm 0.0043$ | $0.0477 \pm 0.0031$ | $-0.0837 \pm 0.0124$ |
| Constrained | $0.805 \pm 0.004$ | **Eodds** | $0.007 \pm 0.005$ | $0.019 \pm 0.017$ | $0.002 \pm 0.001$ |
| BiFair | $0.793 \pm 0.009$ | **Eodds** | $0.036 \pm 0.008$ | $0.085 \pm 0.027$ | $-0.03 \pm 0.016$ |
| FairBatch | $0.8437 \pm 0.0013$ | **Eodds** | $0.0228 \pm 0.0071$ | $0.0411 \pm 0.0105$ | $-0.0245 \pm 0.0183$ |
| Reduction | $0.7059 \pm 0.0277$ | **Eodds** | $0.0542 \pm 0.0158$ | $0.0711 \pm 0.0189$ | $-0.1055 \pm 0.022$ |
| FairGrad | $0.8284 \pm 0.004$ | **Eodds** | $0.0051 \pm 0.0021$ | $0.0078 \pm 0.0068$ | $-0.0078 \pm 0.0054$ |
| Unconstrained | $0.8457 \pm 0.0028$ | **Eopp** | $0.0263 \pm 0.0024$ | $0.0157 \pm 0.0011$ | $-0.0893 \pm 0.0083$ |
| Constant | $0.754 \pm 0.0$ | **Eopp** | $0.0 \pm 0.0$ | $0.0 \pm 0.0$ | $0.0 \pm 0.0$ |
| Weighted ERM | $0.8475 \pm 0.0024$ | **Eopp** | $0.0246 \pm 0.0036$ | $0.0148 \pm 0.002$ | $-0.0837 \pm 0.0124$ |
| Constrained | $0.846 \pm 0.002$ | **Eopp** | $0.011 \pm 0.004$ | $0.039 \pm 0.012$ | $0.0 \pm 0.0$ |
| BiFair | $0.8 \pm 0.009$ | **Eopp** | $0.031 \pm 0.024$ | $0.019 \pm 0.014$ | $-0.107 \pm 0.083$ |
| FairBatch | $0.8457 \pm 0.0016$ | **Eopp** | $0.0098 \pm 0.0068$ | $0.0225 \pm 0.0174$ | $-0.0166 \pm 0.0241$ |
| Reduction | $0.8226 \pm 0.0149$ | **Eopp** | $0.0341 \pm 0.0168$ | $0.116 \pm 0.0575$ | $-0.0204 \pm 0.0098$ |
| FairGrad | $0.8353 \pm 0.0106$ | **Eopp** | $0.0053 \pm 0.006$ | $0.0177 \pm 0.021$ | $-0.0037 \pm 0.0033$ |

Table 5: Results for the Adult dataset with Non Linear Models. All the results are averaged over 5 runs. Here MEAN ABS., MAXIMUM, and MINIMUM represent the mean absolute fairness value, the fairness level of the most well-off group, and the fairness level of the worst-off group, respectively.

| METHOD (NL) | ACCURACY ↑ | FAIRNESS | | | |
|---|---|---|---|---|---|
| | | MEASURE | MEAN ABS. ↓ | MAXIMUM | MINIMUM |
| Unconstrained | $0.8438 \pm 0.0025$ | **AP** | $0.0575 \pm 0.0025$ | $0.0776 \pm 0.0033$ | $-0.0375 \pm 0.0018$ |
| Constant | $0.751 \pm 0.0$ | **AP** | $0.102 \pm 0.0$ | $0.138 \pm 0.0$ | $0.067 \pm 0.0$ |
| Weighted ERM | $0.8469 \pm 0.0035$ | **AP** | $0.0564 \pm 0.003$ | $0.0761 \pm 0.0038$ | $-0.0368 \pm 0.0021$ |
| Adversarial | $0.8364 \pm 0.0063$ | **AP** | $0.0526 \pm 0.0017$ | $0.0709 \pm 0.0025$ | $-0.0343 \pm 0.0009$ |
| Reduction | $0.7015 \pm 0.0225$ | **AP** | $0.0681 \pm 0.0184$ | $0.0444 \pm 0.0122$ | $-0.0917 \pm 0.0247$ |
| FairGrad | $0.8054 \pm 0.0051$ | **AP** | $0.0034 \pm 0.0033$ | $0.0033 \pm 0.0031$ | $-0.0036 \pm 0.0042$ |
| Unconstrained | $0.8299 \pm 0.0142$ | **Eodds** | $0.0448 \pm 0.0109$ | $0.0404 \pm 0.0136$ | $-0.0977 \pm 0.0422$ |
| Constant | $0.748 \pm 0.0$ | **Eodds** | $0.0 \pm 0.0$ | $0.0 \pm 0.0$ | $0.0 \pm 0.0$ |
| Weighted ERM | $0.8285 \pm 0.0085$ | **Eodds** | $0.0102 \pm 0.0025$ | $0.0196 \pm 0.0102$ | $-0.0099 \pm 0.0047$ |
| Adversarial | $0.8202 \pm 0.0068$ | **Eodds** | $0.0145 \pm 0.0052$ | $0.0288 \pm 0.0177$ | $-0.0153 \pm 0.0067$ |
| BiFair | $0.823 \pm 0.017$ | **Eodds** | $0.038 \pm 0.009$ | $0.09 \pm 0.034$ | $-0.038 \pm 0.015$ |
| FairBatch | $0.8379 \pm 0.0009$ | **Eodds** | $0.02 \pm 0.0088$ | $0.0327 \pm 0.0153$ | $-0.0244 \pm 0.0218$ |
| Reduction | $0.729 \pm 0.0252$ | **Eodds** | $0.0636 \pm 0.0176$ | $0.0673 \pm 0.0203$ | $-0.115 \pm 0.0334$ |
| FairGrad | $0.827 \pm 0.0071$ | **Eodds** | $0.0118 \pm 0.0024$ | $0.022 \pm 0.014$ | $-0.0165 \pm 0.0135$ |
| Unconstrained | $0.8382 \pm 0.0076$ | **Eopp** | $0.0242 \pm 0.0031$ | $0.0145 \pm 0.0017$ | $-0.0822 \pm 0.0108$ |
| Constant | $0.754 \pm 0.0$ | **Eopp** | $0.0 \pm 0.0$ | $0.0 \pm 0.0$ | $0.0 \pm 0.0$ |
| Weighted ERM | $0.8293 \pm 0.0091$ | **Eopp** | $0.0051 \pm 0.0033$ | $0.0141 \pm 0.0137$ | $-0.0062 \pm 0.0038$ |
| Adversarial | $0.8324 \pm 0.0058$ | **Eopp** | $0.007 \pm 0.0044$ | $0.0139 \pm 0.0159$ | $-0.0144 \pm 0.0133$ |
| BiFair | $0.815 \pm 0.014$ | **Eopp** | $0.03 \pm 0.015$ | $0.019 \pm 0.009$ | $-0.103 \pm 0.053$ |
| FairBatch | $0.8415 \pm 0.0054$ | **Eopp** | $0.0082 \pm 0.0073$ | $0.0157 \pm 0.0121$ | $-0.017 \pm 0.0271$ |
| Reduction | $0.8343 \pm 0.0059$ | **Eopp** | $0.0294 \pm 0.0164$ | $0.0779 \pm 0.0662$ | $-0.0396 \pm 0.0455$ |
| FairGrad | $0.8373 \pm 0.0043$ | **Eopp** | $0.0053 \pm 0.0047$ | $0.0099 \pm 0.0146$ | $-0.0112 \pm 0.0127$ |

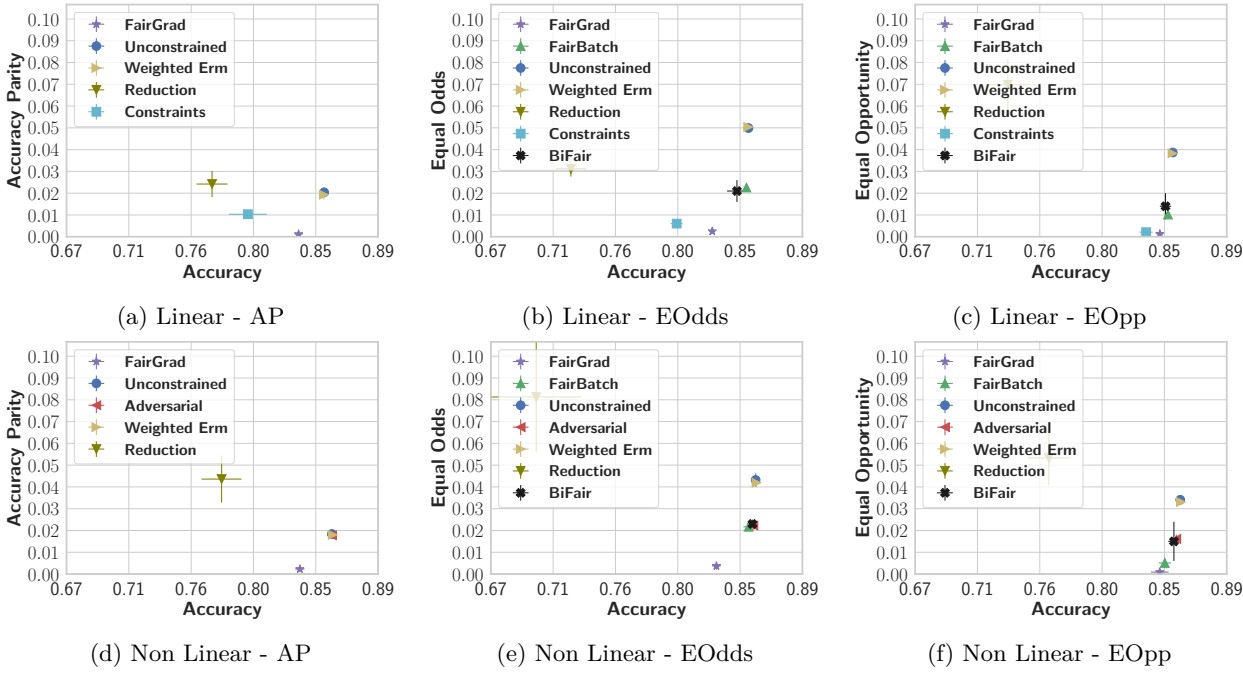

Figure 7: Results for the CelebA dataset with different fairness measures.

Table 6: Results for the CelebA dataset with Linear Models. All the results are averaged over 5 runs. Here MEAN ABS., MAXIMUM, and MINIMUM represent the mean absolute fairness value, the fairness level of the most well-off group, and the fairness level of the worst-off group, respectively.

| METHOD (L) | ACCURACY ↑ | FAIRNESS | | |
|---|---|---|---|---|
| | | MEASURE | MEAN ABS. ↓ | MAXIMUM | MINIMUM |
| Unconstrained | $0.8532 \pm 0.0009$ | **AP** | $0.0204 \pm 0.0022$ | $0.017 \pm 0.0019$ | $-0.0238 \pm 0.0025$ |
| Constant | $0.516 \pm 0.0$ | **AP** | $0.072 \pm 0.0$ | $0.084 \pm 0.0$ | $0.06 \pm 0.0$ |
| Weighted ERM | $0.853 \pm 0.0008$ | **AP** | $0.0193 \pm 0.0021$ | $0.0161 \pm 0.0018$ | $-0.0225 \pm 0.0023$ |
| Constrained | $0.799 \pm 0.013$ | **AP** | $0.01 \pm 0.001$ | $0.012 \pm 0.002$ | $0.009 \pm 0.001$ |
| Reduction | $0.7734 \pm 0.011$ | **AP** | $0.0242 \pm 0.006$ | $0.0282 \pm 0.0071$ | $-0.0201 \pm 0.005$ |
| FairGrad | $0.835 \pm 0.0028$ | **AP** | $0.0012 \pm 0.0009$ | $0.0011 \pm 0.0007$ | $-0.0014 \pm 0.0011$ |
| Unconstrained | $0.8532 \pm 0.0009$ | **Eodds** | $0.0499 \pm 0.0019$ | $0.0538 \pm 0.0024$ | $-0.1011 \pm 0.0033$ |
| Constant | $0.518 \pm 0.0$ | **Eodds** | $0.0 \pm 0.0$ | $0.0 \pm 0.0$ | $0.0 \pm 0.0$ |
| Weighted ERM | $0.853 \pm 0.0009$ | **Eodds** | $0.0504 \pm 0.0019$ | $0.0532 \pm 0.0024$ | $-0.1001 \pm 0.0032$ |
| Constrained | $0.802 \pm 0.004$ | **Eodds** | $0.006 \pm 0.001$ | $0.01 \pm 0.003$ | $0.002 \pm 0.001$ |
| BiFair | $0.845 \pm 0.007$ | **Eodds** | $0.021 \pm 0.005$ | $0.02 \pm 0.003$ | $-0.036 \pm 0.009$ |
| FairBatch | $0.8518 \pm 0.0009$ | **Eodds** | $0.0226 \pm 0.0017$ | $0.0218 \pm 0.0028$ | $-0.0411 \pm 0.0053$ |
| Reduction | $0.7268 \pm 0.011$ | **Eodds** | $0.0312 \pm 0.0036$ | $0.0628 \pm 0.0089$ | $-0.0334 \pm 0.0047$ |
| FairGrad | $0.8274 \pm 0.002$ | **Eodds** | $0.0025 \pm 0.0009$ | $0.0038 \pm 0.0018$ | $-0.0046 \pm 0.0026$ |
| Unconstrained | $0.8532 \pm 0.0009$ | **Eopp** | $0.0387 \pm 0.0014$ | $0.0538 \pm 0.0024$ | $-0.1011 \pm 0.0033$ |
| Constant | $0.518 \pm 0.0$ | **Eopp** | $0.0 \pm 0.0$ | $0.0 \pm 0.0$ | $0.0 \pm 0.0$ |
| Weighted ERM | $0.853 \pm 0.0008$ | **Eopp** | $0.0383 \pm 0.0014$ | $0.0531 \pm 0.0024$ | $-0.0999 \pm 0.0032$ |
| Constrained | $0.834 \pm 0.005$ | **Eopp** | $0.002 \pm 0.001$ | $0.005 \pm 0.002$ | $0.0 \pm 0.0$ |
| BiFair | $0.848 \pm 0.004$ | **Eopp** | $0.014 \pm 0.006$ | $0.02 \pm 0.009$ | $-0.037 \pm 0.017$ |
| FairBatch | $0.8498 \pm 0.001$ | **Eopp** | $0.0102 \pm 0.0016$ | $0.0142 \pm 0.0022$ | $-0.0268 \pm 0.0042$ |
| Reduction | $0.7358 \pm 0.0159$ | **Eopp** | $0.0698 \pm 0.0118$ | $0.1824 \pm 0.0313$ | $-0.0968 \pm 0.0158$ |
| FairGrad | $0.844 \pm 0.0022$ | **Eopp** | $0.0013 \pm 0.0009$ | $0.0025 \pm 0.0021$ | $-0.0028 \pm 0.0018$ |

Table 7: Results for the CelebA dataset with Non Linear Models. All the results are averaged over 5 runs. Here MEAN ABS., MAXIMUM, and MINIMUM represent the mean absolute fairness value, the fairness level of the most well-off group, and the fairness level of the worst-off group, respectively.

| METHOD (NL) | ACCURACY ↑ | FAIRNESS | | | |
|---|---|---|---|---|---|
| | | MEASURE | MEAN ABS. ↓ | MAXIMUM | MINIMUM |
| Unconstrained | $0.8587 \pm 0.0015$ | **AP** | $0.0184 \pm 0.0014$ | $0.0154 \pm 0.0012$ | $-0.0215 \pm 0.0016$ |
| Constant | $0.516 \pm 0.0$ | **AP** | $0.072 \pm 0.0$ | $0.084 \pm 0.0$ | $0.06 \pm 0.0$ |
| Weighted ERM | $0.8593 \pm 0.0018$ | **AP** | $0.018 \pm 0.0017$ | $0.015 \pm 0.0014$ | $-0.021 \pm 0.0019$ |
| Adversarial | $0.8588 \pm 0.0012$ | **AP** | $0.0178 \pm 0.0014$ | $0.0148 \pm 0.0012$ | $-0.0208 \pm 0.0015$ |
| Reduction | $0.7802 \pm 0.0142$ | **AP** | $0.0436 \pm 0.0108$ | $0.0508 \pm 0.0123$ | $-0.0364 \pm 0.0092$ |
| FairGrad | $0.8359 \pm 0.0033$ | **AP** | $0.0023 \pm 0.0012$ | $0.0025 \pm 0.0015$ | $-0.0021 \pm 0.0009$ |
| Unconstrained | $0.8583 \pm 0.0012$ | **Eodds** | $0.0432 \pm 0.003$ | $0.0475 \pm 0.0028$ | $-0.0893 \pm 0.0049$ |
| Constant | $0.518 \pm 0.0$ | **Eodds** | $0.0 \pm 0.0$ | $0.0 \pm 0.0$ | $0.0 \pm 0.0$ |
| Weighted ERM | $0.8589 \pm 0.0009$ | **Eodds** | $0.0419 \pm 0.0021$ | $0.0459 \pm 0.0025$ | $-0.0864 \pm 0.0038$ |
| Adversarial | $0.8567 \pm 0.0014$ | **Eodds** | $0.0223 \pm 0.002$ | $0.0272 \pm 0.0039$ | $-0.0511 \pm 0.0073$ |
| BiFair | $0.856 \pm 0.004$ | **Eodds** | $0.023 \pm 0.002$ | $0.028 \pm 0.005$ | $-0.052 \pm 0.009$ |
| FairBatch | $0.8533 \pm 0.0037$ | **Eodds** | $0.0217 \pm 0.0014$ | $0.0197 \pm 0.0026$ | $-0.0321 \pm 0.005$ |
| Reduction | $0.7021 \pm 0.0323$ | **Eodds** | $0.0813 \pm 0.0253$ | $0.1777 \pm 0.0426$ | $-0.0946 \pm 0.0238$ |
| FairGrad | $0.8304 \pm 0.0031$ | **Eodds** | $0.0037 \pm 0.0017$ | $0.0048 \pm 0.0018$ | $-0.0055 \pm 0.0023$ |
| Unconstrained | $0.8585 \pm 0.0016$ | **Eopp** | $0.0341 \pm 0.002$ | $0.0473 \pm 0.003$ | $-0.0889 \pm 0.0052$ |
| Constant | $0.518 \pm 0.0$ | **Eopp** | $0.0 \pm 0.0$ | $0.0 \pm 0.0$ | $0.0 \pm 0.0$ |
| Weighted ERM | $0.859 \pm 0.0009$ | **Eopp** | $0.0331 \pm 0.0014$ | $0.046 \pm 0.0023$ | $-0.0866 \pm 0.0035$ |
| Adversarial | $0.8557 \pm 0.0019$ | **Eopp** | $0.0161 \pm 0.002$ | $0.0223 \pm 0.0029$ | $-0.0419 \pm 0.0053$ |
| BiFair | $0.854 \pm 0.004$ | **Eopp** | $0.015 \pm 0.009$ | $0.021 \pm 0.012$ | $-0.039 \pm 0.022$ |
| FairBatch | $0.8475 \pm 0.0043$ | **Eopp** | $0.0051 \pm 0.0024$ | $0.007 \pm 0.0033$ | $-0.0131 \pm 0.0063$ |
| Reduction | $0.765 \pm 0.0149$ | **Eopp** | $0.0533 \pm 0.0124$ | $0.1393 \pm 0.033$ | $-0.0738 \pm 0.0167$ |
| FairGrad | $0.8439 \pm 0.0063$ | **Eopp** | $0.0009 \pm 0.0008$ | $0.002 \pm 0.0022$ | $-0.0016 \pm 0.0011$ |

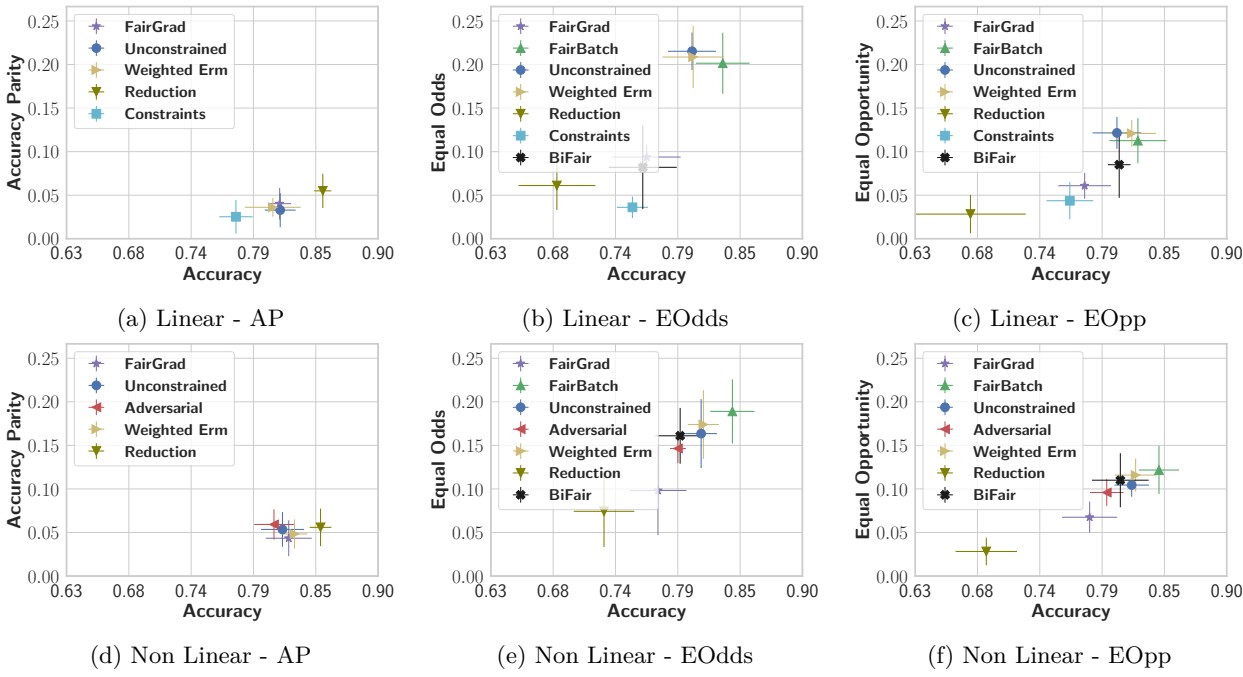

Figure 8: Results for the Crime dataset with different fairness measures.

Table 8: Results for the Crime dataset with Linear Models. All the results are averaged over 5 runs. Here MEAN ABS., MAXIMUM, and MINIMUM represent the mean absolute fairness value, the fairness level of the most well-off group, and the fairness level of the worst-off group, respectively.

| METHOD (L) | ACCURACY ↑ | FAIRNESS | | | |
|---|---|---|---|---|---|
| | | MEASURE | MEAN ABS. ↓ | MAXIMUM | MINIMUM |
| Unconstrained | $0.8145 \pm 0.0136$ | **AP** | $0.0329 \pm 0.0195$ | $0.0258 \pm 0.0162$ | $-0.0399 \pm 0.0229$ |
| Constant | $0.734 \pm 0.0$ | **AP** | $0.272 \pm 0.0$ | $0.377 \pm 0.0$ | $0.168 \pm 0.0$ |
| Weighted ERM | $0.808 \pm 0.0246$ | **AP** | $0.0361 \pm 0.0108$ | $0.0284 \pm 0.0091$ | $-0.0438 \pm 0.0129$ |
| Constrained | $0.775 \pm 0.015$ | **AP** | $0.025 \pm 0.019$ | $0.031 \pm 0.025$ | $0.019 \pm 0.014$ |
| Reduction | $0.8521 \pm 0.0075$ | **AP** | $0.055 \pm 0.0197$ | $0.0426 \pm 0.0147$ | $-0.0673 \pm 0.0253$ |
| FairGrad | $0.814 \pm 0.0102$ | **AP** | $0.0403 \pm 0.0181$ | $0.0316 \pm 0.0147$ | $-0.049 \pm 0.0218$ |
| Unconstrained | $0.8035 \pm 0.0212$ | **Eodds** | $0.2152 \pm 0.0215$ | $0.1038 \pm 0.0231$ | $-0.396 \pm 0.0433$ |
| Constant | $0.677 \pm 0.0$ | **Eodds** | $0.0 \pm 0.0$ | $0.0 \pm 0.0$ | $0.0 \pm 0.0$ |
| Weighted ERM | $0.8045 \pm 0.0271$ | **Eodds** | $0.2086 \pm 0.0357$ | $0.0974 \pm 0.0165$ | $-0.3747 \pm 0.0679$ |
| Constrained | $0.751 \pm 0.014$ | **Eodds** | $0.036 \pm 0.012$ | $0.088 \pm 0.043$ | $0.007 \pm 0.004$ |
| BiFair | $0.76 \pm 0.03$ | **Eodds** | $0.082 \pm 0.048$ | $0.048 \pm 0.03$ | $-0.163 \pm 0.092$ |
| FairBatch | $0.8306 \pm 0.0237$ | **Eodds** | $0.2015 \pm 0.035$ | $0.1054 \pm 0.0333$ | $-0.3704 \pm 0.067$ |
| Reduction | $0.6842 \pm 0.0339$ | **Eodds** | $0.0611 \pm 0.0281$ | $0.0349 \pm 0.0111$ | $-0.1291 \pm 0.047$ |
| FairGrad | $0.7634 \pm 0.03$ | **Eodds** | $0.0938 \pm 0.0144$ | $0.0491 \pm 0.016$ | $-0.1927 \pm 0.0362$ |
| Unconstrained | $0.804 \pm 0.0215$ | **Eopp** | $0.1215 \pm 0.0183$ | $0.1009 \pm 0.0238$ | $-0.3852 \pm 0.0549$ |
| Constant | $0.697 \pm 0.0$ | **Eopp** | $0.0 \pm 0.0$ | $0.0 \pm 0.0$ | $0.0 \pm 0.0$ |
| Weighted ERM | $0.8171 \pm 0.0213$ | **Eopp** | $0.1209 \pm 0.0154$ | $0.0985 \pm 0.0106$ | $-0.3851 \pm 0.0599$ |
| Constrained | $0.762 \pm 0.021$ | **Eopp** | $0.044 \pm 0.021$ | $0.138 \pm 0.066$ | $0.0 \pm 0.0$ |
| BiFair | $0.806 \pm 0.01$ | **Eopp** | $0.085 \pm 0.038$ | $0.073 \pm 0.042$ | $-0.268 \pm 0.112$ |
| FairBatch | $0.8225 \pm 0.0252$ | **Eopp** | $0.1126 \pm 0.0259$ | $0.1002 \pm 0.0281$ | $-0.3501 \pm 0.0821$ |
| Reduction | $0.6747 \pm 0.0488$ | **Eopp** | $0.0283 \pm 0.022$ | $0.0413 \pm 0.0375$ | $-0.0718 \pm 0.0829$ |
| FairGrad | $0.7755 \pm 0.0233$ | **Eopp** | $0.0609 \pm 0.0149$ | $0.0507 \pm 0.0166$ | $-0.193 \pm 0.0456$ |

Table 9: Results for the Crime dataset with Non Linear Models. All the results are averaged over 5 runs. Here MEAN ABS., MAXIMUM, and MINIMUM represent the mean absolute fairness value, the fairness level of the most well-off group, and the fairness level of the worst-off group, respectively.

| METHOD (NL) | ACCURACY ↑ | FAIRNESS | | |
| --- | --- | --- | --- | --- |
| | | MEASURE MEAN ABS. ↓ | MAXIMUM | MINIMUM |
| Unconstrained | $0.8165 \pm 0.019$ | **AP** $0.0535 \pm 0.0199$ | $0.0423 \pm 0.0155$ | $-0.0648 \pm 0.0251$ |
| Constant | $0.734 \pm 0.0$ | **AP** $0.272 \pm 0.0$ | $0.377 \pm 0.0$ | $0.168 \pm 0.0$ |
| Weighted ERM | $0.8271 \pm 0.0114$ | **AP** $0.0483 \pm 0.0167$ | $0.0382 \pm 0.0139$ | $-0.0584 \pm 0.02$ |
| Adversarial | $0.809 \pm 0.0175$ | **AP** $0.0592 \pm 0.0173$ | $0.0464 \pm 0.0135$ | $-0.0719 \pm 0.0223$ |
| Reduction | $0.8501 \pm 0.0096$ | **AP** $0.0559 \pm 0.0215$ | $0.0432 \pm 0.0166$ | $-0.0685 \pm 0.0269$ |
| FairGrad | $0.822 \pm 0.0203$ | **AP** $0.0434 \pm 0.0206$ | $0.0341 \pm 0.0162$ | $-0.0526 \pm 0.0252$ |
| Unconstrained | $0.8115 \pm 0.014$ | **Eodds** $0.1635 \pm 0.0395$ | $0.0854 \pm 0.014$ | $-0.3326 \pm 0.0649$ |
| Constant | $0.677 \pm 0.0$ | **Eodds** $0.0 \pm 0.0$ | $0.0 \pm 0.0$ | $0.0 \pm 0.0$ |
| Weighted ERM | $0.8135 \pm 0.0137$ | **Eodds** $0.1739 \pm 0.0394$ | $0.0861 \pm 0.0212$ | $-0.3309 \pm 0.0778$ |
| Adversarial | $0.791 \pm 0.007$ | **Eodds** $0.1464 \pm 0.0168$ | $0.0797 \pm 0.0192$ | $-0.3001 \pm 0.0296$ |
| BiFair | $0.793 \pm 0.022$ | **Eodds** $0.161 \pm 0.032$ | $0.091 \pm 0.025$ | $-0.339 \pm 0.048$ |
| FairBatch | $0.8391 \pm 0.0195$ | **Eodds** $0.189 \pm 0.0368$ | $0.1106 \pm 0.0313$ | $-0.3828 \pm 0.0671$ |
| Reduction | $0.7258 \pm 0.0267$ | **Eodds** $0.0743 \pm 0.0409$ | $0.0553 \pm 0.014$ | $-0.1556 \pm 0.0976$ |
| FairGrad | $0.7734 \pm 0.0251$ | **Eodds** $0.0982 \pm 0.0513$ | $0.0511 \pm 0.0179$ | $-0.2016 \pm 0.0771$ |
| Unconstrained | $0.817 \pm 0.0152$ | **Eopp** $0.1044 \pm 0.0133$ | $0.0856 \pm 0.0123$ | $-0.3321 \pm 0.0489$ |
| Constant | $0.697 \pm 0.0$ | **Eopp** $0.0 \pm 0.0$ | $0.0 \pm 0.0$ | $0.0 \pm 0.0$ |
| Weighted ERM | $0.8205 \pm 0.0184$ | **Eopp** $0.1159 \pm 0.0191$ | $0.0955 \pm 0.019$ | $-0.368 \pm 0.0642$ |
| Adversarial | $0.795 \pm 0.0148$ | **Eopp** $0.0959 \pm 0.0153$ | $0.0802 \pm 0.0227$ | $-0.3036 \pm 0.042$ |
| BiFair | $0.807 \pm 0.025$ | **Eopp** $0.11 \pm 0.031$ | $0.091 \pm 0.031$ | $-0.351 \pm 0.097$ |
| FairBatch | $0.8411 \pm 0.0177$ | **Eopp** $0.1217 \pm 0.0277$ | $0.1083 \pm 0.0311$ | $-0.3784 \pm 0.0891$ |
| Reduction | $0.6887 \pm 0.0271$ | **Eopp** $0.0282 \pm 0.0159$ | $0.034 \pm 0.0281$ | $-0.0788 \pm 0.0619$ |
| FairGrad | $0.7799 \pm 0.0243$ | **Eopp** $0.0675 \pm 0.0179$ | $0.0556 \pm 0.0147$ | $-0.2143 \pm 0.0592$ |

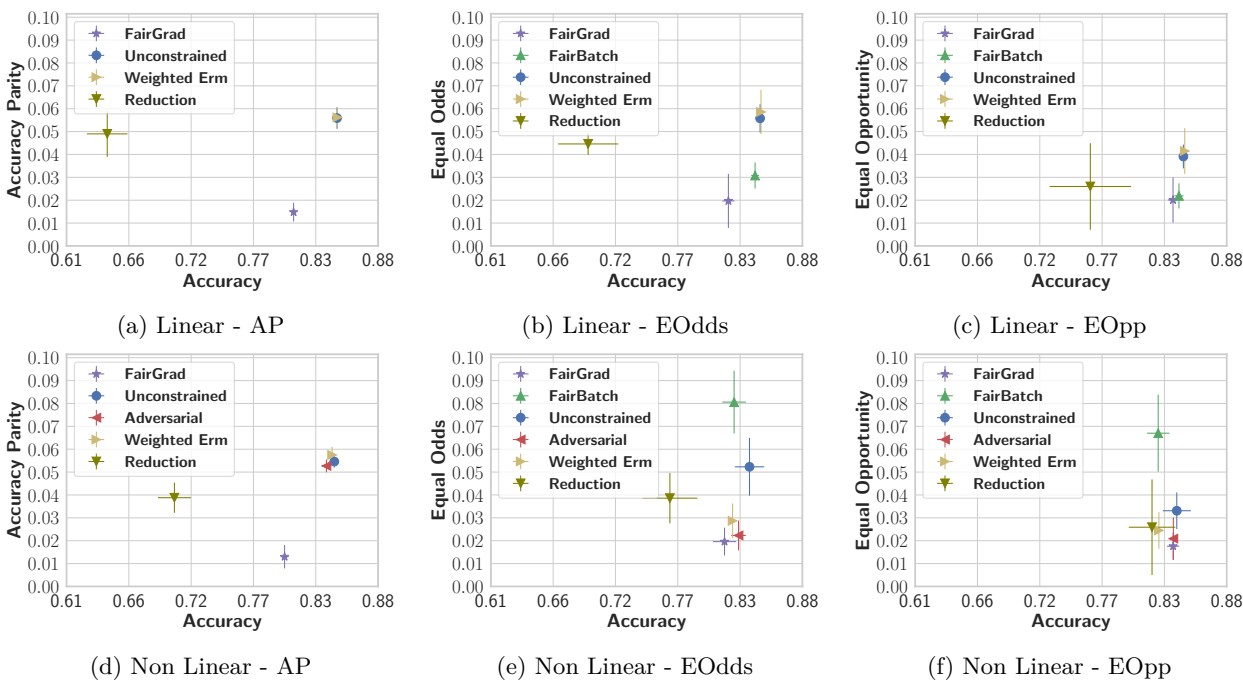

Figure 9: Results for the Adult with multiple groups dataset with different fairness measures.

Table 10: Results for the Adult with multiple groups dataset with Linear Models. All the results are averaged over 5 runs. Here MEAN ABS., MAXIMUM, and MINIMUM represent the mean absolute fairness value, the fairness level of the most well-off group, and the fairness level of the worst-off group, respectively.

| METHOD (L) | ACCURACY ↑ | FAIRNESS | | |
|---|---|---|---|---|
| | | MEASURE MEAN ABS. ↓ | MAXIMUM | MINIMUM |
| Unconstrained | $0.8451 \pm 0.0042$ | **AP** $0.0559 \pm 0.0047$ | $0.0985 \pm 0.0111$ | $-0.042 \pm 0.003$ |
| Constant | $0.754 \pm 0.0$ | **AP** $0.097 \pm 0.0$ | $0.159 \pm 0.0$ | $0.024 \pm 0.0$ |
| Weighted ERM | $0.8454 \pm 0.0032$ | **AP** $0.0562 \pm 0.0042$ | $0.0993 \pm 0.0117$ | $-0.0426 \pm 0.0018$ |
| Reduction | $0.6436 \pm 0.0178$ | **AP** $0.049 \pm 0.01$ | $0.0493 \pm 0.017$ | $-0.0661 \pm 0.0113$ |
| FairGrad | $0.807 \pm 0.0022$ | **AP** $0.0148 \pm 0.0041$ | $0.0256 \pm 0.0048$ | $-0.0107 \pm 0.0045$ |
| Unconstrained | $0.844 \pm 0.0011$ | **Eodds** $0.0558 \pm 0.0062$ | $0.0578 \pm 0.0069$ | $-0.1586 \pm 0.0621$ |
| Constant | $0.75 \pm 0.0$ | **Eodds** $0.0 \pm 0.0$ | $0.0 \pm 0.0$ | $0.0 \pm 0.0$ |
| Weighted ERM | $0.8448 \pm 0.0038$ | **Eodds** $0.0586 \pm 0.0097$ | $0.0567 \pm 0.0048$ | $-0.1702 \pm 0.0776$ |
| FairBatch | $0.8396 \pm 0.0034$ | **Eodds** $0.0308 \pm 0.0057$ | $0.0565 \pm 0.0116$ | $-0.0641 \pm 0.0234$ |
| Reduction | $0.6932 \pm 0.0264$ | **Eodds** $0.0446 \pm 0.0048$ | $0.0806 \pm 0.043$ | $-0.0896 \pm 0.0278$ |
| FairGrad | $0.8162 \pm 0.0052$ | **Eodds** $0.0197 \pm 0.0118$ | $0.0373 \pm 0.0233$ | $-0.0493 \pm 0.0403$ |
| Unconstrained | $0.8431 \pm 0.002$ | **Eopp** $0.0391 \pm 0.0052$ | $0.0297 \pm 0.0131$ | $-0.169 \pm 0.0565$ |
| Constant | $0.762 \pm 0.0$ | **Eopp** $0.0 \pm 0.0$ | $0.0 \pm 0.0$ | $0.0 \pm 0.0$ |
| Weighted ERM | $0.8443 \pm 0.0038$ | **Eopp** $0.0415 \pm 0.01$ | $0.0316 \pm 0.0145$ | $-0.1767 \pm 0.0797$ |
| FairBatch | $0.8392 \pm 0.004$ | **Eopp** $0.0219 \pm 0.0055$ | $0.05 \pm 0.0133$ | $-0.0749 \pm 0.0285$ |
| Reduction | $0.7615 \pm 0.0357$ | **Eopp** $0.026 \pm 0.0189$ | $0.0487 \pm 0.0378$ | $-0.1115 \pm 0.0867$ |
| FairGrad | $0.834 \pm 0.0044$ | **Eopp** $0.0201 \pm 0.0099$ | $0.0442 \pm 0.0415$ | $-0.0679 \pm 0.0808$ |

Table 11: Results for the Adult with multiple groups dataset with Non Linear Models. All the results are averaged over 5 runs. Here MEAN ABS., MAXIMUM, and MINIMUM represent the mean absolute fairness value, the fairness level of the most well-off group, and the fairness level of the worst-off group, respectively.

| METHOD (NL) | ACCURACY ↑ | FAIRNESS | | |
|---|---|---|---|---|
| | | MEASURE | MEAN ABS. ↓ | MAXIMUM | MINIMUM |
| Unconstrained | $0.8427 \pm 0.0041$ | **AP** | $0.0546 \pm 0.0026$ | $0.0966 \pm 0.0098$ | $-0.0421 \pm 0.0022$ |
| Constant | $0.754 \pm 0.0$ | **AP** | $0.097 \pm 0.0$ | $0.159 \pm 0.0$ | $0.024 \pm 0.0$ |
| Weighted ERM | $0.8408 \pm 0.0031$ | **AP** | $0.0575 \pm 0.0035$ | $0.101 \pm 0.0106$ | $-0.0443 \pm 0.0026$ |
| Adversarial | $0.8358 \pm 0.0043$ | **AP** | $0.0527 \pm 0.0028$ | $0.0889 \pm 0.0066$ | $-0.0401 \pm 0.0022$ |
| Reduction | $0.7025 \pm 0.0144$ | **AP** | $0.0388 \pm 0.0066$ | $0.054 \pm 0.0151$ | $-0.0525 \pm 0.0099$ |
| FairGrad | $0.7991 \pm 0.0036$ | **AP** | $0.013 \pm 0.0051$ | $0.0257 \pm 0.0138$ | $-0.0125 \pm 0.0043$ |
| Unconstrained | $0.8347 \pm 0.0129$ | **Eodds** | $0.0523 \pm 0.0126$ | $0.0495 \pm 0.0166$ | $-0.1772 \pm 0.0512$ |
| Constant | $0.75 \pm 0.0$ | **Eodds** | $0.0 \pm 0.0$ | $0.0 \pm 0.0$ | $0.0 \pm 0.0$ |
| Weighted ERM | $0.8199 \pm 0.002$ | **Eodds** | $0.0287 \pm 0.0076$ | $0.0274 \pm 0.0177$ | $-0.1013 \pm 0.0543$ |
| Adversarial | $0.8251 \pm 0.0064$ | **Eodds** | $0.0223 \pm 0.0065$ | $0.0451 \pm 0.0308$ | $-0.0667 \pm 0.0559$ |
| FairBatch | $0.8212 \pm 0.0103$ | **Eodds** | $0.0806 \pm 0.0137$ | $0.0522 \pm 0.0076$ | $-0.2545 \pm 0.0525$ |
| Reduction | $0.7649 \pm 0.0241$ | **Eodds** | $0.0386 \pm 0.011$ | $0.044 \pm 0.02$ | $-0.0954 \pm 0.0465$ |
| FairGrad | $0.8128 \pm 0.0102$ | **Eodds** | $0.0196 \pm 0.0061$ | $0.0392 \pm 0.0176$ | $-0.0443 \pm 0.0342$ |
| Unconstrained | $0.8373 \pm 0.0123$ | **Eopp** | $0.0331 \pm 0.008$ | $0.0183 \pm 0.0045$ | $-0.1587 \pm 0.0643$ |
| Constant | $0.762 \pm 0.0$ | **Eopp** | $0.0 \pm 0.0$ | $0.0 \pm 0.0$ | $0.0 \pm 0.0$ |
| Weighted ERM | $0.8216 \pm 0.0031$ | **Eopp** | $0.0245 \pm 0.008$ | $0.0243 \pm 0.0196$ | $-0.1016 \pm 0.0543$ |
| Adversarial | $0.8343 \pm 0.0036$ | **Eopp** | $0.0209 \pm 0.0093$ | $0.0327 \pm 0.013$ | $-0.0927 \pm 0.0589$ |
| FairBatch | $0.821 \pm 0.0097$ | **Eopp** | $0.067 \pm 0.0168$ | $0.047 \pm 0.0113$ | $-0.2484 \pm 0.0535$ |
| Reduction | $0.8156 \pm 0.0204$ | **Eopp** | $0.0259 \pm 0.0209$ | $0.0472 \pm 0.0325$ | $-0.0968 \pm 0.1117$ |
| FairGrad | $0.8341 \pm 0.0053$ | **Eopp** | $0.0176 \pm 0.0059$ | $0.0302 \pm 0.0272$ | $-0.0731 \pm 0.0543$ |

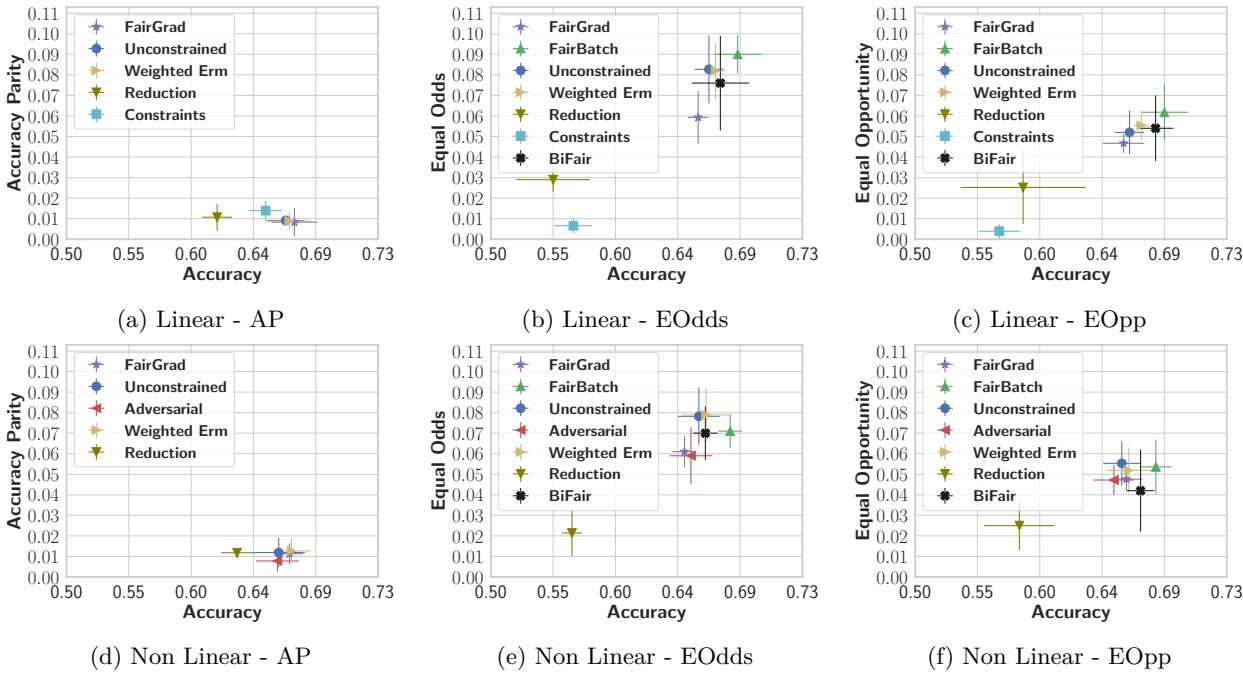

Figure 10: Results for the Compas dataset with different fairness measures.

Table 12: Results for the Compas dataset with Linear Models. All the results are averaged over 5 runs. Here MEAN ABS., MAXIMUM, and MINIMUM represent the mean absolute fairness value, the fairness level of the most well-off group, and the fairness level of the worst-off group, respectively.

| METHOD (L) | ACCURACY ↑ | FAIRNESS | | |
| --- | --- | --- | --- | --- |
| | | MEASURE | MEAN ABS. ↓ | MAXIMUM | MINIMUM |
| Unconstrained | $0.6644 \pm 0.0137$ | **AP** | $0.0091 \pm 0.0025$ | $0.0076 \pm 0.0031$ | $-0.0107 \pm 0.004$ |
| Constant | $0.545 \pm 0.0$ | **AP** | $0.066 \pm 0.0$ | $0.085 \pm 0.0$ | $0.047 \pm 0.0$ |
| Weighted ERM | $0.6671 \pm 0.0169$ | **AP** | $0.0088 \pm 0.004$ | $0.0061 \pm 0.0028$ | $-0.0115 \pm 0.0051$ |
| Constrained | $0.65 \pm 0.012$ | **AP** | $0.014 \pm 0.005$ | $0.018 \pm 0.006$ | $0.009 \pm 0.003$ |
| Reduction | $0.6141 \pm 0.011$ | **AP** | $0.0107 \pm 0.0064$ | $0.009 \pm 0.006$ | $-0.0124 \pm 0.0086$ |
| FairGrad | $0.6708 \pm 0.0166$ | **AP** | $0.0083 \pm 0.0068$ | $0.0057 \pm 0.0048$ | $-0.0108 \pm 0.0088$ |
| Unconstrained | $0.6636 \pm 0.0104$ | **Eodds** | $0.0827 \pm 0.0165$ | $0.0758 \pm 0.0133$ | $-0.1553 \pm 0.0259$ |
| Constant | $0.527 \pm 0.0$ | **Eodds** | $0.0 \pm 0.0$ | $0.0 \pm 0.0$ | $0.0 \pm 0.0$ |
| Weighted ERM | $0.6685 \pm 0.0073$ | **Eodds** | $0.082 \pm 0.0137$ | $0.0697 \pm 0.0115$ | $-0.1618 \pm 0.0222$ |
| Constrained | $0.564 \pm 0.014$ | **Eodds** | $0.007 \pm 0.004$ | $0.014 \pm 0.011$ | $0.002 \pm 0.001$ |
| BiFair | $0.672 \pm 0.021$ | **Eodds** | $0.076 \pm 0.023$ | $0.071 \pm 0.025$ | $-0.15 \pm 0.039$ |
| FairBatch | $0.6847 \pm 0.0175$ | **Eodds** | $0.09 \pm 0.0094$ | $0.0854 \pm 0.0149$ | $-0.1727 \pm 0.0304$ |
| Reduction | $0.5493 \pm 0.027$ | **Eodds** | $0.029 \pm 0.0058$ | $0.0268 \pm 0.0062$ | $-0.0622 \pm 0.0219$ |
| FairGrad | $0.6557 \pm 0.0075$ | **Eodds** | $0.0593 \pm 0.0128$ | $0.0524 \pm 0.0102$ | $-0.1241 \pm 0.0202$ |
| Unconstrained | $0.6609 \pm 0.0106$ | **Eopp** | $0.052 \pm 0.0107$ | $0.062 \pm 0.0145$ | $-0.1461 \pm 0.0286$ |
| Constant | $0.55 \pm 0.0$ | **Eopp** | $0.0 \pm 0.0$ | $0.0 \pm 0.0$ | $0.0 \pm 0.0$ |
| Weighted ERM | $0.6695 \pm 0.0055$ | **Eopp** | $0.0554 \pm 0.0074$ | $0.0659 \pm 0.0107$ | $-0.1557 \pm 0.0194$ |
| Constrained | $0.565 \pm 0.015$ | **Eopp** | $0.004 \pm 0.003$ | $0.011 \pm 0.009$ | $0.0 \pm 0.0$ |
| BiFair | $0.68 \pm 0.013$ | **Eopp** | $0.054 \pm 0.016$ | $0.064 \pm 0.022$ | $-0.15 \pm 0.044$ |
| FairBatch | $0.6865 \pm 0.0171$ | **Eopp** | $0.0618 \pm 0.0134$ | $0.0715 \pm 0.0173$ | $-0.1755 \pm 0.0364$ |
| Reduction | $0.5828 \pm 0.0457$ | **Eopp** | $0.0252 \pm 0.0178$ | $0.03 \pm 0.0216$ | $-0.0707 \pm 0.0498$ |
| FairGrad | $0.6565 \pm 0.0152$ | **Eopp** | $0.0467 \pm 0.0046$ | $0.0554 \pm 0.0071$ | $-0.1313 \pm 0.0119$ |

Table 13: Results for the Compas dataset with Non Linear Models. All the results are averaged over 5 runs. Here MEAN ABS., MAXIMUM, and MINIMUM represent the mean absolute fairness value, the fairness level of the most well-off group, and the fairness level of the worst-off group, respectively.

| METHOD (NL) | ACCURACY ↑ | FAIRNESS | | | |
|---|---|---|---|---|---|
| | | MEASURE | MEAN ABS. ↓ | MAXIMUM | MINIMUM |
| Unconstrained | 0.6593 ± 0.0192 | **AP** | 0.0119 ± 0.0072 | 0.0095 ± 0.004 | -0.0144 ± 0.0107 |
| Constant | 0.545 ± 0.0 | **AP** | 0.066 ± 0.0 | 0.085 ± 0.0 | 0.047 ± 0.0 |
| Weighted ERM | 0.6687 ± 0.0138 | **AP** | 0.0127 ± 0.0061 | 0.011 ± 0.0034 | -0.0145 ± 0.0099 |
| Adversarial | 0.6583 ± 0.0157 | **AP** | 0.0078 ± 0.0051 | 0.0066 ± 0.0044 | -0.009 ± 0.0069 |
| Reduction | 0.6287 ± 0.0117 | **AP** | 0.0118 ± 0.0024 | 0.0103 ± 0.0062 | -0.0134 ± 0.0024 |
| FairGrad | 0.6672 ± 0.0099 | **AP** | 0.0113 ± 0.005 | 0.0095 ± 0.0023 | -0.0131 ± 0.0082 |
| Unconstrained | 0.6562 ± 0.0154 | **Eodds** | 0.0782 ± 0.014 | 0.0715 ± 0.0136 | -0.1521 ± 0.0277 |
| Constant | 0.527 ± 0.0 | **Eodds** | 0.0 ± 0.0 | 0.0 ± 0.0 | 0.0 ± 0.0 |
| Weighted ERM | 0.6615 ± 0.0175 | **Eodds** | 0.0789 ± 0.0131 | 0.0726 ± 0.0077 | -0.1496 ± 0.0313 |
| Adversarial | 0.6504 ± 0.0157 | **Eodds** | 0.059 ± 0.0138 | 0.0549 ± 0.0107 | -0.1294 ± 0.0183 |
| BiFair | 0.661 ± 0.009 | **Eodds** | 0.07 ± 0.013 | 0.068 ± 0.018 | -0.133 ± 0.016 |
| FairBatch | 0.6792 ± 0.0086 | **Eodds** | 0.071 ± 0.0083 | 0.0663 ± 0.0091 | -0.1508 ± 0.0304 |
| Reduction | 0.5631 ± 0.0072 | **Eodds** | 0.0214 ± 0.0112 | 0.024 ± 0.0102 | -0.0489 ± 0.0363 |
| FairGrad | 0.6457 ± 0.0088 | **Eodds** | 0.061 ± 0.0075 | 0.0564 ± 0.0065 | -0.127 ± 0.0081 |
| Unconstrained | 0.6552 ± 0.0137 | **Eopp** | 0.0553 ± 0.0108 | 0.0659 ± 0.015 | -0.1552 ± 0.0281 |
| Constant | 0.55 ± 0.0 | **Eopp** | 0.0 ± 0.0 | 0.0 ± 0.0 | 0.0 ± 0.0 |
| Weighted ERM | 0.6604 ± 0.0163 | **Eopp** | 0.0519 ± 0.0111 | 0.0618 ± 0.0148 | -0.1458 ± 0.0299 |
| Adversarial | 0.6494 ± 0.0148 | **Eopp** | 0.0472 ± 0.0072 | 0.0563 ± 0.0108 | -0.1327 ± 0.0183 |
| BiFair | 0.669 ± 0.01 | **Eopp** | 0.042 ± 0.02 | 0.05 ± 0.025 | -0.117 ± 0.055 |
| FairBatch | 0.6802 ± 0.0114 | **Eopp** | 0.0536 ± 0.0133 | 0.062 ± 0.0167 | -0.1526 ± 0.0367 |
| Reduction | 0.5801 ± 0.0258 | **Eopp** | 0.025 ± 0.0119 | 0.0296 ± 0.0145 | -0.0702 ± 0.0333 |
| FairGrad | 0.6586 ± 0.0118 | **Eopp** | 0.0476 ± 0.0056 | 0.0563 ± 0.0067 | -0.1339 ± 0.0163 |

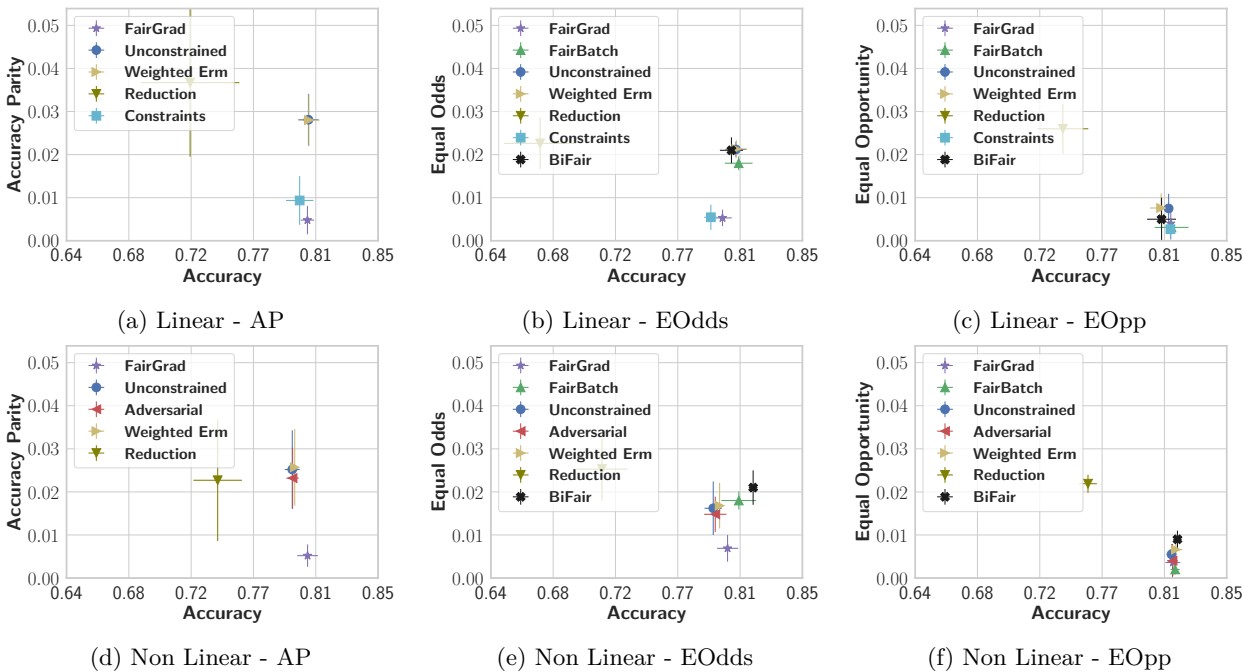

Figure 11: Results for the Dutch dataset with different fairness measures.

Table 14: Results for the Dutch dataset with Linear Models. All the results are averaged over 5 runs. Here MEAN ABS., MAXIMUM, and MINIMUM represent the mean absolute fairness value, the fairness level of the most well-off group, and the fairness level of the worst-off group, respectively.

| METHOD (L) | ACCURACY ↑ | FAIRNESS | | | |
|---|---|---|---|---|---|
| | | MEASURE | MEAN ABS. ↓ | MAXIMUM | MINIMUM |
| Unconstrained | $0.8049 \pm 0.007$ | **AP** | $0.0281 \pm 0.006$ | $0.0281 \pm 0.006$ | $-0.0282 \pm 0.0061$ |
| Constant | $0.524 \pm 0.0$ | **AP** | $0.151 \pm 0.0$ | $0.152 \pm 0.0$ | $0.15 \pm 0.0$ |
| Weighted ERM | $0.8052 \pm 0.0073$ | **AP** | $0.028 \pm 0.006$ | $0.028 \pm 0.006$ | $-0.0281 \pm 0.006$ |
| Constrained | $0.799 \pm 0.009$ | **AP** | $0.009 \pm 0.006$ | $0.009 \pm 0.006$ | $0.009 \pm 0.006$ |
| Reduction | $0.723 \pm 0.0341$ | **AP** | $0.0367 \pm 0.0172$ | $0.0368 \pm 0.0172$ | $-0.0367 \pm 0.0172$ |
| FairGrad | $0.8042 \pm 0.0046$ | **AP** | $0.0048 \pm 0.0033$ | $0.0048 \pm 0.0033$ | $-0.0048 \pm 0.0032$ |
| Unconstrained | $0.8071 \pm 0.0072$ | **Eodds** | $0.0212 \pm 0.0018$ | $0.0322 \pm 0.009$ | $-0.0256 \pm 0.0052$ |
| Constant | $0.522 \pm 0.0$ | **Eodds** | $0.0 \pm 0.0$ | $0.0 \pm 0.0$ | $0.0 \pm 0.0$ |
| Weighted ERM | $0.8074 \pm 0.0074$ | **Eodds** | $0.0213 \pm 0.002$ | $0.032 \pm 0.0086$ | $-0.0254 \pm 0.0051$ |
| Constrained | $0.79 \pm 0.005$ | **Eodds** | $0.005 \pm 0.003$ | $0.009 \pm 0.005$ | $0.002 \pm 0.002$ |
| BiFair | $0.804 \pm 0.008$ | **Eodds** | $0.021 \pm 0.003$ | $0.025 \pm 0.004$ | $-0.033 \pm 0.01$ |
| FairBatch | $0.809 \pm 0.0096$ | **Eodds** | $0.018 \pm 0.0016$ | $0.0262 \pm 0.0039$ | $-0.0211 \pm 0.004$ |
| Reduction | $0.6716 \pm 0.0251$ | **Eodds** | $0.0226 \pm 0.006$ | $0.0333 \pm 0.0107$ | $-0.0404 \pm 0.0213$ |
| FairGrad | $0.7978 \pm 0.0064$ | **Eodds** | $0.0053 \pm 0.0019$ | $0.007 \pm 0.0019$ | $-0.009 \pm 0.0049$ |
| Unconstrained | $0.8129 \pm 0.0021$ | **Eopp** | $0.0075 \pm 0.0034$ | $0.0107 \pm 0.0049$ | $-0.0193 \pm 0.0086$ |
| Constant | $0.524 \pm 0.0$ | **Eopp** | $0.0 \pm 0.0$ | $0.0 \pm 0.0$ | $0.0 \pm 0.0$ |
| Weighted ERM | $0.8077 \pm 0.0078$ | **Eopp** | $0.0076 \pm 0.0034$ | $0.011 \pm 0.0049$ | $-0.0196 \pm 0.0087$ |
| Constrained | $0.814 \pm 0.003$ | **Eopp** | $0.003 \pm 0.002$ | $0.007 \pm 0.006$ | $0.0 \pm 0.0$ |
| BiFair | $0.808 \pm 0.01$ | **Eopp** | $0.005 \pm 0.005$ | $0.008 \pm 0.007$ | $-0.012 \pm 0.012$ |
| FairBatch | $0.8149 \pm 0.0117$ | **Eopp** | $0.0031 \pm 0.0014$ | $0.0044 \pm 0.002$ | $-0.0079 \pm 0.0036$ |
| Reduction | $0.7397 \pm 0.0176$ | **Eopp** | $0.026 \pm 0.0058$ | $0.0669 \pm 0.0149$ | $-0.0372 \pm 0.0083$ |
| FairGrad | $0.8144 \pm 0.0021$ | **Eopp** | $0.004 \pm 0.0037$ | $0.006 \pm 0.0052$ | $-0.0099 \pm 0.0097$ |

Table 15: Results for the Dutch dataset with Non Linear Models. All the results are averaged over 5 runs. Here MEAN ABS., MAXIMUM, and MINIMUM represent the mean absolute fairness value, the fairness level of the most well-off group, and the fairness level of the worst-off group, respectively.

| METHOD (NL) | ACCURACY ↑ | FAIRNESS | | |
| --- | --- | --- | --- | --- |
| | | MEASURE | MEAN ABS. ↓ | MAXIMUM | MINIMUM |
| Unconstrained | 0.7937 ± 0.0052 | **AP** | 0.0252 ± 0.0091 | 0.0252 ± 0.009 | -0.0252 ± 0.0091 |
| Constant | 0.524 ± 0.0 | **AP** | 0.151 ± 0.0 | 0.152 ± 0.0 | 0.15 ± 0.0 |
| Weighted ERM | 0.7954 ± 0.0023 | **AP** | 0.0257 ± 0.0089 | 0.0257 ± 0.0089 | -0.0257 ± 0.0089 |
| Adversarial | 0.7939 ± 0.0043 | **AP** | 0.0232 ± 0.0071 | 0.0232 ± 0.0071 | -0.0232 ± 0.007 |
| Reduction | 0.7421 ± 0.0168 | **AP** | 0.0227 ± 0.0141 | 0.0227 ± 0.0142 | -0.0227 ± 0.0141 |
| FairGrad | 0.8043 ± 0.0071 | **AP** | 0.0052 ± 0.0026 | 0.0052 ± 0.0026 | -0.0052 ± 0.0026 |
| Unconstrained | 0.7914 ± 0.006 | **Eodds** | 0.0162 ± 0.0062 | 0.0193 ± 0.0071 | -0.0263 ± 0.0142 |
| Constant | 0.522 ± 0.0 | **Eodds** | 0.0 ± 0.0 | 0.0 ± 0.0 | 0.0 ± 0.0 |
| Weighted ERM | 0.7958 ± 0.0027 | **Eodds** | 0.0168 ± 0.0053 | 0.0202 ± 0.0048 | -0.0261 ± 0.0131 |
| Adversarial | 0.7928 ± 0.0077 | **Eodds** | 0.0148 ± 0.0041 | 0.0202 ± 0.0066 | -0.0211 ± 0.006 |
| BiFair | 0.819 ± 0.003 | **Eodds** | 0.021 ± 0.004 | 0.03 ± 0.005 | -0.028 ± 0.007 |
| FairBatch | 0.8091 ± 0.012 | **Eodds** | 0.018 ± 0.0021 | 0.0254 ± 0.0058 | -0.0248 ± 0.0062 |
| Reduction | 0.7144 ± 0.0176 | **Eodds** | 0.0253 ± 0.0073 | 0.0347 ± 0.0123 | -0.0323 ± 0.0064 |
| FairGrad | 0.8013 ± 0.0073 | **Eodds** | 0.0069 ± 0.0031 | 0.0099 ± 0.0038 | -0.0095 ± 0.0068 |
| Unconstrained | 0.8149 ± 0.0034 | **Eopp** | 0.0055 ± 0.0024 | 0.0079 ± 0.0035 | -0.014 ± 0.0061 |
| Constant | 0.524 ± 0.0 | **Eopp** | 0.0 ± 0.0 | 0.0 ± 0.0 | 0.0 ± 0.0 |
| Weighted ERM | 0.8179 ± 0.0044 | **Eopp** | 0.0066 ± 0.0026 | 0.0095 ± 0.0037 | -0.017 ± 0.0065 |
| Adversarial | 0.8156 ± 0.0038 | **Eopp** | 0.004 ± 0.0039 | 0.0058 ± 0.0057 | -0.0102 ± 0.01 |
| BiFair | 0.819 ± 0.003 | **Eopp** | 0.009 ± 0.002 | 0.012 ± 0.003 | -0.022 ± 0.006 |
| FairBatch | 0.8174 ± 0.0031 | **Eopp** | 0.002 ± 0.0012 | 0.0029 ± 0.0017 | -0.0052 ± 0.0031 |
| Reduction | 0.7571 ± 0.0061 | **Eopp** | 0.0219 ± 0.0021 | 0.0563 ± 0.0054 | -0.0313 ± 0.0028 |
| FairGrad | 0.8158 ± 0.0051 | **Eopp** | 0.0036 ± 0.0031 | 0.0051 ± 0.0045 | -0.0092 ± 0.0079 |

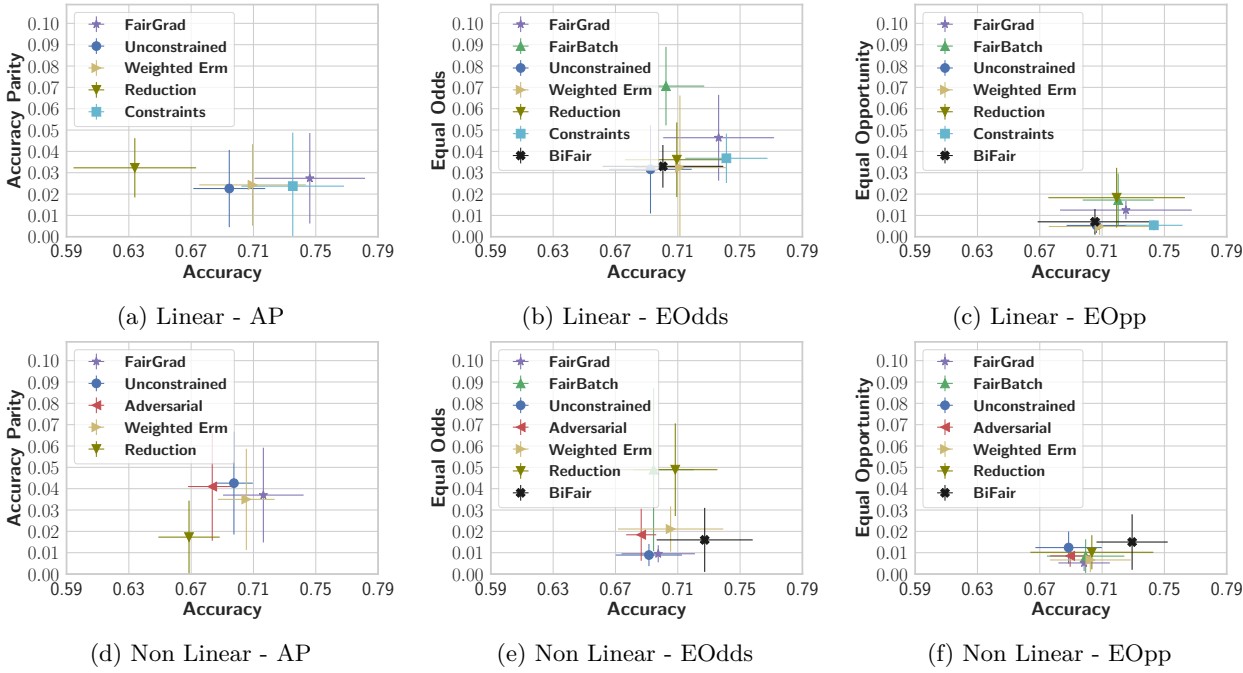

(a) Linear - AP  (b) Linear - EOdds  (c) Linear - EOpp

(d) Non Linear - AP  (e) Non Linear - EOdds  (f) Non Linear - EOpp

Figure 12: Results for the German dataset with different fairness measures.

Table 16: Results for the German dataset with Linear Models. All the results are averaged over 5 runs. Here MEAN ABS., MAXIMUM, and MINIMUM represent the mean absolute fairness value, the fairness level of the most well-off group, and the fairness level of the worst-off group, respectively.

| METHOD (L) | ACCURACY ↑ | FAIRNESS | | | |
|---|---|---|---|---|---|
| | | MEASURE | MEAN ABS. ↓ | MAXIMUM | MINIMUM |
| Unconstrained | $0.692 \pm 0.0232$ | **AP** | $0.0226 \pm 0.0181$ | $0.0169 \pm 0.0111$ | $-0.0284 \pm 0.0256$ |
| Constant | $0.73 \pm 0.0$ | **AP** | $0.05 \pm 0.0$ | $0.069 \pm 0.0$ | $0.031 \pm 0.0$ |
| Weighted ERM | $0.707 \pm 0.0344$ | **AP** | $0.0243 \pm 0.0191$ | $0.0186 \pm 0.0113$ | $-0.0299 \pm 0.027$ |
| Constrained | $0.733 \pm 0.033$ | **AP** | $0.024 \pm 0.025$ | $0.032 \pm 0.033$ | $0.015 \pm 0.017$ |
| Reduction | $0.631 \pm 0.0396$ | **AP** | $0.0323 \pm 0.0139$ | $0.0286 \pm 0.0202$ | $-0.036 \pm 0.0185$ |
| FairGrad | $0.744 \pm 0.0357$ | **AP** | $0.0274 \pm 0.0212$ | $0.0215 \pm 0.0123$ | $-0.0334 \pm 0.0306$ |
| Unconstrained | $0.69 \pm 0.0266$ | **Eodds** | $0.0316 \pm 0.0207$ | $0.0499 \pm 0.0341$ | $-0.0618 \pm 0.0471$ |
| Constant | $0.7 \pm 0.0$ | **Eodds** | $0.0 \pm 0.0$ | $0.0 \pm 0.0$ | $0.0 \pm 0.0$ |
| Weighted ERM | $0.709 \pm 0.0296$ | **Eodds** | $0.0324 \pm 0.0338$ | $0.0461 \pm 0.046$ | $-0.055 \pm 0.0626$ |
| Constrained | $0.739 \pm 0.027$ | **Eodds** | $0.037 \pm 0.012$ | $0.072 \pm 0.025$ | $0.01 \pm 0.004$ |
| BiFair | $0.698 \pm 0.039$ | **Eodds** | $0.033 \pm 0.01$ | $0.052 \pm 0.023$ | $-0.059 \pm 0.029$ |
| FairBatch | $0.7 \pm 0.0247$ | **Eodds** | $0.0706 \pm 0.0184$ | $0.1102 \pm 0.0489$ | $-0.1134 \pm 0.0518$ |
| Reduction | $0.707 \pm 0.0335$ | **Eodds** | $0.0361 \pm 0.0175$ | $0.0716 \pm 0.056$ | $-0.0576 \pm 0.0266$ |
| FairGrad | $0.734 \pm 0.0358$ | **Eodds** | $0.0464 \pm 0.0201$ | $0.0784 \pm 0.0232$ | $-0.0721 \pm 0.0496$ |
| Unconstrained | $0.704 \pm 0.0193$ | **Eopp** | $0.0053 \pm 0.0035$ | $0.0096 \pm 0.004$ | $-0.0116 \pm 0.0117$ |
| Constant | $0.7 \pm 0.0$ | **Eopp** | $0.0 \pm 0.0$ | $0.0 \pm 0.0$ | $0.0 \pm 0.0$ |
| Weighted ERM | $0.706 \pm 0.0328$ | **Eopp** | $0.0048 \pm 0.0039$ | $0.0097 \pm 0.0091$ | $-0.0096 \pm 0.0092$ |
| Constrained | $0.741 \pm 0.019$ | **Eopp** | $0.005 \pm 0.002$ | $0.015 \pm 0.006$ | $0.0 \pm 0.0$ |
| BiFair | $0.703 \pm 0.037$ | **Eopp** | $0.007 \pm 0.006$ | $0.014 \pm 0.015$ | $-0.013 \pm 0.015$ |
| FairBatch | $0.718 \pm 0.0229$ | **Eopp** | $0.0172 \pm 0.0124$ | $0.0272 \pm 0.0187$ | $-0.0416 \pm 0.0396$ |
| Reduction | $0.717 \pm 0.0441$ | **Eopp** | $0.0183 \pm 0.014$ | $0.036 \pm 0.0254$ | $-0.0372 \pm 0.0407$ |
| FairGrad | $0.723 \pm 0.0425$ | **Eopp** | $0.0125 \pm 0.0043$ | $0.0212 \pm 0.0087$ | $-0.0288 \pm 0.0162$ |

Table 17: Results for the German dataset with Non Linear Models. All the results are averaged over 5 runs. Here MEAN ABS., MAXIMUM, and MINIMUM represent the mean absolute fairness value, the fairness level of the most well-off group, and the fairness level of the worst-off group, respectively.

| METHOD (NL) | ACCURACY ↑ | FAIRNESS | | | |
|---|---|---|---|---|---|
| | | MEASURE | MEAN ABS. ↓ | MAXIMUM | MINIMUM |
| Unconstrained | $0.695 \pm 0.0122$ | **AP** | $0.0426 \pm 0.0241$ | $0.0314 \pm 0.0144$ | $-0.0537 \pm 0.0345$ |
| Constant | $0.73 \pm 0.0$ | **AP** | $0.05 \pm 0.0$ | $0.069 \pm 0.0$ | $0.031 \pm 0.0$ |
| Weighted ERM | $0.703 \pm 0.0183$ | **AP** | $0.035 \pm 0.0237$ | $0.0265 \pm 0.0138$ | $-0.0436 \pm 0.0338$ |
| Adversarial | $0.681 \pm 0.0156$ | **AP** | $0.041 \pm 0.0254$ | $0.0327 \pm 0.0165$ | $-0.0492 \pm 0.0368$ |
| Reduction | $0.666 \pm 0.0198$ | **AP** | $0.0173 \pm 0.0171$ | $0.0131 \pm 0.0115$ | $-0.0215 \pm 0.0231$ |
| FairGrad | $0.714 \pm 0.026$ | **AP** | $0.037 \pm 0.0222$ | $0.0291 \pm 0.0119$ | $-0.0448 \pm 0.0331$ |
| Unconstrained | $0.689 \pm 0.0213$ | **Eodds** | $0.0089 \pm 0.0052$ | $0.0117 \pm 0.0045$ | $-0.0144 \pm 0.0116$ |
| Constant | $0.7 \pm 0.0$ | **Eodds** | $0.0 \pm 0.0$ | $0.0 \pm 0.0$ | $0.0 \pm 0.0$ |
| Weighted ERM | $0.703 \pm 0.034$ | **Eodds** | $0.0211 \pm 0.0106$ | $0.0305 \pm 0.0186$ | $-0.0372 \pm 0.0158$ |
| Adversarial | $0.684 \pm 0.0097$ | **Eodds** | $0.0184 \pm 0.0122$ | $0.0263 \pm 0.0201$ | $-0.0339 \pm 0.0237$ |
| BiFair | $0.725 \pm 0.031$ | **Eodds** | $0.016 \pm 0.015$ | $0.021 \pm 0.018$ | $-0.027 \pm 0.018$ |
| FairBatch | $0.692 \pm 0.026$ | **Eodds** | $0.0489 \pm 0.0382$ | $0.0607 \pm 0.0446$ | $-0.0882 \pm 0.0983$ |
| Reduction | $0.706 \pm 0.0272$ | **Eodds** | $0.0489 \pm 0.0217$ | $0.0742 \pm 0.0266$ | $-0.0717 \pm 0.051$ |
| FairGrad | $0.695 \pm 0.0237$ | **Eodds** | $0.0095 \pm 0.004$ | $0.0121 \pm 0.0046$ | $-0.0175 \pm 0.0076$ |
| Unconstrained | $0.686 \pm 0.0215$ | **Eopp** | $0.0124 \pm 0.0075$ | $0.0227 \pm 0.0128$ | $-0.0269 \pm 0.0227$ |
| Constant | $0.7 \pm 0.0$ | **Eopp** | $0.0 \pm 0.0$ | $0.0 \pm 0.0$ | $0.0 \pm 0.0$ |
| Weighted ERM | $0.7 \pm 0.0261$ | **Eopp** | $0.0066 \pm 0.0057$ | $0.0131 \pm 0.0071$ | $-0.0133 \pm 0.0173$ |
| Adversarial | $0.687 \pm 0.0129$ | **Eopp** | $0.0085 \pm 0.0051$ | $0.0203 \pm 0.0147$ | $-0.0137 \pm 0.0099$ |
| BiFair | $0.727 \pm 0.023$ | **Eopp** | $0.015 \pm 0.013$ | $0.023 \pm 0.019$ | $-0.036 \pm 0.038$ |
| FairBatch | $0.697 \pm 0.025$ | **Eopp** | $0.0084 \pm 0.0079$ | $0.0235 \pm 0.0226$ | $-0.0102 \pm 0.0094$ |
| Reduction | $0.701 \pm 0.0397$ | **Eopp** | $0.0102 \pm 0.008$ | $0.0242 \pm 0.024$ | $-0.0167 \pm 0.0134$ |
| FairGrad | $0.696 \pm 0.0166$ | **Eopp** | $0.0052 \pm 0.0038$ | $0.0093 \pm 0.0064$ | $-0.0115 \pm 0.0108$ |

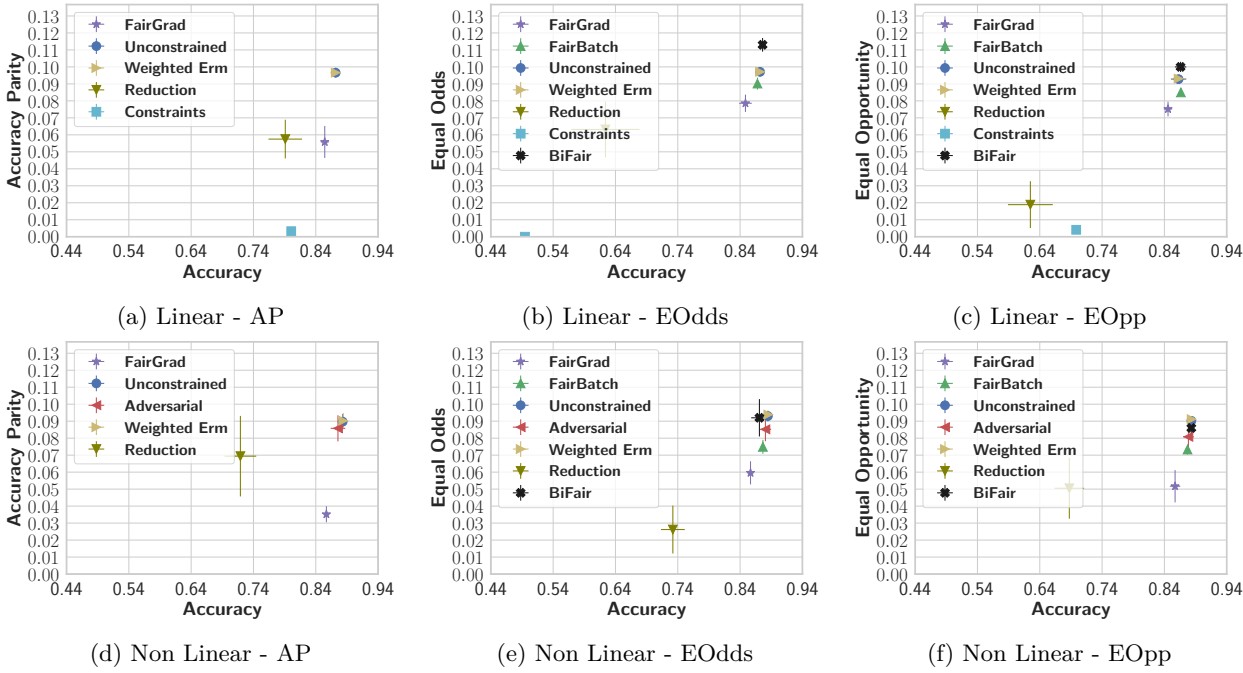

Figure 13: Results for the Gaussian dataset with different fairness measures.

Table 18: Results for the Gaussian dataset with Linear Models. All the results are averaged over 5 runs. Here MEAN ABS., MAXIMUM, and MINIMUM represent the mean absolute fairness value, the fairness level of the most well-off group, and the fairness level of the worst-off group, respectively.

| METHOD (L) | ACCURACY ↑ | FAIRNESS | | | |
|---|---|---|---|---|---|
| | | MEASURE | MEAN ABS. ↓ | MAXIMUM | MINIMUM |
| Unconstrained | $0.8689 \pm 0.0037$ | **AP** | $0.0966 \pm 0.0029$ | $0.0957 \pm 0.0028$ | $-0.0974 \pm 0.0036$ |
| Constant | $0.497 \pm 0.0$ | **AP** | $0.001 \pm 0.0$ | $0.001 \pm 0.0$ | $0.001 \pm 0.0$ |
| Weighted ERM | $0.869 \pm 0.0039$ | **AP** | $0.0966 \pm 0.0026$ | $0.0957 \pm 0.0023$ | $-0.0974 \pm 0.0034$ |
| Constrained | $0.799 \pm 0.004$ | **AP** | $0.003 \pm 0.002$ | $0.003 \pm 0.002$ | $0.003 \pm 0.002$ |
| Reduction | $0.7891 \pm 0.0266$ | **AP** | $0.0575 \pm 0.0114$ | $0.057 \pm 0.0118$ | $-0.0579 \pm 0.0111$ |
| FairGrad | $0.8516 \pm 0.0064$ | **AP** | $0.0558 \pm 0.0094$ | $0.0553 \pm 0.0093$ | $-0.0562 \pm 0.0096$ |
| Unconstrained | $0.869 \pm 0.0037$ | **Eodds** | $0.0971 \pm 0.0026$ | $0.1872 \pm 0.0067$ | $-0.1896 \pm 0.0056$ |
| Constant | $0.499 \pm 0.0$ | **Eodds** | $0.0 \pm 0.0$ | $0.0 \pm 0.0$ | $0.0 \pm 0.0$ |
| Weighted ERM | $0.869 \pm 0.0039$ | **Eodds** | $0.0971 \pm 0.0023$ | $0.1869 \pm 0.0063$ | $-0.1894 \pm 0.0051$ |
| Constrained | $0.497 \pm 0.003$ | **Eodds** | $0.0 \pm 0.0$ | $0.0 \pm 0.0$ | $0.0 \pm 0.0$ |
| BiFair | $0.873 \pm 0.004$ | **Eodds** | $0.113 \pm 0.004$ | $0.21 \pm 0.007$ | $-0.213 \pm 0.004$ |
| FairBatch | $0.8649 \pm 0.0025$ | **Eodds** | $0.0902 \pm 0.0035$ | $0.1717 \pm 0.0046$ | $-0.1719 \pm 0.0079$ |
| Reduction | $0.6241 \pm 0.054$ | **Eodds** | $0.0632 \pm 0.0164$ | $0.0732 \pm 0.0198$ | $-0.074 \pm 0.0226$ |
| FairGrad | $0.8459 \pm 0.01$ | **Eodds** | $0.0786 \pm 0.0051$ | $0.1504 \pm 0.0102$ | $-0.1527 \pm 0.0142$ |
| Unconstrained | $0.8598 \pm 0.0121$ | **Eopp** | $0.0928 \pm 0.0012$ | $0.1845 \pm 0.0041$ | $-0.1869 \pm 0.0041$ |
| Constant | $0.498 \pm 0.0$ | **Eopp** | $0.0 \pm 0.0$ | $0.0 \pm 0.0$ | $0.0 \pm 0.0$ |
| Weighted ERM | $0.8599 \pm 0.0121$ | **Eopp** | $0.0931 \pm 0.0011$ | $0.1849 \pm 0.004$ | $-0.1874 \pm 0.004$ |
| Constrained | $0.698 \pm 0.005$ | **Eopp** | $0.004 \pm 0.002$ | $0.008 \pm 0.005$ | $0.0 \pm 0.0$ |
| BiFair | $0.863 \pm 0.009$ | **Eopp** | $0.1 \pm 0.003$ | $0.2 \pm 0.007$ | $-0.202 \pm 0.006$ |
| FairBatch | $0.8635 \pm 0.0024$ | **Eopp** | $0.085 \pm 0.0023$ | $0.17 \pm 0.0032$ | $-0.1702 \pm 0.0065$ |
| Reduction | $0.6251 \pm 0.0355$ | **Eopp** | $0.0189 \pm 0.0138$ | $0.0379 \pm 0.0271$ | $-0.0378 \pm 0.0282$ |
| FairGrad | $0.8431 \pm 0.0065$ | **Eopp** | $0.0752 \pm 0.0043$ | $0.1494 \pm 0.0087$ | $-0.1514 \pm 0.0094$ |

Table 19: Results for the Gaussian dataset with Non Linear Models. All the results are averaged over 5 runs. Here MEAN ABS., MAXIMUM, and MINIMUM represent the mean absolute fairness value, the fairness level of the most well-off group, and the fairness level of the worst-off group, respectively.

| METHOD (NL) | ACCURACY ↑ | FAIRNESS | | | |
|---|---|---|---|---|---|
| | | MEASURE | MEAN ABS. ↓ | MAXIMUM | MINIMUM |
| Unconstrained | 0.88 ± 0.0038 | **AP** | 0.0897 ± 0.0045 | 0.0888 ± 0.0035 | -0.0905 ± 0.0055 |
| Constant | 0.497 ± 0.0 | **AP** | 0.001 ± 0.0 | 0.001 ± 0.0 | 0.001 ± 0.0 |
| Weighted ERM | 0.8809 ± 0.0048 | **AP** | 0.0903 ± 0.0045 | 0.0894 ± 0.0033 | -0.0911 ± 0.0057 |
| Adversarial | 0.8725 ± 0.0115 | **AP** | 0.0858 ± 0.0077 | 0.0851 ± 0.0076 | -0.0866 ± 0.0081 |
| Reduction | 0.718 ± 0.0251 | **AP** | 0.0694 ± 0.0237 | 0.0699 ± 0.0236 | -0.0689 ± 0.0239 |
| FairGrad | 0.8542 ± 0.0047 | **AP** | 0.0352 ± 0.0047 | 0.0349 ± 0.0048 | -0.0355 ± 0.0046 |
| Unconstrained | 0.8814 ± 0.0024 | **Eodds** | 0.093 ± 0.0032 | 0.1807 ± 0.0066 | -0.183 ± 0.005 |
| Constant | 0.499 ± 0.0 | **Eodds** | 0.0 ± 0.0 | 0.0 ± 0.0 | 0.0 ± 0.0 |
| Weighted ERM | 0.8821 ± 0.0031 | **Eodds** | 0.0939 ± 0.0013 | 0.1826 ± 0.0042 | -0.185 ± 0.0033 |
| Adversarial | 0.8775 ± 0.0091 | **Eodds** | 0.0852 ± 0.007 | 0.1643 ± 0.0125 | -0.1666 ± 0.0146 |
| BiFair | 0.868 ± 0.013 | **Eodds** | 0.092 ± 0.011 | 0.167 ± 0.035 | -0.168 ± 0.031 |
| FairBatch | 0.8735 ± 0.0032 | **Eodds** | 0.0749 ± 0.0041 | 0.1455 ± 0.0059 | -0.1456 ± 0.0056 |
| Reduction | 0.7309 ± 0.0189 | **Eodds** | 0.0262 ± 0.0141 | 0.0438 ± 0.0257 | -0.0435 ± 0.0265 |
| FairGrad | 0.8539 ± 0.0056 | **Eodds** | 0.0596 ± 0.0068 | 0.1013 ± 0.0147 | -0.1025 ± 0.0144 |
| Unconstrained | 0.8801 ± 0.004 | **Eopp** | 0.0902 ± 0.0017 | 0.1792 ± 0.0041 | -0.1816 ± 0.0053 |
| Constant | 0.498 ± 0.0 | **Eopp** | 0.0 ± 0.0 | 0.0 ± 0.0 | 0.0 ± 0.0 |
| Weighted ERM | 0.8805 ± 0.0046 | **Eopp** | 0.0912 ± 0.0008 | 0.1812 ± 0.0024 | -0.1837 ± 0.0045 |
| Adversarial | 0.8754 ± 0.0086 | **Eopp** | 0.0808 ± 0.0066 | 0.1605 ± 0.0128 | -0.1628 ± 0.0143 |
| BiFair | 0.88 ± 0.003 | **Eopp** | 0.086 ± 0.005 | 0.17 ± 0.013 | -0.172 ± 0.009 |
| FairBatch | 0.874 ± 0.0035 | **Eopp** | 0.0733 ± 0.0029 | 0.1465 ± 0.0054 | -0.1467 ± 0.0066 |
| Reduction | 0.6868 ± 0.0234 | **Eopp** | 0.0505 ± 0.0179 | 0.1015 ± 0.0359 | -0.1005 ± 0.036 |
| FairGrad | 0.8543 ± 0.0082 | **Eopp** | 0.0517 ± 0.0095 | 0.1028 ± 0.0191 | -0.1041 ± 0.0192 |

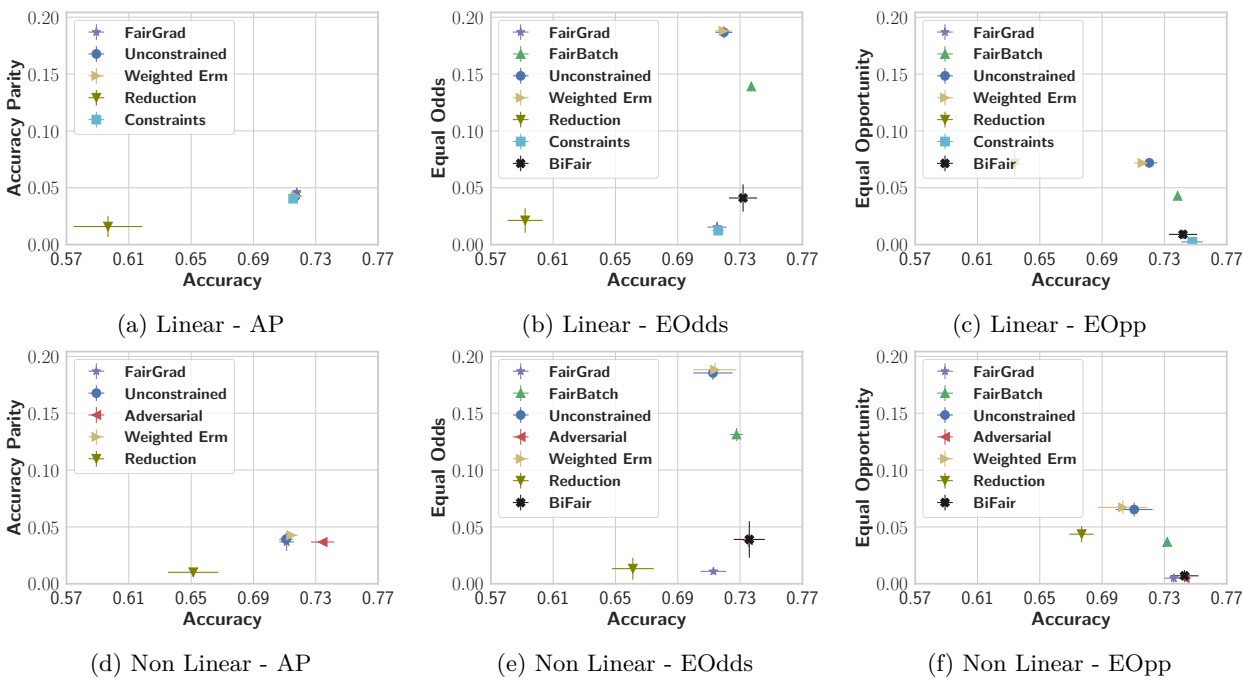

Figure 14: Results for the Twitter Sentiment dataset with different fairness measures.

Table 20: Results for the Twitter Sentiment dataset with Linear Models. All the results are averaged over 5 runs. Here MEAN ABS., MAXIMUM, and MINIMUM represent the mean absolute fairness value, the fairness level of the most well-off group, and the fairness level of the worst-off group, respectively.

| METHOD (L) | ACCURACY ↑ | | FAIRNESS | | |
|---|---|---|---|---|---|
| | | MEASURE | MEAN ABS. ↓ | MAXIMUM | MINIMUM |
| Unconstrained | 0.7211 ± 0.004 | **AP** | 0.0426 ± 0.0011 | 0.0426 ± 0.0011 | -0.0426 ± 0.0011 |
| Constant | 0.5 ± 0.0 | **AP** | 0.0 ± 0.0 | 0.0 ± 0.0 | 0.0 ± 0.0 |
| Weighted ERM | 0.7212 ± 0.0044 | **AP** | 0.0426 ± 0.0011 | 0.0426 ± 0.0011 | -0.0426 ± 0.0011 |
| Constrained | 0.72 ± 0.002 | **AP** | 0.04 ± 0.003 | 0.04 ± 0.003 | 0.04 ± 0.003 |
| Reduction | 0.6008 ± 0.022 | **AP** | 0.0159 ± 0.0092 | 0.0159 ± 0.0092 | -0.0159 ± 0.0092 |
| FairGrad | 0.7219 ± 0.0027 | **AP** | 0.0462 ± 0.0021 | 0.0462 ± 0.0021 | -0.0462 ± 0.0021 |
| Unconstrained | 0.7237 ± 0.0054 | **Eodds** | 0.1867 ± 0.0052 | 0.2287 ± 0.0078 | -0.2288 ± 0.0078 |
| Constant | 0.5 ± 0.0 | **Eodds** | 0.0 ± 0.0 | 0.0 ± 0.0 | 0.0 ± 0.0 |
| Weighted ERM | 0.7234 ± 0.0054 | **Eodds** | 0.188 ± 0.0033 | 0.2314 ± 0.0056 | -0.2315 ± 0.0056 |
| Constrained | 0.72 ± 0.004 | **Eodds** | 0.012 ± 0.002 | 0.019 ± 0.005 | 0.006 ± 0.005 |
| BiFair | 0.736 ± 0.009 | **Eodds** | 0.041 ± 0.012 | 0.056 ± 0.022 | -0.056 ± 0.022 |
| FairBatch | 0.7413 ± 0.0014 | **Eodds** | 0.1391 ± 0.0043 | 0.1755 ± 0.0084 | -0.1756 ± 0.0084 |
| Reduction | 0.5962 ± 0.0113 | **Eodds** | 0.0213 ± 0.0108 | 0.0314 ± 0.0211 | -0.0314 ± 0.021 |
| FairGrad | 0.7193 ± 0.0062 | **Eodds** | 0.0154 ± 0.0051 | 0.0204 ± 0.0098 | -0.0204 ± 0.0098 |
| Unconstrained | 0.7244 ± 0.0051 | **Eopp** | 0.0719 ± 0.0012 | 0.1439 ± 0.0023 | -0.1438 ± 0.0023 |
| Constant | 0.5 ± 0.0 | **Eopp** | 0.0 ± 0.0 | 0.0 ± 0.0 | 0.0 ± 0.0 |
| Weighted ERM | 0.72 ± 0.0054 | **Eopp** | 0.0718 ± 0.0013 | 0.1437 ± 0.0026 | -0.1436 ± 0.0026 |
| Constrained | 0.752 ± 0.004 | **Eopp** | 0.002 ± 0.001 | 0.005 ± 0.001 | 0.0 ± 0.0 |
| BiFair | 0.746 ± 0.009 | **Eopp** | 0.009 ± 0.004 | 0.017 ± 0.009 | -0.017 ± 0.009 |
| FairBatch | 0.7426 ± 0.001 | **Eopp** | 0.0429 ± 0.0005 | 0.0858 ± 0.0011 | -0.0858 ± 0.0011 |
| Reduction | 0.6381 ± 0.0039 | **Eopp** | 0.0712 ± 0.0117 | 0.1424 ± 0.0234 | -0.1425 ± 0.0234 |
| FairGrad | 0.7518 ± 0.0069 | **Eopp** | 0.0024 ± 0.002 | 0.0049 ± 0.004 | -0.0049 ± 0.004 |

Table 21: Results for the Twitter Sentiment dataset with Non Linear Models. All the results are averaged over 5 runs. Here MEAN ABS., MAXIMUM, and MINIMUM represent the mean absolute fairness value, the fairness level of the most well-off group, and the fairness level of the worst-off group, respectively.

| METHOD (NL) | ACCURACY ↑ | FAIRNESS | | | |
|---|---|---|---|---|---|
| | | MEASURE | MEAN ABS. ↓ | MAXIMUM | MINIMUM |
| Unconstrained | $0.715 \pm 0.0043$ | **AP** | $0.0392 \pm 0.0055$ | $0.0392 \pm 0.0055$ | $-0.0392 \pm 0.0055$ |
| Constant | $0.5 \pm 0.0$ | **AP** | $0.0 \pm 0.0$ | $0.0 \pm 0.0$ | $0.0 \pm 0.0$ |
| Weighted ERM | $0.7183 \pm 0.0042$ | **AP** | $0.0427 \pm 0.0019$ | $0.0427 \pm 0.0019$ | $-0.0427 \pm 0.0019$ |
| Adversarial | $0.7385 \pm 0.0075$ | **AP** | $0.0367 \pm 0.0027$ | $0.0367 \pm 0.0027$ | $-0.0368 \pm 0.0027$ |
| Reduction | $0.6555 \pm 0.0162$ | **AP** | $0.0101 \pm 0.0038$ | $0.0101 \pm 0.0038$ | $-0.0101 \pm 0.0038$ |
| FairGrad | $0.7154 \pm 0.0047$ | **AP** | $0.0368 \pm 0.0079$ | $0.0367 \pm 0.0078$ | $-0.0368 \pm 0.0079$ |
| Unconstrained | $0.7167 \pm 0.0126$ | **Eodds** | $0.1854 \pm 0.0061$ | $0.2349 \pm 0.0091$ | $-0.235 \pm 0.0091$ |
| Constant | $0.5 \pm 0.0$ | **Eodds** | $0.0 \pm 0.0$ | $0.0 \pm 0.0$ | $0.0 \pm 0.0$ |
| Weighted ERM | $0.718 \pm 0.0137$ | **Eodds** | $0.1882 \pm 0.0062$ | $0.2379 \pm 0.0073$ | $-0.2381 \pm 0.0073$ |
| Adversarial | $0.7393 \pm 0.0024$ | **Eodds** | $0.0382 \pm 0.0056$ | $0.06 \pm 0.0151$ | $-0.06 \pm 0.0151$ |
| BiFair | $0.74 \pm 0.01$ | **Eodds** | $0.039 \pm 0.016$ | $0.058 \pm 0.017$ | $-0.058 \pm 0.017$ |
| FairBatch | $0.7318 \pm 0.004$ | **Eodds** | $0.1313 \pm 0.0057$ | $0.1724 \pm 0.0055$ | $-0.1725 \pm 0.0055$ |
| Reduction | $0.6653 \pm 0.0134$ | **Eodds** | $0.0133 \pm 0.0097$ | $0.0199 \pm 0.0172$ | $-0.0199 \pm 0.0173$ |
| FairGrad | $0.717 \pm 0.0082$ | **Eodds** | $0.0109 \pm 0.0027$ | $0.0165 \pm 0.0053$ | $-0.0165 \pm 0.0053$ |
| Unconstrained | $0.7147 \pm 0.0118$ | **Eopp** | $0.0653 \pm 0.0062$ | $0.1306 \pm 0.0124$ | $-0.1306 \pm 0.0124$ |
| Constant | $0.5 \pm 0.0$ | **Eopp** | $0.0 \pm 0.0$ | $0.0 \pm 0.0$ | $0.0 \pm 0.0$ |
| Weighted ERM | $0.7074 \pm 0.0158$ | **Eopp** | $0.0672 \pm 0.0062$ | $0.1346 \pm 0.0125$ | $-0.1345 \pm 0.0125$ |
| Adversarial | $0.7471 \pm 0.0042$ | **Eopp** | $0.005 \pm 0.0035$ | $0.0099 \pm 0.007$ | $-0.0099 \pm 0.007$ |
| BiFair | $0.747 \pm 0.009$ | **Eopp** | $0.007 \pm 0.005$ | $0.013 \pm 0.01$ | $-0.013 \pm 0.01$ |
| FairBatch | $0.7359 \pm 0.0011$ | **Eopp** | $0.0368 \pm 0.0012$ | $0.0736 \pm 0.0025$ | $-0.0736 \pm 0.0025$ |
| Reduction | $0.681 \pm 0.0078$ | **Eopp** | $0.0436 \pm 0.0071$ | $0.0871 \pm 0.0143$ | $-0.0871 \pm 0.0143$ |
| FairGrad | $0.7401 \pm 0.0059$ | **Eopp** | $0.0049 \pm 0.0041$ | $0.0099 \pm 0.0083$ | $-0.0099 \pm 0.0083$ |

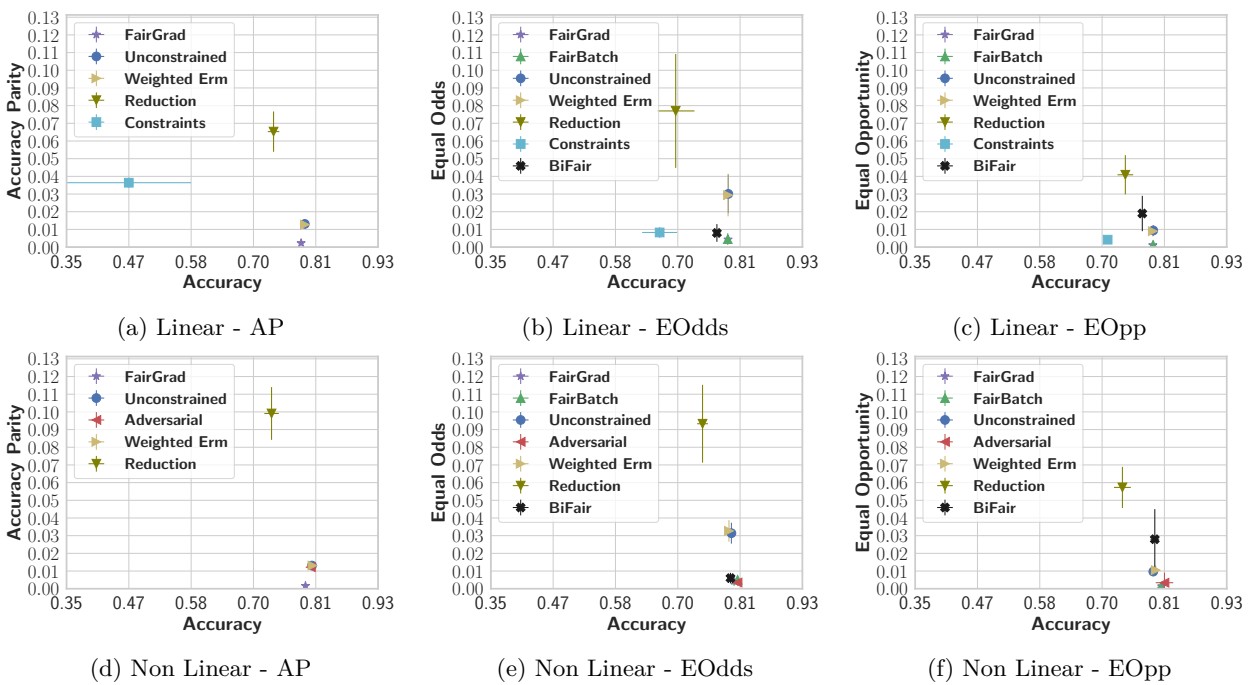

Figure 15: Results for the Folktables Adult dataset with different fairness measures.

Table 22: Results for the Folktables Adult dataset with Linear Models. All the results are averaged over 5 runs. Here MEAN ABS., MAXIMUM, and MINIMUM represent the mean absolute fairness value, the fairness level of the most well-off group, and the fairness level of the worst-off group, respectively.

| METHOD (L) | ACCURACY ↑ | FAIRNESS | | | |
|---|---|---|---|---|---|
| | | MEASURE | MEAN ABS. ↓ | MAXIMUM | MINIMUM |
| Unconstrained | $0.7905 \pm 0.0033$ | **AP** | $0.0131 \pm 0.0021$ | $0.0123 \pm 0.0021$ | $-0.0138 \pm 0.0022$ |
| Constant | $0.666 \pm 0.0$ | **AP** | $0.053 \pm 0.0$ | $0.056 \pm 0.0$ | $0.051 \pm 0.0$ |
| Weighted ERM | $0.7906 \pm 0.0032$ | **AP** | $0.0127 \pm 0.0023$ | $0.0119 \pm 0.0022$ | $-0.0134 \pm 0.0024$ |
| Constrained | $0.467 \pm 0.115$ | **AP** | $0.036 \pm 0.003$ | $0.039 \pm 0.003$ | $0.034 \pm 0.003$ |
| Reduction | $0.733 \pm 0.0106$ | **AP** | $0.0653 \pm 0.0114$ | $0.0614 \pm 0.011$ | $-0.0692 \pm 0.0118$ |
| FairGrad | $0.7837 \pm 0.0049$ | **AP** | $0.0023 \pm 0.0009$ | $0.0023 \pm 0.001$ | $-0.0022 \pm 0.0008$ |
| Unconstrained | $0.789 \pm 0.0026$ | **Eodds** | $0.0301 \pm 0.011$ | $0.0377 \pm 0.0153$ | $-0.0458 \pm 0.0184$ |
| Constant | $0.667 \pm 0.0$ | **Eodds** | $0.0 \pm 0.0$ | $0.0 \pm 0.0$ | $0.0 \pm 0.0$ |
| Weighted ERM | $0.7886 \pm 0.0032$ | **Eodds** | $0.0294 \pm 0.012$ | $0.0364 \pm 0.0169$ | $-0.0443 \pm 0.0206$ |
| Constrained | $0.663 \pm 0.032$ | **Eodds** | $0.008 \pm 0.003$ | $0.013 \pm 0.004$ | $0.004 \pm 0.002$ |
| BiFair | $0.768 \pm 0.007$ | **Eodds** | $0.008 \pm 0.005$ | $0.011 \pm 0.006$ | $-0.011 \pm 0.008$ |
| FairBatch | $0.788 \pm 0.0027$ | **Eodds** | $0.0045 \pm 0.0033$ | $0.0069 \pm 0.0065$ | $-0.0063 \pm 0.0049$ |
| Reduction | $0.6922 \pm 0.0346$ | **Eodds** | $0.077 \pm 0.0322$ | $0.0761 \pm 0.0257$ | $-0.0903 \pm 0.0378$ |
| FairGrad | $0.7885 \pm 0.0027$ | **Eodds** | $0.0043 \pm 0.0019$ | $0.0073 \pm 0.0037$ | $-0.0068 \pm 0.0045$ |
| Unconstrained | $0.7902 \pm 0.0038$ | **Eopp** | $0.0094 \pm 0.0031$ | $0.0162 \pm 0.0053$ | $-0.0215 \pm 0.0071$ |
| Constant | $0.667 \pm 0.0$ | **Eopp** | $0.0 \pm 0.0$ | $0.0 \pm 0.0$ | $0.0 \pm 0.0$ |
| Weighted ERM | $0.7893 \pm 0.0031$ | **Eopp** | $0.009 \pm 0.003$ | $0.0155 \pm 0.0051$ | $-0.0206 \pm 0.0069$ |
| Constrained | $0.706 \pm 0.002$ | **Eopp** | $0.004 \pm 0.0$ | $0.01 \pm 0.001$ | $0.0 \pm 0.0$ |
| BiFair | $0.77 \pm 0.002$ | **Eopp** | $0.019 \pm 0.01$ | $0.033 \pm 0.017$ | $-0.044 \pm 0.023$ |
| FairBatch | $0.79 \pm 0.0031$ | **Eopp** | $0.0012 \pm 0.0015$ | $0.0022 \pm 0.0026$ | $-0.0026 \pm 0.0034$ |
| Reduction | $0.7388 \pm 0.0144$ | **Eopp** | $0.0409 \pm 0.0111$ | $0.0932 \pm 0.025$ | $-0.0704 \pm 0.0194$ |
| FairGrad | $0.7893 \pm 0.0026$ | **Eopp** | $0.0011 \pm 0.0009$ | $0.0024 \pm 0.002$ | $-0.0021 \pm 0.0016$ |

Table 23: Results for the Folktables Adult dataset with Non Linear Models. All the results are averaged over 5 runs. Here MEAN ABS., MAXIMUM, and MINIMUM represent the mean absolute fairness value, the fairness level of the most well-off group, and the fairness level of the worst-off group, respectively.

| METHOD (NL) | ACCURACY ↑ | FAIRNESS | | | |
|---|---|---|---|---|---|
| | | MEASURE | MEAN ABS. ↓ | MAXIMUM | MINIMUM |
| Unconstrained | $0.8037 \pm 0.0037$ | **AP** | $0.0131 \pm 0.0017$ | $0.0123 \pm 0.0016$ | $-0.0139 \pm 0.0017$ |
| Constant | $0.666 \pm 0.0$ | **AP** | $0.053 \pm 0.0$ | $0.056 \pm 0.0$ | $0.051 \pm 0.0$ |
| Weighted ERM | $0.8046 \pm 0.0049$ | **AP** | $0.0131 \pm 0.0014$ | $0.0123 \pm 0.0014$ | $-0.0138 \pm 0.0015$ |
| Adversarial | $0.8016 \pm 0.0053$ | **AP** | $0.0122 \pm 0.0016$ | $0.0115 \pm 0.0015$ | $-0.0129 \pm 0.0016$ |
| Reduction | $0.7293 \pm 0.0133$ | **AP** | $0.0991 \pm 0.0149$ | $0.0932 \pm 0.0139$ | $-0.1051 \pm 0.016$ |
| FairGrad | $0.7917 \pm 0.0025$ | **AP** | $0.0016 \pm 0.0011$ | $0.0016 \pm 0.0011$ | $-0.0016 \pm 0.001$ |
| Unconstrained | $0.7947 \pm 0.0078$ | **Eodds** | $0.0314 \pm 0.0059$ | $0.0373 \pm 0.0058$ | $-0.0454 \pm 0.0066$ |
| Constant | $0.667 \pm 0.0$ | **Eodds** | $0.0 \pm 0.0$ | $0.0 \pm 0.0$ | $0.0 \pm 0.0$ |
| Weighted ERM | $0.7902 \pm 0.0049$ | **Eodds** | $0.0327 \pm 0.0061$ | $0.04 \pm 0.0067$ | $-0.0488 \pm 0.0077$ |
| Adversarial | $0.806 \pm 0.0047$ | **Eodds** | $0.0035 \pm 0.0018$ | $0.0051 \pm 0.0021$ | $-0.0053 \pm 0.0028$ |
| BiFair | $0.793 \pm 0.006$ | **Eodds** | $0.006 \pm 0.003$ | $0.007 \pm 0.003$ | $-0.007 \pm 0.004$ |
| FairBatch | $0.8061 \pm 0.0044$ | **Eodds** | $0.0051 \pm 0.0015$ | $0.0087 \pm 0.0048$ | $-0.0084 \pm 0.0029$ |
| Reduction | $0.7416 \pm 0.01$ | **Eodds** | $0.0933 \pm 0.022$ | $0.1517 \pm 0.0311$ | $-0.1244 \pm 0.026$ |
| FairGrad | $0.7997 \pm 0.0087$ | **Eodds** | $0.0045 \pm 0.0029$ | $0.0067 \pm 0.0045$ | $-0.0071 \pm 0.0058$ |
| Unconstrained | $0.7902 \pm 0.0044$ | **Eopp** | $0.0097 \pm 0.0026$ | $0.0168 \pm 0.0045$ | $-0.0222 \pm 0.006$ |
| Constant | $0.667 \pm 0.0$ | **Eopp** | $0.0 \pm 0.0$ | $0.0 \pm 0.0$ | $0.0 \pm 0.0$ |
| Weighted ERM | $0.7947 \pm 0.0022$ | **Eopp** | $0.0105 \pm 0.0027$ | $0.0181 \pm 0.0047$ | $-0.024 \pm 0.0062$ |
| Adversarial | $0.8108 \pm 0.0161$ | **Eopp** | $0.0034 \pm 0.0057$ | $0.0041 \pm 0.0057$ | $-0.0095 \pm 0.017$ |
| BiFair | $0.793 \pm 0.008$ | **Eopp** | $0.028 \pm 0.017$ | $0.048 \pm 0.029$ | $-0.064 \pm 0.039$ |
| FairBatch | $0.8038 \pm 0.0063$ | **Eopp** | $0.0008 \pm 0.0005$ | $0.0014 \pm 0.0009$ | $-0.0018 \pm 0.0012$ |
| Reduction | $0.7334 \pm 0.0155$ | **Eopp** | $0.0573 \pm 0.0116$ | $0.1307 \pm 0.0265$ | $-0.0986 \pm 0.0199$ |
| FairGrad | $0.8058 \pm 0.0035$ | **Eopp** | $0.0014 \pm 0.0014$ | $0.003 \pm 0.0031$ | $-0.0026 \pm 0.0024$ |

