# OpenReview forum: "FairGrad: Fairness Aware Gradient Descent"
_TMLR — Accepted by TMLR_

### Review · Reviewer_Akhq · 2023-06-11

**Summary Of Contributions:**

The paper proposes a new in-processing approach, called FairGrad, to enforce statistical group fairness notions such as error parity, equality of opportunity, and equalized odds.

**Audience:**

Yes

**Broader Impact Concerns:**

Broader impacts are addressed in a satisfactory manner.

**Claims And Evidence:**

Yes

**Requested Changes:**

I would appreciate it if the paper addresses the following issues:

-- Are there any situations in which FairGrad outperforms the existing in-processing approach to achieve a better fairness/accuracy tradeoff?

-- A major omitted baseline is the reduction approach of Agarwal et al., 2018 which is perhaps the most commonly used in-processing baseline in the literature.

-- The paper misses some crucial discussion on in-processing approaches for an arbitrary number of groups e.g., https://proceedings.mlr.press/v80/hebert-johnson18a.html and https://proceedings.mlr.press/v80/kearns18a.html.

-- Many of the experimental details are unclear. For example, what is the non-linear model used in the experiments? How does the performance change as a function of different model classes? What is the justification for using linear models?

**Strengths And Weaknesses:**

The main strengths of the paper are as follows:

--The proposed approach is very simple.

--The proposed approach can deal with multi-class classification, more than two subgroups, and can handle several notions of fairness.


However, the paper suffers from some shortcomings:

--Given the plethora of papers dealing with proposing new in-processing methods to provide group fairness, it is unclear how much the new approach can add to the existing approaches. In particular, the approach does not meaningfully outperform previous in-processing methods as shown in the experiments.

-- Given that the major contribution of the paper is empirical, there are a few major issues with the experiments. In particular, some baselines are not included, and also some experimental details and rationale behind the choices are not clear in the paper (see the Requested Changes for more details).

---

### Review · Reviewer_EMqe · 2023-06-19

**Summary Of Contributions:**

This paper proposes a modification to gradient descent algorithms to incorporate/enforce fairness constraints. The procedure follows closely in spirit with some modifications to the reductions-based algorithms for solving fairness-constrained optimization problems in Agarwal et al. (2018) and Cotter et al. (2019).

For a particular class of fairness definitions, the authors consider solving a standard fairness constrained optimization problem that they re-interpret as an unconstrained problem with some lagrange multipliers. As in Agarwal et al. (2018) and Cotter et al. (2019), the authors propose to solve this unconstrained problem by searching for saddle points via an iterative search -- in each step, first the $\lambda$-player moves to select $\lambda$ to drive up the objective function, and then the model player updates the prediction function's weights to lower loss. The author's main insight is that for the class of fairness definitions considered in the paper, the min-player's objective function is just a particular weighted average of group-specific losses. As a result, we can also apply take gradient descent steps to optimize the min-player's objective function. Altogether, the author's main contribution effectively is to p ropose that we can solve fairness constrained optimization problems by applying a gradient descent ascent algorithm with an objective function that updates at each step. Such algorithms have been extensively analyzed in a literature on adversarial learning such as Madry et al. (2019).

References:
Madry et al. (2019): Towards Deep Learning Models Resistant to Adversarial Attacks

**Audience:**

Yes

**Claims And Evidence:**

Yes

**Requested Changes:**

(1) I don't think it's fair to write ``They are also limited in the range of problems to which they can be applied. For example, the work
of Agarwal et al. (2018) can only be applied in a binary classification setting...'' in the introduction. This a correct statement about the specific paper (Agarwal et al. 2018) but the broader in-processing literature certainly provides extensions of these algorithms to richer settings. See, for example, Agarwal et al. (2019) that extends these procedures to cases with a continuous outcome. By contrast, the main text of this paper focuses on a case with discrete outcomes.

(2) In the main text, you should clarify that the group definitions Tk can also depend on the true label y. This is important with how you can nest definitions like equalized odds into the fairness definition.

(3) Per the weaknesses above, I think the authors need to incorporate a more up-front discussion about how their analysis is limited relative to existing work on fairness-constrained optimization.

(4) The authors should discuss how their proposed algorithm is closely connected to procedures that are popular in adversarial learning (e.g., see Madry et al. (2019) and related citations). I view the author's main contributions here as showing that we can apply the gradient descent-ascent optimization procedures that are popular in adversarial learning to do fairness constrained optimization over the particular combination of loss function and fairness definitions considered.

References:
Agarwal et al. (2019): Fair Regression: Quantitative Definitions and Reduction-based Algorithms

**Strengths And Weaknesses:**

Strengths:
(1) FairGrad is a simple gradient descent algorithm for solving a wide class of fairness constrained optimization problems. The experimental results suggest that it works well in a variety of settings and is computationally more efficient than competing methods.

Weakness:

(1) Throughout the paper, the authors assume "we will assume that what was measured on our finite sample is sufficiently close to what
would be obtained if one had access to the overall distribution". In other words, their analysis of FairGrad provides no guarantees about its generalization to the true underlying distribution F(). This stands in contrast to the literature on fairness-constrained optimization problems, where, for example, papers like Agarwal et al. (2018) and Agarwal et al. (2019) provide generalization bounds on their returned solutions (i.e., they are close to the true population optimum with high probability).

(2) In light of (1), the paper can be thought of as just providing a particular optimization routine for the sampled data more in the spirit of the analysis in Cotter et al. (2019). However, here too the authors analysis is lacking -- in particular, Cotter et al. (2019) are able to provide some optimization guarantees about when their reductions based approach will in fact converge (i.e., how must the hyper-parameters be set, how many iterations must be run etc).

---

### Review · Reviewer_C7Px · 2023-06-19

**Summary Of Contributions:**

The paper presents FairGrad, a method for inducing fairness in gradient-based algorithms. Fairness here is defined as loss functions compared across groups, and the FairGrad method reweights gradients based on the group membership of the associated individual. The paper presents empirical results on four datasets and show that the FairGrad method has comparable and at times better results than other baselines.


**Audience:**

Yes

**Broader Impact Concerns:**

Existing Limitations and Social Impact section is sufficient.

**Claims And Evidence:**

Yes

**Requested Changes:**

More guidance about when to use FairGrad compared to alternative fairness methods would strengthen the paper. As it stands, there are a variety of fairness methods who have similar accuracy-fairness tradeoffs. Highlighting what FairGrad contributes to the landscape will help the community leverage the method.


**Strengths And Weaknesses:**

S1) The empirical results are thorough and show extensive results with four datasets, (up to) five baselines, and four fairness metrics. The empirical results on batch size and epsilon values is helpful for better understanding the effects of these parameters.

W1) My biggest question of the paper is that it's is unclear where to place FairGrad within the pantheon of existing fairness algorithms. As explained in Section 4.4, there is no fairness method that is consistently better than the others over all of different datasets. Why should we consider FairGrad compared to the others?

The introduction claims that the "complexity of the existing models" is the main obstacle for existing methods. Ease of implementation is a key selling point for FairGrad, but it appears that the other models are merely missing an easily accessilbe code package? I had originally thought that FairGrad's strength would be that it could be used for off-the-shelf algorithms as opposed to having to know the loss function, but it seems that other methods have that as well.

I had thought that FairGrad was more computationally efficient than other algorithms, hence one of its strengths. Despite Table 3 going in depth about computational speed, the paper doesn't mention FairGrad as being superior in runtime than other methods.

Some clarify around this would be helpful.

Small comment that Algorithm 1 line 5 is italicized and it's unclear why.

---

### Author Response · Authors · 2023-06-27
**Addressing general comments of the reviewers**

We would like to thank all the reviewers for their careful reviews and the valuable insights that will help us improve the paper. We answer their main concerns individually below and will update the paper in the next few days to reflect the promised changes. We kindly ask the reviewers to let us know if we missed some of their points or if our comments and the other reviews raised additional questions.

---

> ### Author Response · Authors · 2023-06-29
>
> We have updated the manuscript integrating the changes as requested by the reviewers. Please find the list of changes in the section above, and let us know if we missed any suggested changes. Once again, we would like to thank all the reviewers for their comments and valuable insights.

---

### Decision · Action_Editors · 2023-08-03

**Recommendation:** Accept as is

**Comment:**

The three reviewers have positive views on this manuscript. The revised version has incorporated reviewers’ comments including adding experimental results related of Agarwal et al., 2018, and adding discussion on when the proposed method outperforms other existing in-processing approaches to achieve a better fairness/accuracy tradeoff.

**Audience:**

Yes, algorithmic fairness researchers and practitioners would be interested. The proposed method can be implemented by simply replacing an existing loss from PyTorch with their provided custom loss (provided in the supplementary material) and passing along some meta data, while the rest of the training loop remains identical.

**Claims And Evidence:**

The manuscript claims to propose a method to enforce group fairness that is easy to implement, accommodates various standard fairness definitions, can handle both multiple sensitive groups and multiclass problems, can fine tune existing classifiers to achieve better fairness, and comes with minimal overhead. The manuscript provides extensive experiments and shows that the proposed method is competitive with several standard baselines (inc. the fair reduction method by Agarwal et al. (2018)) in fairness on both standard datasets (Adult Income and CelebA datasets) as well as complex natural language processing (Twitter Sentiment dataset) and computer vision (UTKFace dataset) tasks. Four different fairness measures are used: Equalized Odds (EOdds), Equality of Opportunity (EOpp), Accuracy Parity (AP), and Demographic Parity (DP).